# Kinesin-1 mediates proper ER folding of the Ca$_V$1.2 channel and maintains mouse glucose homeostasis

Yosuke Tanaka [ID] [1], Atena Farkhondeh [ID] [1], Wenxing Yang[1], Hitoshi Ueno[1], Mitsuhiko Noda[2] & Nobutaka Hirokawa [ID] [1,3✉]

## Abstract

Glucose-stimulated insulin secretion (GSIS) from pancreatic beta cells is a principal mechanism for systemic glucose homeostasis, of which regulatory mechanisms are still unclear. Here we show that kinesin molecular motor KIF5B is essential for GSIS through maintaining the voltage-gated calcium channel Ca$_V$1.2 levels, by facilitating an Hsp70-to-Hsp90 chaperone exchange to pass through the quality control in the endoplasmic reticulum (ER). Phenotypic analyses of KIF5B conditional knockout (cKO) mouse beta cells revealed significant abolishment of glucose-stimulated calcium transients, which altered the behaviors of insulin granules via abnormally stabilized cortical F-actin. KIF5B and Hsp90 colocalize to microdroplets on ER sheets, where Ca$_V$1.2 but not K$_{ir}$6.2 is accumulated. In the absence of KIF5B, Ca$_V$1.2 fails to be transferred from Hsp70 to Hsp90 via STIP1, and is likely degraded via the proteasomal pathway. KIF5B and Hsc70 overexpression increased Ca$_V$1.2 expression via enhancing its chaperone binding. Thus, ER sheets may serve as the place of KIF5B- and Hsp90-dependent chaperone exchange, which predominantly facilitates Ca$_V$1.2 production in beta cells and properly enterprises GSIS against diabetes.

Keywords Kinesin; Hsp90; Insulin Secretion; Calcium Channel; ER Sheets
Subject Categories Cell Adhesion, Polarity & Cytoskeleton; Metabolism; Translation & Protein Quality

## Introduction

Diabetes is a major metabolic syndrome, which is predicted to have a >50% worldwide prevalence by 2045 (Perreault Skyler and Rosenstock, 2021). Glucose-stimulated insulin secretion (GSIS) is one of the central processes of systemic glucose homeostasis (Matschinsky, 1996). This process is regarded as a major target for therapeutic approaches to diabetes, so that detailed elucidation of stimulation-secretion-coupling is very important (Islam, 2020). The insulin secretion process largely consists of basal and glucose-stimulated secretion, the latter of which is classified into the first phase during 0–10 min and the second phase later than 10 min after the glucose stimulation (Wang and Thurmond, 2009). Basal insulin secretion is normally suppressed by hyperpolarization of the membrane potential, mainly due to K$_{ATP}$ channel activity, which is augmented by the PIP$_2$ synthesizing enzyme, PIP5Kα (de la Cruz et al, 2016; Liang et al, 2014). Na/K ATPase and voltage- and calcium-sensitive big K channel (BK$_{Ca}$) contribute to this process as well, and circumvent beta-cell hyperplasia (Arystarkhova et al, 2013; Dufer et al, 2009; Fridlyand et al, 2013). For the regulated secretion, glucose stimuli depolarize the membrane due to the K$_{ATP}$ channel closure (Rorsman and Ashcroft, 2018), and induce Ca$^{2+}$ influx mainly via the L-type voltage-gated calcium channel (VGCC) subunit Ca$_V$1.2 and the R-type subunit Ca$_V$2.3 (Rutter, 2001; Wilson et al, 2001). This Ca$^{2+}$ influx is accompanied by glucose-stimulated sequential activation of Src-family kinases (SFKs) and the Rho-family GTPases Cdc42 and Rac1, which remodel subplasmalemmal F-actin bundles and stabilize insulin granule exocytosis events especially in the second phase (Arous and Halban, 2015; Daniel et al, 2002; Kalwat and Thurmond, 2013; Wang and Thurmond, 2010; Yoder et al, 2014). These Rho-family GTPase pathway may synergize with Ca$^{2+}$-induced B-Raf/Raf1 pathways to activate PAK1/ERK kinases that directly drive the actin remodeling (Kalwat and Thurmond, 2013). Detailed molecular mechanism and the degree of involvement of Ca$^{2+}$ in each glucose-stimulated signaling component is still largely elusive (Komatsu et al, 2013; Shigeto et al, 2006).

KIF5B is the founding member of kinesin superfamily proteins (KIFs) (Hirokawa et al, 2009). This ubiquitously expressed heavy chain of conventional kinesin-1 molecular motor was first cloned from mouse pancreatic beta-cell cDNA library as a "beta cell kinesin" (Meng et al, 1997). Mice with complete knockout of this molecule are embryonic lethal (Tanaka et al, 1998). KIF5B plays an essential role in the second phase of GSIS (Cui et al, 2011; Donelan et al, 2002; Meng et al, 1997; Varadi et al, 2002; Varadi et al, 2003), but the precise molecular mechanism is still unclear. It transports various kinds of membrane organelles including mitochondria and lysosomes (Tanaka et al, 1998) and AMPA-type glutamate receptor-containing vesicles (Setou et al, 2002). This molecule also pulls out a tubular structure from the ER cisternae in vitro (Wozniak and Allan, 2006), and distributes the ER exit sites throughout the cytoplasm (Gupta et al, 2008). However, the relevance of these KIF5B-mediated processes in the functional regulation of protein folding in the ER are still largely unknown.

[1]Department of Cell Biology and Anatomy, Graduate School of Medicine, The University of Tokyo, Hongo, Tokyo 113-0033, Japan. [2]Department of Diabetes, Metabolism and Endocrinology, Ichikawa Hospital, International University of Health and Welfare, Chiba 272-0827, Japan. [3]Department of Advanced Morphological Imaging, Graduate School of Medicine, Juntendo University, 2-1-1, Hongo, Bunkyo-ku, Tokyo 113-8421, Japan. ✉E-mail: hirokawa@m.u-tokyo.ac.jp

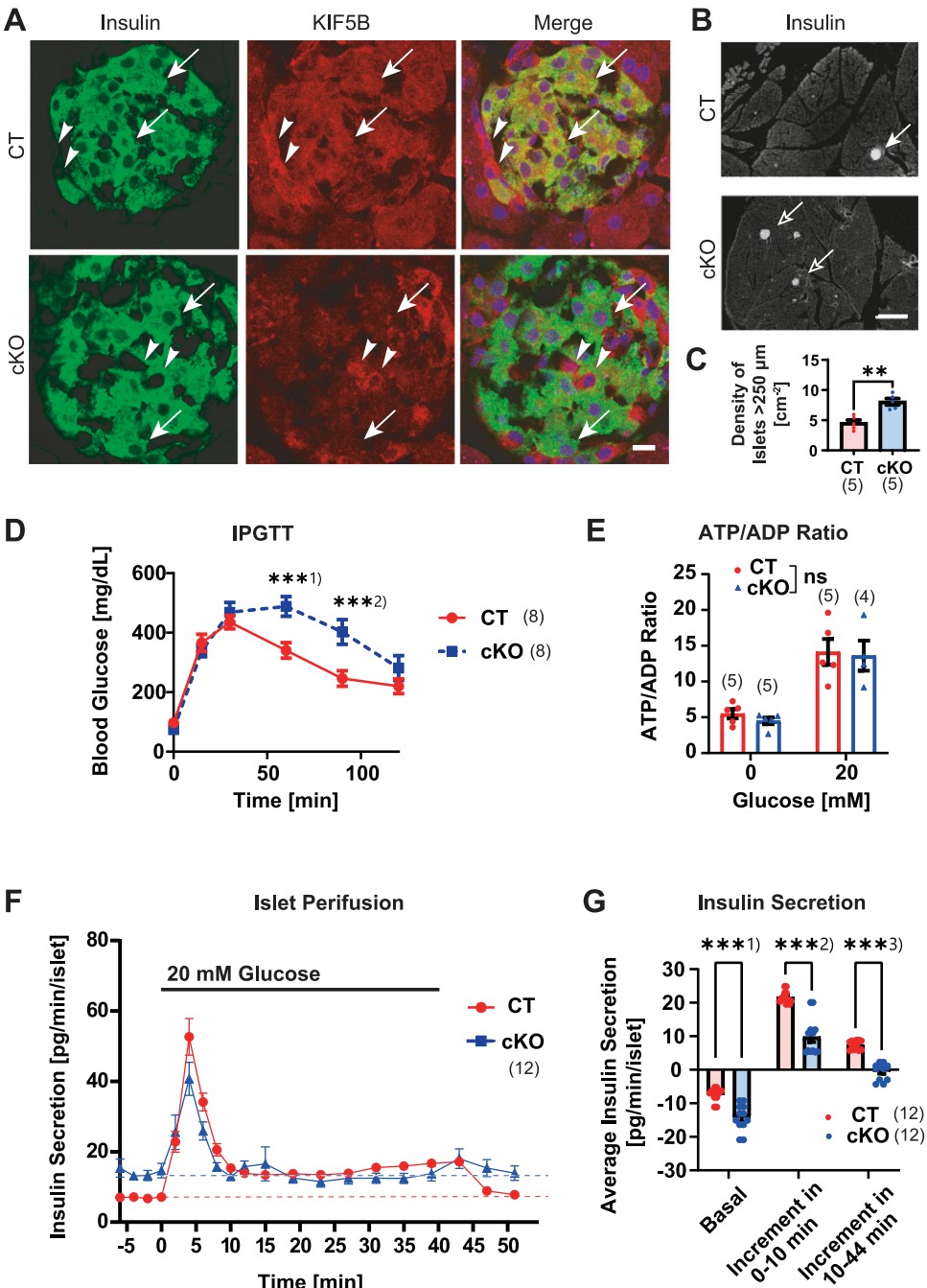

**Figure 1. KIF5B is essential for proper GSIS and blood glucose homeostasis.**

(A–C) Immunohistochemistry of *Kif5b^flox/flox Rip2-Cre^•/•* (CT) and *Kif5b^flox/flox Rip2-Cre^tg/•* (cKO) mouse pancreas (A, B); against insulin (green in (A) and gray in (B)) and/or KIF5B (red in (A)); accompanied by the islet morphometry (C). Bars, 10 μm in (A) and 1 mm in (B). **$P = 0.0014$, two-sided unpaired Welch's $t$ test, $n = 5$ mice. Data are represented by the mean ± SEM. Arrows in (A), beta cells. Arrowheads in (A), non-beta cells. Arrows in (B), islets. Corresponding to Fig. EV1. (D) Intraperitoneal glucose tolerance test (IPGTT) of 4-month-old CT and cKO mice. ***1)$P = 0.0023$; ***2)$P = 0.0045$; one-sided unpaired Welch's $t$ test, $n = 8$ mice at each time point. Data are represented by the mean ± SEM. (E) ATP/ADP ratio measurements in the islets of the indicated genotypes stimulated for 3 min. ns, $P = 0.5323$, two-way ANOVA, $n = 4$–5 biological replicates. Data are represented by the mean ± SEM. (F, G) Perifusion assay of CT and cKO mouse pancreatic islets stimulated with 20 mM glucose for 40 min (F), quantified for the respective amounts of basal secretion (plotted in an inverted manner) and the first- (0–10 min) and second phase (10–44 min) of GSIS increments. ***1)$P = 0.000060$; ***2)$P = 0.000008$; ***3)$P = 2.66 \times 10^{-8}$; two-sided unpaired Welch's $t$ test, $n = 12$ biological replicates. Data are represented by the mean ± SEM.

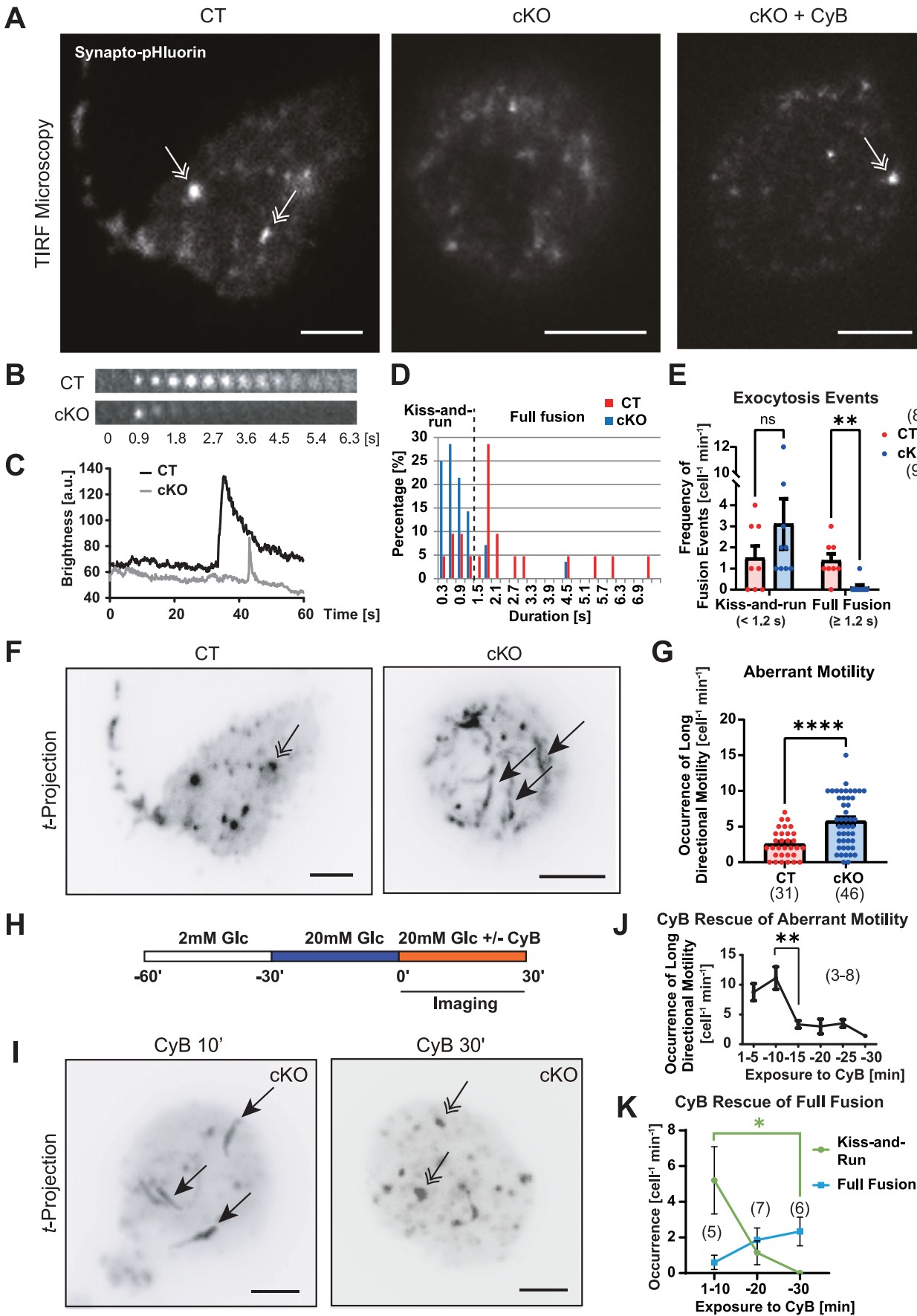

**Figure 2. KIF5B stabilizes insulin exocytosis through actin remodeling.**

(A) Typical images of the surface of synapto.pHluorin-transduced primary beta cells on the indicated conditions in TIRF microscopy. Double arrows, full fusion. CyB, 10 μg/mL cytochalasin B treatment. Scale bars, 5 μm. Corresponding to Movie EV1. (B, C) Typical time-lapse images of synapto.pHluorin exocytosis on the cell surface of the indicated genotypes by time-lapse total internal reflection fluorescence (TIRF) microscopy (B) and their typical traces of fluorescence intensity (C), recorded at 30–60 min after glucose stimulation. (D) Histogram of the duration of each exocytosis event at 30–60 min of glucose stimulation. (E) Quantification of the occurrence of full-fusion events (defined to be longer than 1.2 s in (D)) and kiss-and-run events (shorter than 1.2 s) in each genotype. Note that KIF5B deficiency significantly reduced the occurrence of full-fusion exocytosis. ns, $P = 0.2480$; **$P = 0.0054$, two-sided unpaired Welch's $t$ test, $n = 8$–9 cells. Data are represented by the mean ± SEM. (F, G) Temporal projection of the time-lapse images of CT and cKO primary beta cells (F) and its quantification for the occurrence of long directional motility (G). Scale bars, 5 μm. ****$P = 5.46 \times 10^{-6}$, two-sided unpaired Welch's $t$ test, $n = 31$–46 cells. Double arrow, full fusion; single arrows, cortical long-range directional motilities. (H–K) Time course of the pharmacological rescue of insulin granule dynamics of cKO cells by CyB treatment, represented by an experimental design (H); temporal projection of time-lapse images (I); time course of the occurrence of peripheral directional motility after CyB treatment (J), and time course of the occurrence of indicated types of exocytosis after CyB treatment (K). Scale bars, 5 μm. *$P = 0.0323$; **$P = 0.0020$; one-sided unpaired Welch's $t$ test (J) and Dunn's multiple comparisons test following a Kruskal–Walls test (K), $n = 5$–7 cells (J, K). Data are represented by the mean ± SEM. Double arrow, full fusion; single arrows, cortical long-range directional motilities.

The ER is the location for the folding of de novo synthesized proteins (Braakman and Hebert, 2013). Being translated in the rough ER, membrane proteins undergo the folding process assisted by the ER-resident luminal chaperones calnexin-1 and derlin-1; and by the heat-shock proteins (HSPs) that largely behave as cytoplasmic chaperones (Freeman and Morimoto, 1996; Yahara et al, 1998). In particular, some ER client proteins need a process termed "chaperone exchange," in which the client-binding Hsp70 chaperone is replaced by the Hsp90 chaperone with the help of the STIP1 (Hop) protein (Donnelly et al, 2013; Moran Luengo et al, 2018), as well as the ER-resident chaperones calnexin-1 and derlin-1 (Ramos et al, 2007; Volpi et al, 2016). Because this process plays an essential role in ER quality control (ERQC), the proteins that cannot undergo chaperone exchange are degraded by ER-dependent protein degradation (ERAD) involving the proteasomal pathway (Hoseki et al, 2010). Thus, the chaperone exchange is a critical checkpoint of protein folding for a subset of ER clients. However, its relationship to specific ER morphological compartment and kinesin molecular motor is still unclear.

In this study, we have investigated the molecular roles of KIF5B in beta cells in detail using the beta-cell-specific KIF5B conditional knockout (cKO) mice and KIF5B-knockdown MIN6 insulinoma cells. We propose that KIF5B is a crucial factor for Hsp70-to-Hsp90 chaperone exchange for $Ca_V1.2$ acting on ER sheets and thus essentially maintains $Ca_V1.2$ expression level that is indispensable for $Ca^{2+}$-mediated GSIS for glucose homeostasis. This process involves $Ca^{2+}$-mediated SNARE complex formation as well as $Ca^{2+}$-dependent cortical F-actin remodeling, which allows insulin granules for short-range directional movements and full-fusion exocytosis. We also showed that $K_{ATP}$ subunits can be folded irrespective of KIF5B and Hsp90, suggesting a selective role of the KIF5B–Hsp machinery for a subset of membrane protein synthesis. As we present that dual overexpression of KIF5B and Hsc70 in MIN6 cells results in upregulation of $Ca_V1.2$ protein levels, KIF5B–Hsp machinery may serve as an important regulatory system that enhances the GSIS performance of beta cells, upon changing the systemic nutrition balance of the body.

# Results

## Beta-cell-specific KIF5B deletion in mice leads to glucose intolerance

We generated pancreatic beta-cell-specific conditional knockout mice for the *Kif5b* gene (*Kif5b^flox/flox^ Rip2-Cre*; cKO) using homologous recombination in mouse ES cells (Fig. EV1). The significant deficiency of KIF5B in insulin-expressing islet beta cells was verified by immunohistochemistry (Fig. 1A). The cKO islets revealed significant hyperplasia (Fig. 1B,C), which may partly compensate the defect in GSIS.

In the intraperitoneal glucose tolerance test (IPGTT), the cKO mice exhibited significant glucose intolerance compared with the floxed control (*Kif5b^flox/flox^*; CT; Fig. 1D). Because the isolated islets did not show significant alteration in the degree of elevating the ATP/ADP ratio after glucose stimulation (Fig. 1E), the metabolic state of glucose was supposed to be largely intact in cKO islets.

To investigate temporal transition in GSIS, we performed perifusion experiments of isolated islets (Fig. 1F,G). Interestingly, the basal level of insulin secretion in cKO islets was significantly elevated twice that in CT islets. In contrast, the peak size of the first-phase GSIS significantly decreased half of the control. The second-phase GSIS was further significantly abolished. These data were largely consistent with the results of previous genetic studies (Cui et al, 2011; Donelan et al, 2002; Meng et al, 1997; Varadi et al, 2002; Varadi et al, 2003), as well as the results of our preliminary study using aged *Kif5b^+/−^* mice and their islets, so the consequence of genetic artifacts could be largely neglected.

## KIF5B is essential for full-fusion insulin granule exocytosis

To analyze the relevance of KIF5B in quantum events of exocytosis, we transduced the primary culture of beta cells with synapto.pHluorin (Miesenbock et al, 1998; Tsuboi and Rutter, 2003), and observed them by TIRF microscopy between 30 and 60 min after glucose stimulation in the period of second-phase GSIS (Fig. 2A; Movie EV1). The membrane fusion events were classified into kiss-and-run fusion events (<1.2 s) and full-fusion events (≥1.2 s) according to previous insights (Ohara-Imaizumi et al, 2007; Takahashi et al, 2004; Takahashi et al, 2002) that is consistent with our observation with histogram analysis (Fig. 2B–D). Interestingly, cKO beta cells tended to exhibit more kiss-and-run fusion events but significantly fewer full-fusion events (Fig. 2E).

By examining the *t*-projection views, we found that insulin granules in cKO beta cells significantly exhibited long-range movements passing through the fusion site, which reduced the duration of fusion events (Fig. 2F,G).

Because F-actin remodeling is known to stabilize full-fusion exocytosis (Eitzen, 2003), we challenged cKO cells with a low dose (10 μg/mL) of the F-actin depolymerizing drug, cytochalasin B

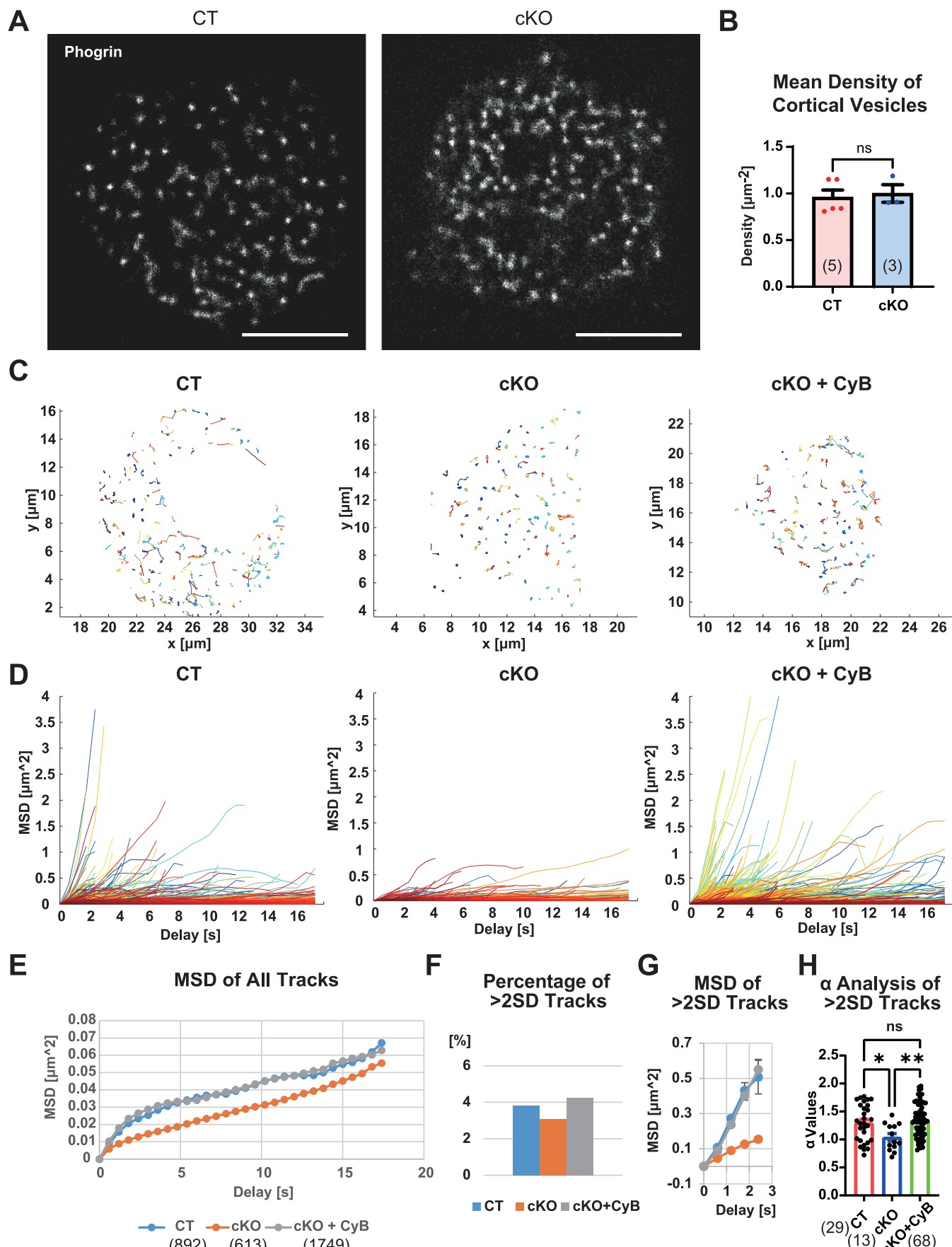

Figure 3. KIF5B facilitates short directional motility of insulin granules.

(A, B) Images of cortical insulin granules of control (CT) and cKO primary beta cells, transfected with a *phogrin-Dronpa-Green1* expression vector, starved, and stimulated by 20 mM glucose for 30 min (A), accompanied by quantification of cortical insulin granules (B). Bar, 2 μm. ns, $P = 0.7508$, $n = 3$–5 cells, two-sided unpaired Welch's $t$ test. Data are represented by the mean ± SEM. (C–H) Quantification of insulin granule motility tagged by phogrin-EGFP with a 5LIVE-Duo confocal microscope, in primary beta cells of the indicated genotypes and treatments for 100 s, at 30 min after glucose stimulation; represented by particle tracks (C), MSD trajectories (D), MSD curves of all tracks (E), percentage of >2 SD tracks (F), MSD curves of >2 SD tracks (G), and α analysis of >2 SD tracks for directional movements (H); corresponding to Movie EV2. Color coding in (C, D), the time sequence. ns, $P = 0.7468$; *$P = 0.0308$; **$P = 0.0030$; one-way ANOVA with Tukey's multiple comparison test, $n = 13$–68 tracks (G, H). Note that cKO beta-cell granules were significantly less motile than CT granules, which were significantly reversed by the CyB treatment. Data are represented by the mean ± SEM.

(CyB; Fig. 2H–K). Consequently, cKO cells treated by CyB for more than 10 min showed significantly and progressively fewer peripheral motility and kiss-and-run fusion events, and a greater occurrence of full-fusion events. Accordingly, KIF5B's role in suppressing long-range directional transport of insulin granules for full fusion may be related with cortical actin remodeling.

## KIF5B allows directional short-range movements of insulin granules

To further investigate the dynamics of insulin granules, we labeled the granules of primary beta cells with phogrin-Dronpa-Green1 and subjected them to TIRF/PALM microscopy (Fig. 3A). The cortical density of insulin granules was largely unaltered by KIF5B deficiency (Fig. 3B).

Then, we labeled them with phogrin-EGFP and performed high-speed confocal imaging using an LSM 5LIVE-Duo microscope followed by particle tracking (Movie EV2). In this way, we detected loss of short-range directional movements of insulin granules in cKO beta cells (Fig. 3C), which again tended to be restored by CyB treatment.

These data were further subjected to mean square displacement (MSD) analyses (Fig. 3D–H), according to previously described methods (Tarantino et al, 2014). In the summary of individual trajectories, the slopes of CT and cKO + CyB ones were apparently steeper than cKO trajectories (Fig. 3D). The curves of MSD of all tracks indicated that the ones for CT and cKO + CyB were similar but the one for cKO was lower than those, mainly because of the existence of a convex upward component in 0–3 s of delay (Fig. 3E).

To further characterize this component, we depicted >2 SD trajectories (Fig. 3F) and again summarized them, which revealed more significant changes in the slopes (Fig. 3G). Finally, we calculated the α values of each >2 SD trajectory (Fig. 3H), where α = 1 represents freely diffusive movements and the higher values represent the existence of active transport. Very interestingly, the CT and cKO + CyB groups had similar average values of 1.3 but the cKO group had an average value of 1.0. Thus, KIF5B was suggested to enable active short-range actomyosin-based transport of insulin granules that may serve for the perpendicular access of granules to plasma membrane as previously observed in other systems (Ueno et al, 2011). Because CyB treatment could nicely compensate the short-range directional motilities of insulin granules, KIF5B's role in enhancing short-range directional transport of insulin granules for full fusion was also considered to be related with cortical actin remodeling.

## KIF5B deficiency impairs F-actin remodeling during the second phase

Then we investigated whether glucose-stimulated F-actin remodeling is really disorganized in KIF5B-deficient cKO cells. In starved cells, the intensity of cortical F-actin was largely unaltered (Non-Stim; Fig. 4A). However, glucose stimulation for 60 min revealed a difference (Glucose; Fig. 4A). In CT cells, cortical F-actin was significantly remodeled to become thinner during this period, as previously reported (Nevins and Thurmond, 2003; Olofsson et al, 2009). Because F-actin tended to accumulate into foci on the cell surface, mutual sliding and depolymerization may be involved in this remodeling process. In contrast, in the cKO cells, cortical F-actin was not remodeled but was even thickened by glucose stimulation. The relative median intensity of cortical F-actin of CT cells became half in this period, but that of cKO cells was almost unchanged (Fig. 4B). Because forced $[Ca^{2+}]_i$ elevation by ionomycin could restore the F-actin remodeling in both genotypes, the role of KIF5B in F-actin remodeling was considered to be dependent on or in parallel to $Ca^{2+}$ transients.

This phenotype was reproduced in a time-lapse manner using primary beta cells of *Lifeact-mCherry* transgene-carrying mice (Fig. EV2A; Movie EV3). We conducted *Kif5b* gene silencing as described later. Scrambled-control miRNA-transduced (SC) cells exhibited gradual remodeling of peripheral actin bundles by 5–25 min after glucose stimulation. Interestingly, large pseudopods were dynamically produced in KIF5B-knockdown (KD) cells, but peripheral F-actin remained intact upon glucose stimulation. As a control, KIF5B-EYFP overexpression in cKO cells also significantly restored cortical F-actin remodeling (Fig. EV2B–D), suggesting that these phenotypes were truly from KIF5B deficiency. The application of CyB during glucose stimulation completely disrupted and fragmented the cortical F-actin of cKO cells (Fig. 4C,D).

To study upstream signaling pathway leading to F-actin remodeling (Veluthakal and Thurmond, 2021), we measured the time course of the level of phosphorylated and activated SFK (pSFK) using immunofluorescence, and those of the Cdc42 and Rac1 GTPase activities using FRET biosensors (Yoshizaki et al, 2003). In CT primary beta cells, a transient SFK activation at 1 min, Cdc42 activation at 1–3 min, and a sustained Rac1 activation later than 5 min were sequentially observed (Fig. 4E–G). However, in cKO primary beta cells, these activities were significantly abolished. These data collectively indicated an essential role of KIF5B in F-actin remodeling through its relevance in SFK–Rho-family GTPase signal transduction cascade.

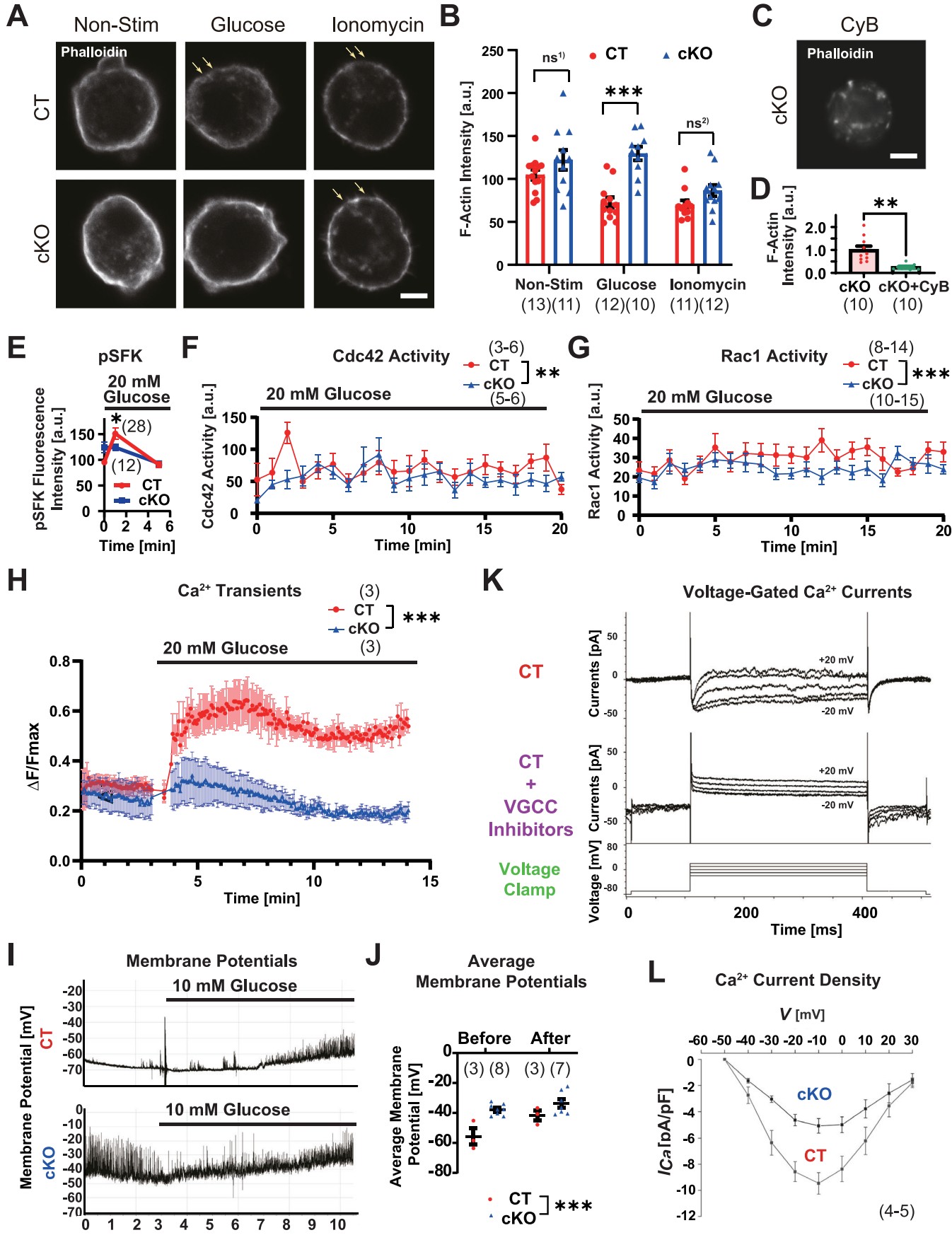

Figure 4. KIF5B is essential for Ca$^{2+}$ transients and actin remodeling of beta cells.

(A–D) Actin remodeling assay using fluorescent phalloidin-labeled CT and cKO primary beta cells with the indicated treatments (A, C), accompanied by quantification (B, D). Scale bars, 5 µm. ns[1), P = 0.2014; ns[2), P = 0.0616; ***P = 0.000021; multiple unpaired Welch's t test, n = 10–13 cells. Quantified at the positions of largest diameter. Arrows, the sites of cortical F-actin remodeling. Data are represented by the mean ± SEM. Corresponding to Fig. EV2A–D and Movie EV3. (E–G) Glucose-stimulated activation of SFK (E), Cdc42 (F), and Rac1 (G) in primary beta cells of the indicated genotypes after 20 mM glucose stimulation from time 0, measured by immunofluorescence microscopy (E) and the respective FRET biosensors (F, G). *P = 0.0475; two-sided unpaired Welch's t test, n = 28–42 cells (E); **P = 0.0047, ***P = 0.0001, two-way ANOVA, n = 3–14 cells (F, G). Data are represented by the mean ± SEM. (H) 20 mM glucose-stimulated calcium transients of the primary beta cells labeled with Fluo4-AM. ***P = 3.7 × 10$^{-222}$, two-way ANOVA on periods after the stimulation, n = 3 cells. Data are represented by the mean ± SEM. (I, J) Membrane potentials of resting and 10 mM glucose-stimulated primary beta cells according to whole-cell patch-clamp recordings (I); and quantification of the mean membrane potentials before and after the glucose stimulation (J). ***P = 0.0007, two-way ANOVA, n = 3–8 cells. Data are represented by the mean ± SEM. (K) Traces of voltage-gated Ca$^{2+}$ currents in patch-clamp recording of mouse primary beta cells of the indicated genotypes in 10 mM glucose, with or without the VGCC inhibitors cocktail, containing 20 µM nifedipine, 1 µM SNX482, and 0.3 mM ascorbate, at the range of −20 to +20 mV. Note that the inhibitor treatment significantly abolished the voltage-gated inward currents. (L) Ca$^{2+}$ inward current density curves of the primary beta cells of the indicated genotypes measured by whole-cell patch clamp. n = 4–5 cells. Data are represented by the mean ± SEM.

## KIF5B supports electrophysiological and Ca$^{2+}$ activities in beta cells

Interestingly, glucose-induced Ca$^{2+}$ transients were significantly impaired in cKO beta cells (Fig. 4H). In CT primary beta cells, glucose application significantly elevated the $[Ca^{2+}]_i$ immediately after application to reach the maximum in 3 min. However, in cKO cells, the level of glucose-stimulated $[Ca^{2+}]_i$ elevation was less than one-fifth that in CT cells.

Then we conducted patch-clamp recording of voltage-gated Ca$^{2+}$ currents in primary beta cells. First, we conducted a whole-cell patch clamp of primary beta cells with glucose elevation from 0 to 10 mM in the external solution (Fig. 4I,J). The resting membrane potential of CT cells was properly maintained at −60 mV. However, those of cKO beta cells were significantly depolarized to −40 mV. Consequently, spontaneous action potentials tended to increase in the resting state in cKO beta cells. This membrane excitation tendency can partly explain the cause of beta-cell and islet hyperplasia (Li et al, 2014).

Then, voltage-gated Ca$^{2+}$ currents were recorded in 10 mM glucose with stepwise changes in membrane potentials (Fig. 4K). The observed voltage-gated inward currents were suppressed by a VGCC inhibitor cocktail. Under this condition, the Ca$^{2+}$ current densities in cKO beta cells were significantly smaller than those of CT beta cells (Fig. 4L). Accordingly, cKO beta cells exhibited continuous membrane excitation, while the voltage-gated inward Ca$^{2+}$ current densities in 10 mM glucose were significantly reduced, leading to the abolishment in the glucose-stimulated Ca$^{2+}$ transients.

## KIF5B deficiency leads to downregulation of membrane proteins, including Ca$_V$1.2

To investigate the molecular mechanism between KIF5B and Ca$^{2+}$ transients, we conducted gene silencing by expressing *Kif5b*-antisense knockdown (KD) or scrambled sequence control (SC) miRNAs using mammalian expression vectors in MIN6 insulinoma cells (Fig. 5A,B). As a result, the levels of Ca$_V$1.2, Ca$_V$2.3, PIP5Kα, BK$_{Ca}$, syntaxin-1, Na/K ATPase, and Hsp70 proteins were significantly reduced. In contrast, the K$_{ATP}$ channel subunits K$_{ir}$6.2 and SUR, as well as α-tubulin and calnexin-1 were largely unaltered. Because the changes in transcription levels of Ca$_V$1.2, Ca$_V$2.3, and K$_{ir}$6.2 were subtle (Fig. 5C), the reduction appeared to occur primarily at the post-transcriptional level.

We then conducted immunofluorescence microscopy in primary beta cells. Ca$_V$1.2 and Ca$_V$2.3 were significantly downregulated from the surface of cKO cells in TIRF microscopy (Fig. 5D,E). However, K$_{ir}$6.2 remained intact or even slightly increased. To investigate whether a Ca$_V$1.2 activation can compensate the GSIS failure due to KIF5B deficiency, we incubated the cKO islets with high glucose and/or the L-type VGCC agonist, Bay-K 8644 (Satin et al, 1995). Although high glucose could scarcely stimulate the GSIS of cKO islets, application of 1 µM Bay-K 8644 along with high glucose stimulation yielded 8-fold increase of GSIS (Fig. 5F). These data suggested that pharmacological activation of the reduced level of Ca$_V$1.2 in cKO islets could still restore the GSIS. Thus, we considered that cKO mouse islets may preserve the capacity for GSIS if proper Ca$^{2+}$ transients were formed, and that the reduced expression of Ca$_V$1.2 in cKO islets could be a major cause of the GSIS failure. The negative effects of 'cellular fatigue' through constitutive depolarization (Khaldi et al, 2004; Pertusa et al, 2002; Roche et al, 1998) was thus unlikely especially in the case of this second-phase GSIS.

In addition, PIP5Kα and its enzymatic product PIP$_2$ were significantly downregulated in cKO beta cells (Fig. 5G,H). Because PIP$_2$ is a key regulator of K$_{ATP}$, VGCC, and focal adhesion kinase (FAK) activities (Lin et al, 2005; Suh et al, 2012; Zhou et al, 2015), this downregulation may partly explain the accessory phenotypes in KIF5B cKO mice. Those changes in the stimulation-secretion coupling of KIF5B-deficient beta cells were schematically represented in Fig. 5I.

## Ca$_V$1.2 undergoes Hsp70-to-Hsp90 chaperone exchange and is stabilized by KIF5B

Focusing on the behavior of the VGCCs, we compared protein stability using cycloheximide (CHX) treatment that inhibited whole protein synthesis (Fig. 6A–C). Ca$_V$1.2 and Ca$_V$2.3 protein signals in cKO cells degraded significantly faster than those in CT cells, whereas the degradation rate of α-tubulin was unchanged. To determine whether this downregulation of Ca$_V$1.2 was caused by aberrant post-Golgi antero-grade trafficking, we conducted a brefeldin-A (BFA) washout experiment (Fig. 6D,E; Movie EV4). The time needed for Cav1.2-EGFP to reach the plasma membrane after BFA washout in cKO cells was only slightly decreased. Thus, kinesin-1 is largely dispensable for post-Golgi trafficking of Ca$_V$1.2, which cannot be the major cause of the Ca$_V$1.2 downregulation in cKO cells.

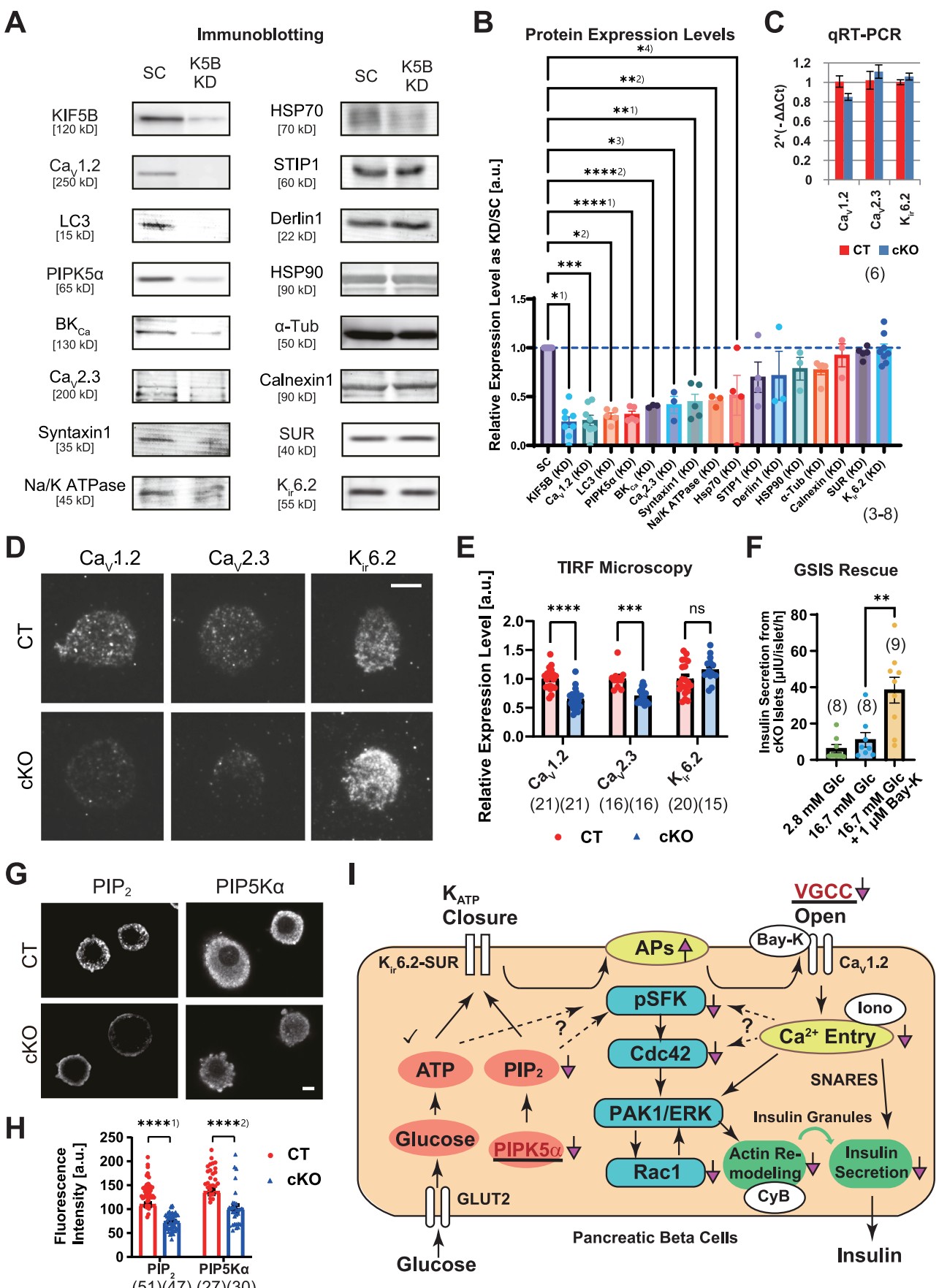

Figure 5. KIF5B is essential for Ca$_V$1.2 protein expression in beta cells.

(A, B) Immunoblotting of a *Kif5b* gene silencing system in MIN6 cells using scrambled-control (SC) and KIF5B-knockdown (KD) miRNA, for the indicated proteins (A); and its normalized quantification against the SC-transduced cells (B). [*1]$P = 0.0178$; [*2]$P = 0.0229$; [*3]$P = 0.0108$; [*4]$P = 0.0490$; [**1]$P = 0.0076$; [**2]$P = 0.0020$; [***]$P = 0.006$; [****1]$P = 0.0002$; [****2]$P = 9.59 \times 10^{-5}$; one-sided unpaired Welch's *t* test between KD and SC, $n = 3–8$ biological replicates. Data are represented by the mean ± SEM. (C) Quantitative RT–PCR results of the KD system among the indicated genes, $n = 6$. Data are represented by the mean ± SEM. (D, E) Immunofluorescence microscopy of primary beta cells of the indicated genotypes, against Ca$_V$1.2, Ca$_V$2.3, and K$_{ir}$6.2, using TIRF microscopy (D), and its quantification (E). Bar, 5 μm. ns, $P = 0.0597$; [***]$P = 0.0002$; [****]$P = 1.45 \times 10^{-7}$; two-sided unpaired Welch's *t* test, $n = 11–21$ cells. Data are represented by the mean ± SEM. (F) Rescue of impaired GSIS from cKO mouse islets by 1 μM Bay-K 8644. [**]$P = 0.0019$, one-way ANOVA. $n = 8–9$. Data are represented by the mean ± SEM. (G, H) Immunofluorescence microscopy of primary beta cells of the indicated genotypes, against PIP$_2$ and PIP5Kα using a confocal laser scanning microscope (CLSM, G), and quantification (H). Bars, 5 μm. [****1]$P = 1.154 \times 10^{-12}$; [****2]$p = 1.805 \times 10^{-4}$; one-sided unpaired Welch's *t* test, $n = 27–51$ cells. Data are represented by the mean ± SEM. (I) Schematic representation of possible changes in the stimulation-secretion coupling of GSIS in KIF5B cKO beta cells. Checkmark, normal expression. Red arrows, changes according to KIF5B deficiency. Ca$_V$1.2 and PIP5Kα protein downregulation (underlined) in KIF5B-deficient beta cells may primarily result in the abolishment of Ca$^{2+}$ transients and downregulation of PIP$_2$, respectively. White ovals, pharmacological reagents that directly stimulated the respective pathways: Bay-K Bay-K 8644, Iono ionomycin, CyB cytochalasin B.

We conducted TIRF/STORM microscopy of primary beta cells against KIF5B and Ca$_V$1.2 (Fig. 6F). Very interestingly, these two proteins partially colocalized to ER-like networks beneath the plasma membrane of wild-type cells. We also conducted TIRF/STORM microscopy for Ca$_V$1.2 and Ca$_V$2.3 in CT and cKO primary beta cells (Fig. 6G). These two VGCCs were also partially colocalized to ER-like network, and reduced by KIF5B deficiency. Thus, we sought to investigate the relevance of KIF5B in ER-mediated protein folding.

We focused on Hsp70-to-Hsp90 chaperone exchange in the middle way of ER protein folding (Fig. 6H), as kinesin-1 was previously identified to interact with Hsc70 in neuronal axons (Terada et al, 2010). We conducted a proximity ligation assay (PLA) to investigate the possible changes in chaperone-binding capacities of Ca$_V$1.2 protein in CT and cKO primary beta cells (Fig. 6I,J). Interestingly, KIF5B deficiency significantly affected the Ca$_V$1.2–Hsp90 interaction, rather than Ca$_V$1.2–Hsp70 interaction, suggesting its relevance in chaperone exchange of Ca$_V$1.2.

STIP1 is an essential co-chaperone for the chaperone exchange system (Bhattacharya and Picard, 2021; Bhattacharya et al, 2020; Harrison et al, 2024). We sought to investigate if STIP deficiency could affect the expression levels of the client protein candidates. Gene silencing of *Stip1* in MIN6 cells reproducibly resulted in decrease of Ca$_V$1.2 and BK$_{Ca}$ proteins as well as STIP1 protein itself (Fig. 6K), suggesting the involvement of chaperone exchange for the folding of at least these two proteins.

These data collectively suggested that KIF5B is required for chaperone exchange of a subset of ER clients, including Ca$_V$1.2.

## KIF5B expression facilitates chaperone-binding capacities of Ca$_V$1.2

To further characterize the cause of Ca$_V$1.2 protein turnover in CHX-treated cKO beta cells, we applied the lysosomal inhibitor leupeptin or the proteasomal inhibitor MG-132 into the culture medium. In the consequence, MG-132, but not leupeptin, significantly restored the Ca$_v$1.2 expression (Fig. 7A,B), suggesting the involvement of proteasomal system in its turnover, as reported in other systems (Altier et al, 2011).

We then compared the chaperone- and co-chaperone-binding capacities of Ca$_V$1.2 and K$_{ir}$6.2 by vesicle immunoprecipitation (IP) among the KIF5B-KD system (Fig. 7C,D), where the Ca$_V$1.2 level was partially rescued by MG-132 treatment. Interestingly, the association of Hsp70 with Ca$_V$1.2 tended to be unaltered. In contrast, the association of Ca$_V$1.2 with STIP1, Hsp90, and ER-resident chaperones

calnexin-1 and derlin-1 decreased significantly. On the other hand, the Hsp70-binding fraction of K$_{ir}$6.2 was unaltered, but K$_{ir}$6.2 barely bound to Ca$_V$1.2, calnexin-1, derlin-1, STIP1, or Hsp90. K$_{ir}$6.2 was still associated with KIF5B possibly in the context of endosome/lysosome trafficking (Tanaka et al, 1998).

Furthermore, we compared the level of interaction between STIP1 and Hsp90 among the KIF5B-KD system in the presence of MG-132 (Fig. 7E). According to vesicle IP, Hsp90 binding capacity of STIP1 was reproducibly decreased upon KIF5B deficiency. This suggested that KIF5B mediates STIP1–Hsp90 binding for maintaining the integrity of chaperone exchange machinery. In the consequence, we could define a co-chaperone property in KIF5B, dedicated to a chaperone exchange machinery for a subset of ER protein folding (Fig. 7F).

To investigate the possibility of the KIF5B–Hsp machinery to strengthen the beta-cell function, we conducted overexpression trials in MIN6 cells to test if Ca$_V$1.2 immunofluorescence was upregulated (Fig. 7G,H). Although overexpression of EYFP-KIF5B alone did not apparently affect the Ca$_V$1.2 expression, co-overexpression of EYFP-KIF5B and tagRFP-Hsc70 significantly enhanced it. This may suggest a synergistic role of KIF5B–Hsp machinery in the maturation of their clients.

This KIF5B- and Hsc70 overexpression was found to alter the chaperone-binding capacities of Ca$_V$1.2 by vesicle IP. The binding capacities of Ca$_V$1.2 to calnexin-1 and Hsp90 were predominantly increased, but that to derlin-1 was decreased (Fig. 7I). The increase in Hsp90-bound Ca$_V$1.2 nicely supported our hypothesis that KIF5B enhances the Hsp70-to-Hsp90 chaperone exchange. Calnexin-1 balances protein production rates through the calnexin/calreticulin cycle (Kozlov and Gehring, 2020), and its gene silencing significantly increased the Ca$_V$1.2 production rate in mouse neonatal cardiomyocytes (Bousette et al, 2014). Thus, overproduced Ca$_V$1.2 protein might be partially accumulated in this rate-limiting step before exiting the ER. As derlin-1 behaves like a protein degrading chaperone for ERAD (Altier et al, 2011), the decrease in the Ca$_V$1.2–derlin-1 complex was also reasonable. Accordingly, KIF5B–Hsp system appeared to enhance the Ca$_V$1.2 protein folding via the ER chaperones.

## KIF5B facilitates Hsp90-containing microdroplet dynamics on ER sheets

To visualize the KIF5B–Hsp machinery in living cells, we co-transduced primary cKO beta cells with Hsp90-tagRFP and KIF5B-

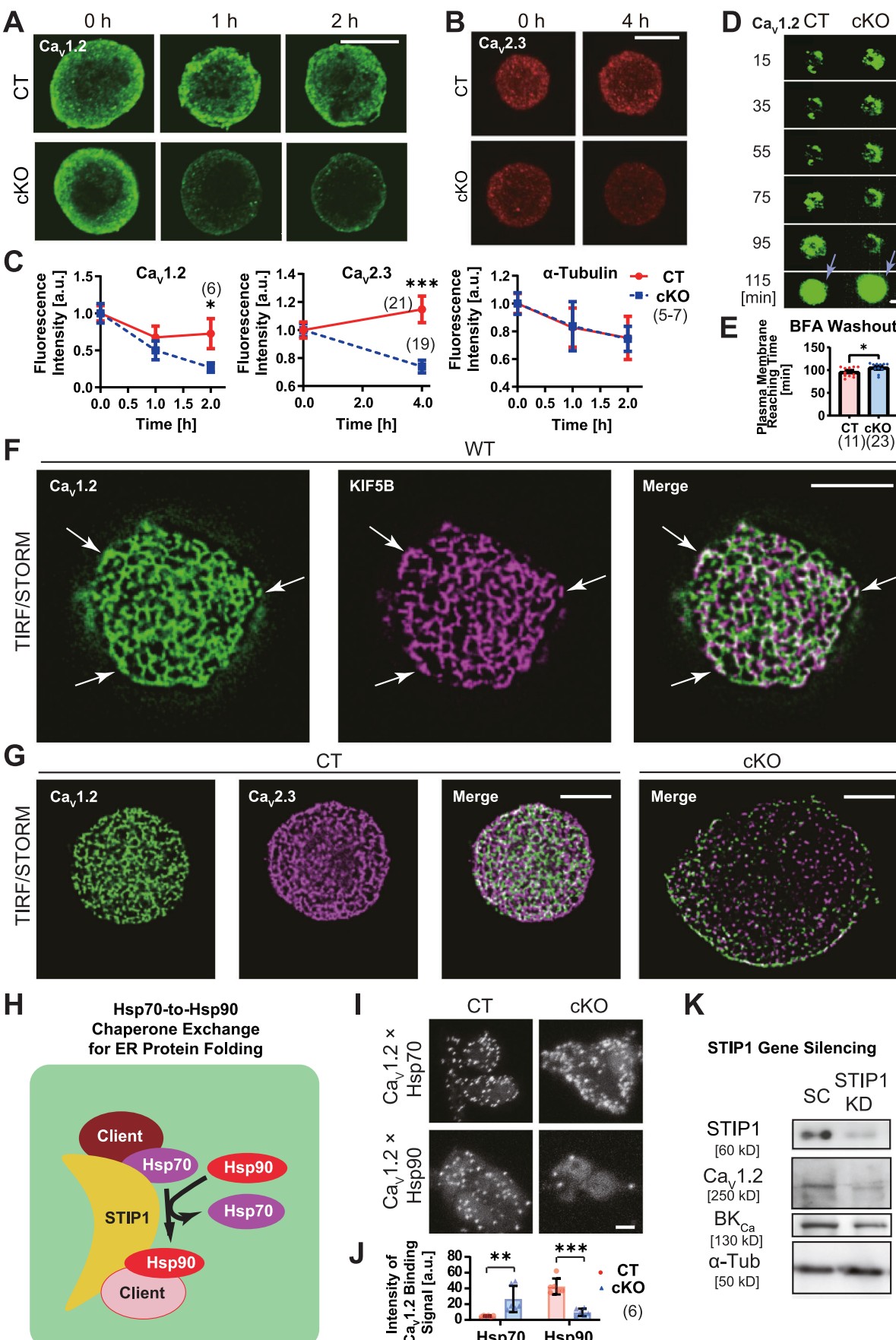

**Figure 6.  KIF5B facilitates chaperone exchange for Ca$_V$1.2 protein expression.**

(A–C) Degradation assay in immunofluorescence against Ca$_V$1.2 (A) and Ca$_V$2.3 (B) of primary beta cells of the indicated genotypes after the CHX treatment for the indicated periods; accompanied by their quantification along with that of α-tubulin (C). Bars, 5 μm. *$P = 0.02561$, ***$P = 0.00292$; one-sided unpaired Welch's $t$ test at the indicated time points; $n = 6$ cells (Ca$_V$1.2), 17–24 cells (Ca$_V$2.3), 5–7 cells (α-tubulin). Data are represented by the mean ± SEM. (D, E) Brefeldin-A (BFA) washout assay with an LSM 5LIVE-Duo microscope, assessing the speeds of post-Golgi trafficking of Ca$_V$1.2-EGFP proteins expressed in primary beta cells of the indicated genotypes (D), accompanied by its quantification (E). Time after BFA washout is indicated. Bar, 5 μm. *$P = 0.0207$, one-sided unpaired Welch's $t$ test, $n = 11$ cells. Arrows, the timing of plasma membrane fusion. Data are represented by the mean ± SEM. Corresponding to Movie EV4. (F) TIRF/STORM microscopy of a wild-type primary mouse beta-cell immunolabeled against Ca$_V$1.2 and KIF5B. Scale bar, 5 μm. Arrows, colocalizing spots. (G) TIRF/STORM microscopy of primary mouse beta cells of the indicated genotypes immunolabeled against Ca$_V$1.2 (green) and Ca$_V$2.3 (magenta). Scale bars, 5 μm. (H) Schematic representation of STIP1-dependent Hsp70-to-Hsp90 chaperone exchange machinery. (I, J) Proximity ligation assay in CT and cKO primary beta cells with $z$-projection views, indicating protein binding signals between Ca$_V$1.2 and the indicated Hsp proteins (I); accompanied by quantification (J). Scale bar, 5 μm. **$P = 0.0122$; ***$P = 8.42 \times 10^{-5}$; one-sided unpaired Welch's $t$ test, $n = 6$ cells. Data are represented by the mean ± SEM. (K) Immunoblotting of scramble control (SC) and STIP1-knockdown (KD) MIN6 cells against the indicated epitopes. Note that STIP1 deficiency induced downregulation of Ca$_V$1.2 and BK$_{Ca}$ proteins. Reproduced twice.

EYFP and conducted live cell imaging. As revealed by the stereoscopic presentation of $z$-stack images, tagRFP-Hsp90 and KIF5B-EYFP tended to co-accumulate on patch-like microdroplets especially in the cell bottom (Fig. EV2E). Time-lapse microscopy revealed that Hsp90 formed fine meshwork, along which the microdroplets containing Hsp90 and KIF5B underwent very dynamic movements (Fig. 8A; Movie EV5). Occasionally, KIF5B- and Hsp90-containing tubules were extended from one microdroplet toward another, suggesting their dynamic material-exchanging properties that may serve for liquid-liquid phase separation (LLPS) (Naz et al, 2024).

According to immunocytochemistry of KIF5B-rescued cKO beta cells, Hsp70, Ca$_V$1.2, and the focal adhesion marker paxillin tended to co-accumulate on the microdroplets with KIF5B and Hsp90 (Fig. 8B,C). However, K$_{ir}$6.2 was excluded from there, reflecting its independence on Hsp90. The colocalizing features between KIF5B and Hsp90 on microdroplets were further confirmed by intrinsic immunocytochemistry of wild-type primary beta cells (Fig. 8D).

To investigate the relationship between the ER and those microdroplets, we co-transduced primary beta cells with tagRFP-Hsp90 and the ER marker, mEmerald-Sec61β (Nixon-Abell et al, 2016) (Fig. 8E–G). Airyscan microscopy revealed that Hsp90-including microdroplets consisted of a bright core and rather amorphous halo. They were frequently colocalizing to finer parts of ER meshwork (Fig. 8F), which were termed "ER sheets" (Shibata et al, 2010). Time-lapse analysis further revealed that those ER sheets and Hsp90 dynamically migrated together (Fig. 8G; Movie EV6).

These data collectively suggested that KIF5B is essentially involved in Hsp70-to-Hsp90 ER chaperone exchange system that occurs in microdroplets associated with ER sheets, and this may facilitate the proper maturation of a subset of ER clients including Ca$_V$1.2 (Fig. 8H). This may be the subcellular basis of KIF5B-mediated control of Ca$_V$1.2 expression that may finely tune the GSIS for maintaining glucose homeostasis.

## Discussion

In this study, we found that KIF5B kinesin is essential for the first- and second-phase GSIS, mainly through a protein folding process in the ER sheets involving the KIF5B- and Hsp90-containing microdroplets (Fig. 8). These microdroplets may be transported and scaffolded by the KIF5B molecular motor, and may be essential for Hsp70-to-Hsp90 chaperone exchange as well as ER chaperone binding for proper folding of a subset of membrane proteins

including Ca$_V$1.2 (Figs. 6 and 7). Ca$_V$1.2 enterprises glucose-stimulated Ca$^{2+}$ transients that controls first- and second-phase GSIS, partly through Ca$^{2+}$-dependent SNARE complex formation (Trexler and Taraska, 2017) and partly through cortical F-actin remodeling that modulates insulin granule dynamics (Kalwat and Thurmond, 2013). This finding uncovers an outstanding molecular mechanism in the stimulation-secretion coupling of GSIS that acts against diabetes, as well as the relevance of kinesin-1 molecular motor in protein folding in the ER. Because our beta-cell specific KIF5B cKO mice largely phenocopied the Ca$_V$1.2 cKO mice (Schulla et al, 2003), rather than Ca$_V$2.3 KO mice (Jing et al, 2005), and that the Ca$_V$1.2 agonist Bay-K 8644 nicely restored the GSIS in KIF5B cKO islets (Fig. 5F), the Ca$_V$1.2 channel was considered as the most physiologically relevant client of the KIF5B–Hsp machinery in pancreatic beta cells.

The relationship between ER and kinesin-1 has long been suggested by many groups (Gupta et al, 2008; Raiborg et al, 2015; Woźniak et al, 2009), but the KIF5B's specific role in protein folding has not been determined yet. In our previous knockout mouse study (Tanaka et al, 1998), we reported that the overall ER distribution and BFA-mediated Golgi-to-ER trafficking were largely unaltered, but mitochondria and lysosomes revealed significant perinuclear clustering in KIF5B-deficient extraembryonic cells. On the other hand, Allan and colleagues showed that dominant-negative kinesin-1 inhibits centripetal ER tubule motility in VERO cells (Woźniak et al, 2009), suggesting that kinesin-1 still serves as one of the molecular motors that synergistically act on ER morphogenesis. As of associating factors, the Rab18 and kinectin-1 (KNT-1) interaction may facilitate the dynamics of ER-focal adhesion (FA) contact as well as kinesin-1-dependent ER extension (Guadagno et al, 2020; Zheng et al, 2022), and KNT-1 is reported to be enriched in ER sheets (Shibata et al, 2010). Recently, it has also been reported that KIF5B transports vimentin (Robert et al, 2019), and that vimentin controls ER remodeling during ER stress (Cremer et al, 2023). These findings support our current idea that KIF5B regulates the structure and function of the ER in beta cells. As we have shown in an overexpression study (Fig. 7G–I), these KIF5B- and Hsp-dependent processes enhance the production of Ca$_V$1.2 protein in pancreatic beta cells, to serve for enhancing the stimulation-secretion coupling for glucose homeostasis.

Our data suggested that KIF5B deficiency augmented the basal insulin secretion as well as constitutive depolarization (Figs. 1F and 4I). This could be mostly explained by the downregulation of Na/K ATPase and BK$_{Ca}$ in KIF5B-KD cells (Fig. 5A,B). Alternatively,

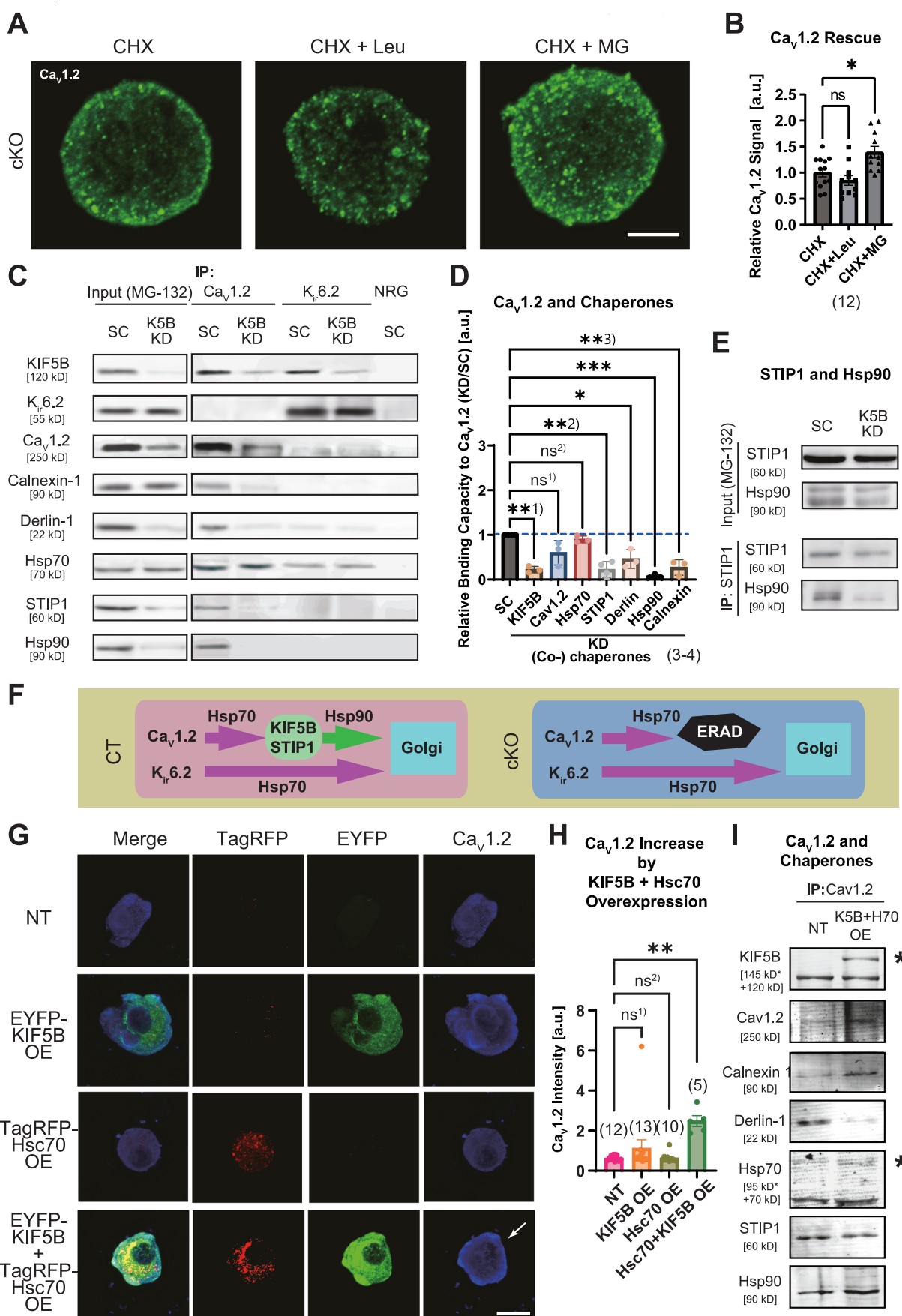

**Figure 7.  KIF5B–Hsp machinery that facilitates Ca$_V$1.2 expression.**

(A, B) Rescue of Ca$_V$1.2 degradation in cKO primary beta cells in the presence of CHX by leupeptin (Leu) or MG-132 (MG) for 4 h. Scale bar, 5 μm. ns, $P = 0.4331$; *$P = 0.0112$; one-way ANOVA with Dunnett's multiple comparison test, $n = 12$ cells. Data are represented by the mean ± SEM. (C, D) Vesicle IP of MG-132-treated MIN6 cell lysates transduced with scrambled-control (SC) and KIF5B-knockdown (KD) miRNAs, precipitated using Ca$_V$1.2 or K$_{ir}$6.2 antibodies or normal rabbit IgG (NRG) and immunoblotted for the indicated proteins (C), accompanied by quantification of Ca$_V$1.2-coprecipitated fractions (D). ns[1], $P = 0.0634$; ns[2], $P = 0.2434$; *$P = 0.0235$; **[1]$P = 0.0014$; **[2]$P = 0.0016$; **$P = 0.0092$; ***$P = 3.54 \times 10^{-4}$; one-sided unpaired Welch's $t$ test between KD and SC, $n = 3$–4 biological replicates. Data are represented by the mean ± SEM. Note that the Ca$_V$1.2-binding capacities of derlin-1, calnexin-1, and Hsp90 chaperones and that of the adaptor protein STIP1 in KD cell lysates were significantly lower than those in SC cell lysates. (E) Vesicle IP of the MG-132-treated MIN6 cell lysates among the KIF5B-KD system against STIP1. Note that the Hsp90 level in the STIP1 immunoprecipitants (IP) was greatly decreased by KIF5B deficiency. Repeated twice. (F) Schematic representation of the working hypothesis on differential KIF5B- and heat-shock-protein (Hsp)-dependencies of opposing ER clients Ca$_V$1.2 and K$_{ir}$6.2 in control (CT) and KIF5B conditional knockout (cKO) mouse beta cells. In cKO cells, Ca$_V$1.2 fails in chaperone exchange to undergo ERAD-mediated degradation, but K$_{ir}$6.2 is intact because it is independent on the KIF5B–Hsp machinery. (G, H) Ca$_V$1.2 immunocytochemistry of MIN6 cells that had been transduced with EYFP-KIF5B and/or TagRFP-Hsc70 or without them (NT; G); accompanied by their quantification (H). Scale bar, 5 μm. ns[1], $P = 0.4266$; ns[2], $P > 0.9999$; **$P = 0.0017$; one-way ANOVA with Dunnett's multiple comparison between KD and SC, $n = 5$–13 biological replicates. Data are represented by the mean ± SEM. Arrow in (G), enhanced Ca$_V$1.2 expression according to dual overexpression. (I) Vesicle IP of non-transduced (NT) and KIF5B- and Hsc70-overexpressing (K5 + H70 OE) MIN6 cell lysates against Ca$_V$1.2. Asterisks, tagged protein bands. The tagRFP-Hsc70 band was overlapped with a band of possibly ubiquitinated form. Reproduced twice.

Ca$_V$1.2 knockout mice exhibited a similar elevation of basal insulin secretion with significant GSIS decrease (Schulla et al, 2003). Because Ca$_V$1.2 augments BK$_{Ca}$ activity through a direct interaction (Plante et al, 2021), the Ca$_V$1.2 decrease itself may also account for the baseline changes in the KIF5B cKO cells.

Regarding the second-phase insulin secretion, our data have revealed that KIF5B is essential for a glucose-stimulated sequence of SFK and Rho-family GTPase activation (Fig. 4E–G), actin remodeling (Fig. 4A,B), and changes in insulin granule kinetics and full-fusion exocytosis probability (Figs. 2 and 3), which collectively maintain the second-phase GSIS. Although the pSFK–Cdc42–PAK1–Rac1 signaling pathway can partly crosstalk with Ca$_V$1.2-mediated Ca$^{2+}$ transients (Veluthakal and Thurmond, 2021), the glucose-stimulated activation mechanism of SFK within 1 min is still elusive as far as we know. The significant decrease of PIP$_2$ in KIF5B-deficient beta cells (Fig. 5G,H) may affect PIP$_2$-dependent FAK activation (Zhou et al, 2015) as a prerequisite to FAK–SFK coactivation (Huveneers and Danen, 2009). We have previously identified that KIF26A and KIF21B kinesins regulate the FAK–SFK interaction and Rac1 nucleotide cycling, respectively, in neurons (Morikawa et al, 2018; Wang et al, 2018). If our prediction was correct, gene manipulation of those related kinesins, as well as that of KIF5B (Fig. 7G,H), can enhance GSIS, which will be subjected to future research.

In KIF5B cKO beta cells, a small first-phase peak of GSIS was still remaining (Fig. 1F). Because Ca$^{2+}$ transients and pSFK elevation were almost completely abolished (Fig. 4E,H), this peak should be driven by other second messenger systems possibly involving protein kinases A and C (Komatsu et al, 2013) and/or Na$^+$ channels (Shigeto et al, 2006). KIF5B-deficient beta cells will thus serve as an ideal system for studying a Ca$^{2+}$- or SFK-independent component of first-phase GSIS, which will contribute to next-generation diabetes therapeutics (Sola et al, 2015; Wang et al, 2021).

In this study, we investigated the role of KIF5B in ERQC system for the Ca$_V$1.2 protein, which may be especially relevant for proper GSIS in beta cells. These cell biological findings will provide insights into the role of kinesin-1 in insulin secretion, from a completely unexpected view of cell biology. These findings will greatly stimulate future research and development on islet cell biology. Simultaneously, they will progress the basic cell biology of ER compartmentalization, which will be mostly dedicated to the geographical understanding of essential subcellular mechanisms.

# Methods

## Reagents and tools table

| Reagent/resource | Reference or source | Identifier or catalog number |
|---|---|---|
| **Experimental models** | | |
| KIF5B conditional knockout mice | This study | Figure EV1 |
| Rip2-Cre mice | Jackson Lab | B6.Cg-Tg(Ins2-cre)25Mgn/J |
| ROSA-STOP mice | Dr. P. Soriano | |
| C57BL/6J (*M. musculus*) | CLEA Japan | C57BL/6JJcl |
| Lifeact-mCherry mice | Dr. M. Sixt and R. Wedlich-Soldner | |
| 804G cells | Dr. J. Jones | |
| MIN6 cells | Dr. J. Miyazaki | |
| Ins1 cells | Dr. C.B. Wollheim | |
| **Recombinant DNA** | | |
| Kif5b-knockdown vectors | This study | Methods |
| Stip1-knockdown vector | This study | Methods |
| Scrambled-control (sc) knockdown vector | This study | Methods |
| *TagRFP-Hsc70* expression vector | This study | Methods |
| *TagRFP-Hsp90* expression vector | This study | Methods |
| *Phogrin-Dronpa-Green1* expression vector | This study | Methods |
| *Phogrin-EGFP* expression vector | This study | Methods |
| *mEmerald-Sec61β* expression vector | Dr. J. Lippincott-Schwartz | Addgene Cat # 90992 |
| *Ca$_V$1.2-EGFP* expression vector | Dr. L.F. Santana | |
| *Synapto.pHluorin* expression vector | Dr. G. Miesenboeck | |
| *pRaichu 1011x* expression vector | Dr. M. Matsuda | |
| *pRaichu 1054x* expression vector | Dr. M. Matsuda | |
| *Kif5b-EYFP, EYFP-Kif5b, ECFP-Kif5b* expression vectors | This study | Materials and Methods |
| **Antibodies** | | |
| Rabbit anti-Ca$_V$1.2 | Santa Cruz Biotechnology | Cat # sc-16229-R |
| Rabbit anti-K$_{ir}$6.2 | Santa Cruz Biotechnology | Cat # sc-20809 |

| Reagent/resource | Reference or source | Identifier or catalog number |
|---|---|---|
| Goat anti-PIP5Kα | Santa Cruz Biotechnology | Cat # sc-11775 |
| Goat anti-SUR1 | Santa Cruz Biotechnology | Cat # sc-11226 |
| Mouse anti-PIP$_2$ | Echelon | Cat # Z-A045 |
| Rabbit anti-GFP | MBL | Cat # 598 |
| Mouse anti-LC3 | Cosmo Bio | Cat # CTB-LC3-2-IC |
| Rabbit anti-BK$_{Ca}$ | Alomone Labs | Cat # APC-151 |
| Rabbit anti-Ca$_V$2.3 | Alomone Labs | Cat # ACC-006 |
| Mouse anti-syntaxin-1 | Millipore | Cat # MAB336 |
| Mouse anti-Na/K ATPase beta 2 | BD Transduction Labs | Cat # 610914 |
| Mouse anti-paxillin | BD Transduction Labs | Cat # 610051 |
| Mouse anti-Hsc70/Hsp70 | Enzo | Cat # ADI-SPA-822-D |
| Rat anti-Hsp90 | Enzo | Cat # ADI-SPA-835-D |
| Rabbit anti-STIP1 | Proteintech | Cat # 15218-1-AP |
| Rabbit anti-calnexin-1 | Proteintech | Cat # 10427-2-AP |
| Mouse anti-derlin-1 | Sigma Aldrich | Cat # SAB4200148 |
| Mouse anti-α-tubulin | Sigma Aldrich | Cat # CP06 |
| Mouse anti-insulin | Sigma Aldrich | Cat # I-2018 |
| Rabbit anti-phospho Src | Epitomics/ Abcam | Cat # ab32078 |
| Rabbit anti-KIF5B | This laboratory (Tanaka et al, 1998) | |
| Horseradish-peroxidase-conjugated protein A | Cytiva | Cat # NA9120V |
| Horseradish-peroxidase-conjugated sheep anti-mouse IgG | Cytiva | Cat # NA931V |
| Alkaline-phosphatase-conjugated goat anti-rabbit IgG, goat anti-mouse IgG, goat anti-rat IgG | Cappel | Cat # 59298, 59296, 59291 |
| Alkaline-phosphatase-conjugated rabbit anti-goat IgG | Zymed | Cat # 61-1622 |
| Alexa-conjugated goat and donkey anti-rabbit IgG, goat and donkey anti-mouse IgG, donkey anti-goat IgG | Thermo Fisher | |
| **Oligonucleotides and other sequence-based reagents** | | |
| PCR primers | This study | Methods |
| **Chemicals, enzymes, and other reagents** | | |
| Bay-K 8644 | Selleck | Cat # S7924 |
| Alexa-conjugated phalloidin | Thermo Fisher | |
| Cytochalasin B | Sigma Aldrich | Cat # C6762 |
| Ionomycin | Sigma Aldrich | Cat # 19657 |
| Nifedipine | Sigma Aldrich | Cat # N7634 |
| SNX482 | Alomone Labs | Cat # RTS-500 |
| L-ascorbate | Sigma Aldrich | Cat # A4034 |
| Cycloheximide | Fuji-Film WAKO | Cat # 037-20991 |
| Leupeptin | Fuji-Film WAKO | Cat # 336-40413 |
| MG-132 | Enzo | Cat # P1102 |
| CanGetSignal Immunostain Solution A | TOYOBO | Cat # NKB-501 |
| CanGetSignal Immunoreaction enhancer Solution 1 and 2 | TOYOBO | Cat # NKB-101 |
| Paraformaldehyde | Fuji-Film WAKO | Cat # 162-16065 |

| Reagent/resource | Reference or source | Identifier or catalog number |
|---|---|---|
| Glutaraldehyde | Nakalai Tesque | Cat # 170-03 |
| Triton X-100 | ICN | Cat # 2923E |
| Brefeldin A | Sigma Aldrich | Cat # B6542 |
| Fluo4-AM | Dojindo | Cat # F311 |
| Protein A Sepharose beads fast flow | Cytiva | Cat # 17-5132-01 |
| µMACS Protein A microbeads | Miltenyi Biotech | Cat # 120-000-396 |
| SYBR Primix Ex Taq (Tli RNaseH Plus) | TaKaRa | Cat # RR420 |
| **Software** | | |
| ZEN | ZEISS | |
| ImageJ ver. 1.54i | | https://imagej.net/ij/ |
| Cell Outliner plugin | | https://imagej.net/ij/plugins/cell-outliner.html |
| Fiji | | https://imagej.net/software/fiji/ |
| MicroManager ver 1.4 | | https://micro-manager.org/ |
| Imaris X64 ver. 7.7.1 | Oxford Instruments | |
| Prism ver. 10 | GraphPad | |
| Axon pCLAMP ver. 10 | Molecular Devices | |
| MATLAB | MathWorks | |
| MSD Analyzer plugin | | https://tinevez.github.io/msdanalyzer/ |
| **Other** | | |
| LBIS Mouse Insulin ELISA Kit | WAKO-Shibayagi | |
| ApoSENSOR ADP/ATP Ratio Assay Kit | BioVision | Cat #K255-200 |
| ViraPower Adenoviral Expression System | Thermo Fisher | Cat # K493000 |
| Block-iT PolII miR RNAi Expression vector kit | Thermo Fisher | Cat # K493600 |
| Lipofectamine-LTX transfection reagent | Thermo Fisher | Cat # A12621 |
| Total RNA isolation mini kit | Agilent | Cat # 5185-6000 |
| 1$^{st}$ strand synthesis kit | Origine | Cat # NP100041 |
| Amersham ECL Prime Western Blotting Detection System | Cytiva | Cat # RPN2109 |
| DuoLink System | Sigma Aldrich | |

## Mouse models

For generating KIF5B cKO mice, homologous recombination in the ES cells was conducted. A promoter-trap 3-*loxP* type targeting vector floxing the 74 bp p-loop exon (Fig. EV1A) was transfected into the J1 line of mouse embryonic stem (ES) cells by electroporation, as previously described (Tanaka et al, 1998). Homologous recombinants were screened by genomic Southern blotting with an efficiency of more than 70% (Fig. EV1B). Then, the *pCre-Pac* vector was transiently transfected into the homologous recombinants in the presence of puromycin, to generate *2-loxP* alleles as previously described (Teng et al, 2005). Each colony was genotyped with PCR using the following primer sets: neo, 5′-TGGG

CACAACAGACAATCGG-3′ and 5′-ACTTCGCCCAATAGCAGC CAG-3′; 5′-floxed region, 5′-CCAGATAACAGTTAAAAGCAGTG AAGG-3′ and 5′-CCATTATAGCCCTCAAGAACATCTATG-3′; 3′-floxed region (flox), 5′-CCCACACGATGGAGGTAATGTTTC-3′ and 5′-CCTGGCTGATATAGACAATCTTATGAGAAG-3′; and Cre, 5′-AGGTTCGTTCACTCATGGA-3′ and 5′-TCGACCAGTTT AGTTACCC-3′ (Fig. EV1C). The *2-loxP* allele was transmitted to the germline using blastocyst injection to establish the line #38. Rat insulin promoter-driven *Cre* (*Rip2-Cre*) transgenic mice (Kulkarni et al, 1999; Postic et al, 1999) were obtained from the Jackson Laboratory and crossed with line #38. The ROSA-STOP reporter mice (Soriano, 1999) were kindly provided by Dr. P. Soriano (Mt. Sinai School of Medicine) and crossed with *Rip2-Cre* mice to verify the specificity of Cre-expressing cells (Fig. EV1D). KIF5B deficiency in cKO cells was verified by immunofluorescence microscopy as described previously (Tanaka et al, 1998). These mice were maintained on a C57BL/6J background in a specific pathogen-free environment under a 14/10-h light/dark cycle under institutional regulations. *Lifeact-mCherry* mice were a kind gift of Drs. Michael Sixt and Roland Wedlich-Soldner (Max Planck Institute for Biochemistry, Germany) (Riedl et al, 2010).

Institutional approval for mouse experiments was received from the Institutional Animal Care and Use Committee of the University of Tokyo Graduate School of Medicine (#MP15-118, #MP20-92), and the experiments were conducted under ethical regulations.

## Histology

For immunohistochemistry and enzyme histochemistry of the tail region of the pancreas, mice were anesthetized and perfused with 4% paraformaldehyde in PBS to generate cryosections. β-galactosidase activity was stained with 0.5 mg/mL X-gal, 10 mM $K_3[Fe(CN)_6]$, 10 mM $K_4[Fe(CN)_6]$, 0.01% Na desoxycholate, 0.02% Nonidet P-40, 20 mM Tris-HCl, and 100 mM K-phosphate buffer [pH 7.4] according to previously described methods (Joyner, 2000). Immunohistochemistry was performed as previously described (Kanai et al, 2000). Fixed cells were labeled with a rabbit anti-KIF5B antibody (Tanaka et al, 1998) and a mouse anti-insulin antibody (Sigma), followed by Alexa-conjugated secondary antibodies (Invitrogen), observed using LSM510 or LSM710 confocal laser scanning microscopes (ZEISS), and analyzed using the MosaicJ plugin (Thevenaz and Unser, 2007) on ImageJ software (Abramoff et al, 2004).

## Blood sugar test

The intraperitoneal glucose tolerance test (IPGTT) was performed as previously described (Ohtsubo et al, 2005; Yang et al, 2014). In brief, 4-month-old male mice were starved overnight and intraperitoneally injected with 2 g glucose/kg body weight. Their blood glucose levels were measured at the indicated time points using a Medisafe-Mini blood glucose meter (Terumo, Tokyo, Japan).

## Pancreatic islets

Pancreatic islets were recovered from male mice using 8 g/L collagenase digestion (C-7657, Sigma) in Krebs Ringer Bicarbonate (KRB) solution (Daniel et al, 2002) as described previously (Szot et al, 2007; Yang et al, 2014). Prior to the experiments, the islets were cultured in RPMI-1640 medium (Invitrogen) with 10% FCS

and antibiotics in a 5% $CO_2$ atmosphere at 37 °C for 3 h to overnight as described previously (Carter et al, 2009).

Islet perifusion assay (Fig. 1F) was performed as previously described (Noda et al, 1996). In all, 6–8-month-old mouse islets were stimulated by KRB solution containing 20 mM glucose, following a preincubation with KRB solution containing 2 mM glucose. The perifusates were subjected to ELISAs using LBIS Mouse Insulin ELISA Kit (WAKO-Shibayagi, Japan) according to the manufacturer's protocol. Only the samples that reverted to the initial level of secretion after the stimulation period were subjected to statistical analyses. The mean insulin secretion on the period without stimulants was calculated as basal insulin secretion of each trial. Increments in 0–10 min and that in 10–44 min over the basal level secretion were calculated as first- and second-phase GSIS, respectively (Fig. 1G).

For the bulk insulin secretion assay (Fig. 5F), four islets from adult cKO mice in each tube were preincubated in KRB solution supplemented with 2.8 mM glucose. Then they were once rinsed with the same solution, and stimulated with 1 mL of KRB solution containing either 2.8 mM glucose, 16.7 mM glucose, or 16.7 mM glucose plus 1 µM Bay-K 8644 (#S7924, Selleck) at 37 °C for 1 h in a water bath. The supernatants were subjected to LBIS Mouse Insulin ELISA Kit (Luminescent type; WAKO-Shibayagi, Japan) according to the manufacturer's protocol.

The ATP/ADP ratio was measured as described previously (Lu et al, 2010) using an ApoSENSOR ADP/ATP Ratio Assay Kit (#K255-200, BioVision) following the manufacturer's instructions.

## Cell lines

The MIN6 cells (Miyazaki et al, 1990) were kindly provided by Dr. Jun-ichi Miyazaki (Osaka University). The 804 G cells (Bosco et al, 2000) were kindly provided by Dr. Jonathan Jones (Northwestern University Medical School). The Ins1 cells (Asfari et al, 1992) were kindly provided by Dr. Claes B. Wollheim (University Medical Center, Geneva, Switzerland).

## Primary culture of pancreatic beta cells

For the primary culture, Lab-Tek chambered coverslips (Nalge Nunc) or 35-mm glass bottom dishes (Matsunami) were coated with a conditioned medium of 804 G cells (Bosco et al, 2000; Langhofer et al, 1993). Precultured islets were incubated with 0.04% EDTA in PBS at 37 °C for 10 min and gently dissociated into single cells by pipetting. Following a medium change through centrifugation, islet cells were plated and cultured in KRB medium or in Ins1 medium [10 mM HEPES pH 7.4 (Gibco), 10% fetal bovine serum (Sigma Aldrich), 1 mM Na pyruvate (Gibco), 50 mM 2-mercaptoethanol (Sigma Aldrich), penicillin–streptomycin (Gibco), RPMI-1640 with L-glutamine (Gibco)] (Asfari et al, 1992) in a 5% $CO_2$ atmosphere at 37 °C and subjected to analyses within 1 week. Beta cells were identified as round and >10 µm cells. Coating with 804 G conditioned medium significantly stabilized the attachment of beta cells to the matrix and improved their viability.

## Antibodies

A rabbit anti-$Ca_V$1.2 antibody (N-17-R, #sc-16229-R, RRI-D:AB_2228387), a rabbit anti-$K_{ir}$6.2 antibody (H-55; #sc-20809;

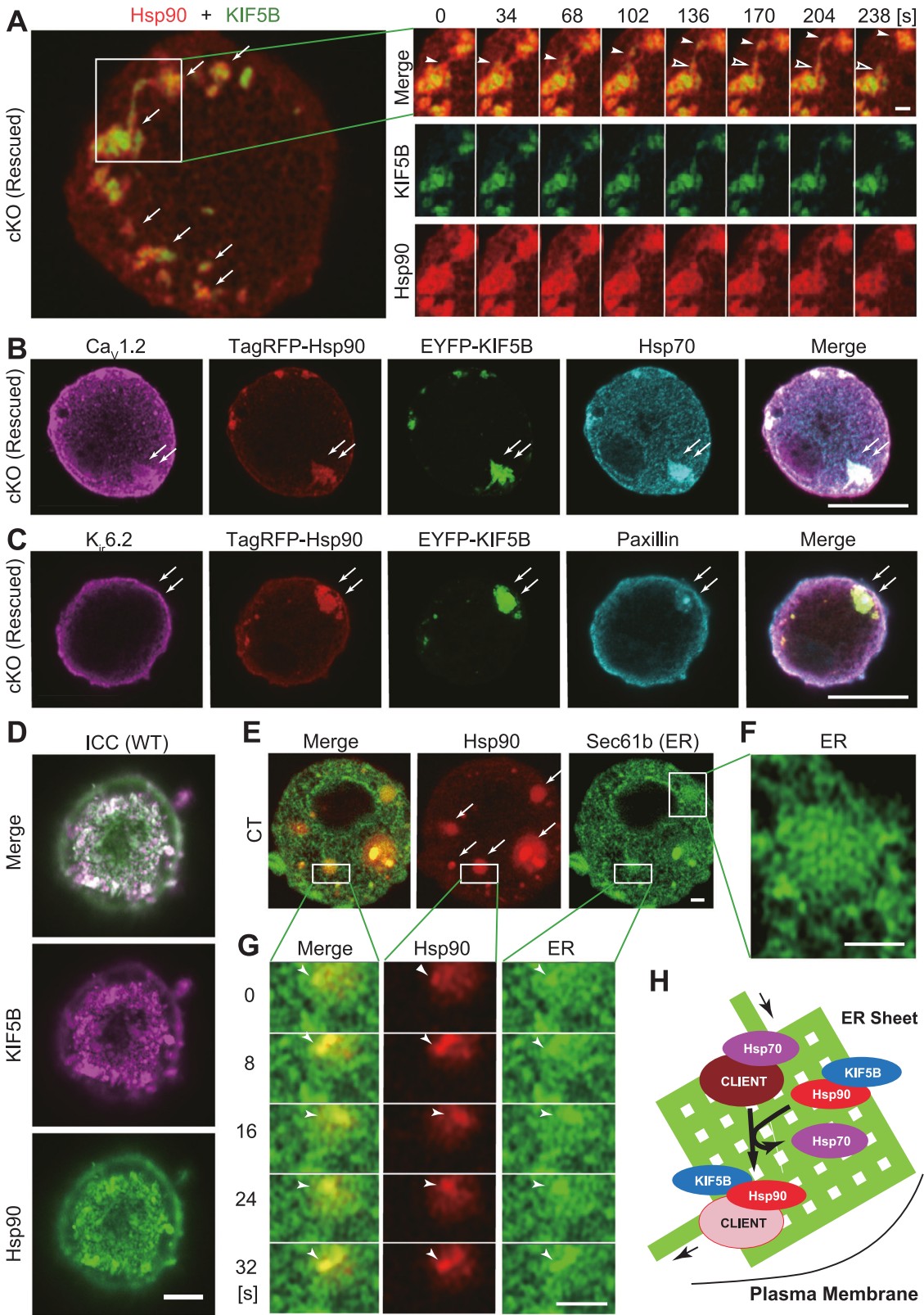

**Figure 8. KIF5B recruits Hsp90 onto ER sheets for proper Ca$_V$1.2 protein folding.**

(A) Time-lapse imaging in the bottom of a rescued primary cKO beta-cell, expressing tagRFP-Hsp90 (red) and KIF5B-EYFP (green). Scale bar, 1 μm. Arrows, microdroplets. Note that two co-accumulated microdroplets appeared to exchange their components through a double-labeled dynamic tubule (open and closed arrowheads). Corresponding to Fig. EV2E and Movie EV5. (B, C) Immunofluorescence microscopy in the bottom of primary cKO beta cells expressing tagRFP-Hsp90 (red) and KIF5B-EYFP (green), against Ca$_V$1.2 and Hsp70 (B) and against K$_{ir}$6.2 and Paxillin (C). Scale bars, 10 μm. Arrows, co-accumulated microdroplets. Reproduced 5–10 times. (D) Immunocytochemistry of a wild-type islet cell against KIF5B (magenta) and Hsp90 (green) using Airyscan microscopy. Scale bar, 5 μm. Reproduced twice. (E–G) Live fluorescence microscopy of the bottom of CT primary beta cells expressing tagRFP-Hsp90 (red) and the ER marker mEmerald-Sec61β (green) using Airyscan microscopy with low (E) and high (F, G) magnifications and in a time sequence (G). Scale bars, 1 μm. Arrows, microdroplets. Arrowheads, a portion of the ER sheet accompanied by Hsp90 microdroplets. Corresponding to Movie EV6. (H) Schematic representation of a ER sheet where the Hsp70-to-Hsp90 chaperone exchange occurs for the quality control of ER client protein folding by the help of KIF5B molecular motor.

RRID:AB_2130466), a goat anti-PIP5Kα (PIPKIα) antibody (M-20, #sc-11775; RRID:AB_2268303), and a goat anti-SUR1 antibody (N-18, #sc-11226; RRID:AB_2130475) were purchased from Santa Cruz Biotechnology; a mouse anti-PIP$_2$ IgM antibody (#Z-A045, RRID:AB_427211) was from Echelon Research labs; a rabbit anti-GFP antibody (#598, RRID:AB_591819) was from MBL; a mouse anti-LC3 antibody (Clone LC3-1703, #CTB-LC3-2-IC, RRID:AB_10707197) was from Cosmo Bio; a rabbit anti-BK$_{Ca}$ (K$_{Ca}$1.1) antibody (#APC-151, RRID:AB_10915895) and a rabbit anti-Ca$_V$2.3 antibody (#ACC-006, RRID:AB_2039777) were from Alomone Labs; a mouse anti-syntaxin-1 antibody (#MAB336, RRID:AB_2196527) was from Millipore; a mouse anti-Na/K ATPase beta 2 antibody (#610914; RRID:AB_398231) and a mouse anti-paxillin antibody (#610051, RRID:AB_397463) were from BD Transduction Labs; a mouse anti-Hsc70/Hsp70 antibody (Clone BB70, #ADI-SPA-822-D, RRID:AB_2039252) and a rat anti-Hsp90 antibody (Clone 16F1, #ADI-SPA-835-D, RRID:AB_2039281) were from Enzo; a rabbit anti-STIP1 antibody (#15218-1-AP; RRID:AB_2255518) and a rabbit anti-calnexin-1 (CANX) antibody (#10427-2-AP, RRID:AB_2069033) were from Proteintech; a mouse anti-derlin-1 antibody (#SAB4200148, RRID:AB_10624068), a mouse anti-α-tubulin antibody (Clone DM1A; #CP06; RRID:AB_2617116), and a mouse anti-insulin antibody (Clone K36aC10, #I-2018, Sigma Aldrich; RRID:AB_260137) were from Sigma Aldrich; a rabbit anti-phospho Src antibody for pSFK (#ab32078; RRID:AB_2286707) was from Epitomics/Abcam; and a rabbit anti-KIF5B antibody (RRID:AB_2571745) was previously described (Tanaka et al, 1998). Horseradish-peroxidase-conjugated protein A (#NA9120V) and sheep anti-mouse IgG antibody (#NA931V) were purchased from Cytiva. Alkaline-phosphatase-conjugated goat anti-rabbit IgG (#59298), goat anti-mouse IgG (#59296), goat anti-rat IgG (#59291) antibodies were purchased from Cappel. Alkaline-phosphase-conjugated rabbit anti-goat IgG antibody (#61-1622) was purchased from Zymed. Alexa-conjugated phalloidin, anti-rabbit IgG, anti-mouse IgG and anti-goat IgG antibodies were purchased from Thermo Fisher.

## Gene silencing

For gene silencing, antisense miRNAs were transcribed from mammalian PolII-based expression vectors derived from a Block-iT Pol II miR RNAi Expression vector kit (Thermo Fisher). For *Kif5b* knockdown, the oligonucleotides 5′-TGCTGTTTCAGGGCTGA CTCCAAAGCGTTTTGGCCACTGACTGACGCTTTGGACAGC CCTGAAA-3′ and 5′-CCTGTTTCAGGGCTGTCCAAAGCGTCA GTCAGTGGCCAAAACGCTTTGGAGTCAGCCCTGAAAC-3′ (KD1); and 5′-TGCTGTTCTCTGTGACTCTGGATCTGGTTTTG GCCACTGACTGACCAGATCCAGTCACAGAGAA-3′ and 5′-CC

TGTTCTCTGTGACTGGATCTGGTCAGTCAGTGGCCAAAAC CAGATCCAGAGTCACAGAGAAC-3′ (KD2) were obtained from Invitrogen, annealed, and inserted into the provided vector with *tagRFP* cDNA as an expression marker and equally sujected to the knockdown analyses to obtain consistent results. For *Stip1* knockdown, the oligonucleotides 5′-CCTGTGCAGCCGTCCATAGGG CCTGTCAGTCAGTGGCCAAAACAGGCCTATGAGGACGGCT GCAC-3′ and 5′-TGCTGTGCAGCCGTCCTCATAGGCCTGTTT TGGCCACTGACTGACAGGCCTATGGACGGCTGCA-3′ (#Mmi520981, Invitrogen) were used. As a negative control, a scrambled sequence provided by the kit were used to exclude the possibility of nonspecific adverse effects of miRNA expression. Those miRNA expression vectors were introduced into adenoviral vectors using a ViraPower Adenoviral Expression System (Thermo Fisher), purified by CsCl centrifugation, and subjected to cells according to the manufacturer's protocols.

## Expression vectors

For the generation of a *TagRFP-Hsc70* expression vector, the *Hsc70* cDNA was recovered from a previously described *ECFP-Hsc70* expression vector (Yang et al, 2014) by *Sal*I and *Xho*I, enzymes, and ligated to *TagRFP-C* vector (Evrogen).

For generation of a *TagRFP-Hsp90* expression vector, the FANTOM I1C0020P08 clone (RIKEN) containing *Hsp90aa1* cDNA (MGI:3564069) was amplified by 5′-ACCCTCGAGCTATGCCTGA GGAAACCCAGACCCAAG-3′ and 5′-ACCGGTACCTTAGTCTA CTTCTTCCATGCGTGATGTGTC-3′ and ligated into *pTagRFP-C vector* (Evrogen) with *Xho*I and *Kpn*I sites.

For the *phogrin-Dronpa-Green1* and *phogrin-EGFP* expression vectors, mouse *phogrin* cDNA was ligated with *pDronpa-Green1* (MBL, Japan) or *pEGFP-N1* (Clontech), and transferred to the ViraPower Adenoviral Expression System (Thermo Fisher).

The *mEmerald-Sec61β* expression vector was a kind gift from Prof. Jennifer Lippincott-Schwartz through Addgene #90992 (Nixon-Abell et al, 2016). The *Ca$_V$1.2-EGFP* expression vector was a kind gift from Dr. Luis F. Santana (Navedo et al, 2010). A *synapto.pHluorin* Adenoviral expression vector was kindly provided by Dr. Gero Miesenboeck, Sloan-Kettering Institute for Cancer Research (Miesenbock et al, 1998). The FRET biosensors *pRaichu 1011x* and *pRaichu 1054x* were kind gifts of Dr. Michiyuki Matsuda (Univ Kyoto).

For rescuing the KIF5B cKO cells, mouse *Kif5b* full-length cDNA (Tanaka et al, 1998) was ligated into the *pEYFP-N1, pEYFP-C1* or *pECFP-C1* vectors (Clontech). The expression vectors were transduced into primary islet cells using the ViraPower Adenoviral Expression System (Thermo Fisher) or using lipofectamine-LTX transfection reagent

(Thermo Fisher). The mobility of expressed proteins on SDS–PAGE was verified by immunoblotting of transfected rat Ins1 cell lysates (Asfari et al, 1992). For KIF5B- and Hsc70 overexpression, MIN6 cells were transduced by Adenoviral expression vectors for ECFP-KIF5B and/or tagRFP-Hsc70 for 48 h, and subjected to immunocytochemistry or vesicle IP.

## Pharmacology

Cytochalasin B (CyB) was purchased from Sigma Aldrich (#C6762) and applied to cells at 10 μg/mL (Lacy et al, 1973). Ionomycin was purchased from Sigma Aldrich (#I9657) and applied to cells at 1 μM (Sekine et al, 2006). For the VGCC antagonists, 20 μM nifedipine (#N7634, Sigma Aldrich) against L-type VGCC (Gilon et al, 1997), 1 μM SNX482 (#RTS-500, Alomone Labs) against R-type VGCC (Bourinet et al, 2001), and 0.3 mM L-ascorbate (#A4034, Sigma Aldrich) against T-type VGCC (Nelson et al, 2007) were applied to cells for a negative control for electrophysiology. Bay-K 8644 was purchased from Selleck (#S7924) and applied to islets at 1 μM (Satin et al, 1995). Cycloheximide was purchased from Fuji-Film WAKO (#037-20991) and applied to cells at 30 μg/mL (Yang et al, 2014), in accordance with 25 μM leupeptin (#336-40413, Fuji-Film WAKO) or 10 μM MG-132 (#P1102, Enzo) as described (Chou and Deshaies, 2011; Di Biase et al, 2011).

## Fluorescence microscopy

Immunofluorescence microscopy was performed as described previously (Yang et al, 2014). Especially, primary cultured mouse pancreatic beta cells were fixed with 2% paraformaldehyde (PFA)/0.1% glutaraldehyde (GA) at 37 °C for 10 min, permeabilized with 0.1% Triton X-100 in PBS for 5 min at room temperature, blocked in 10% bovine serum albumin (BSA) in PBS for 10 min at room temperature, and subjected for immunostaining using CanGet-Signal Immunostain Solution A (TOYOBO) according to the manufacturer's protocol. The samples were then subjected to a confocal laser scanning microscope (LSM710 equipped with a GaAsP detector and LSM780-Airyscan, ZEISS), fast confocal laser scanning microscope (LSM 5LIVE-Duo, ZEISS), or a TIRF/STORM microscope (ELYRA P.1, ZEISS, equipped with an iXon+ EM-CCD camera, Andor) as previously described (Nakata et al, 2011; Tanaka et al, 2016; Wang et al, 2022) and analyzed using ImageJ software (Schindelin et al, 2015). For quantification of the membrane channels in an optical section, an area within 0.8 μm of the surface was selected and measured for mean signal intensity.

For degradation assays, cells were treated with 30 μg/mL CHX with either 25 μM leupeptin or 10 μM MG-132 as described (Chou and Deshaies, 2011; Di Biase et al, 2011).

For the BFA washout assay, the cells were transduced with a rabbit $Ca_v1.2$-EGFP expression vector (Navedo et al, 2010) in the presence of 1 μM brefeldin A (Sigma Aldrich) at 19.5 °C overnight. Then, the cells were washed out using complete medium twice, cultured at 37 °C for the indicated times, and subjected to confocal laser scanning microscopy as described previously (Nakata and Hirokawa, 2003; Tanaka et al, 2023).

Insulin granule exocytosis was imaged using TIRF microscopy with a synapto.pHluorin adenoviral expression vector (Miesenbock et al, 1998; Tsuboi and Rutter, 2003), treated with secretagogues and/or drugs as indicated, and observed with an ELYRA P.1 microscope (ZEISS) at 30 ms intervals for 1 min, as previously

described (Aoki et al, 2010; Ohara-Imaizumi et al, 2007). For F-actin imaging, cells were treated with 10 μg/mL cytochalasin B (CyB; Sigma Aldrich) (Lacy et al, 1973), or 1 μM ionomycin (Sigma Aldrich) (Sekine et al, 2006), fixed, and stained with Alexa-conjugated phalloidin (Thermo Fisher) according to the manufacturer's protocols. For time-lapse analysis of F-actin, primary beta cells were recovered from Lifeact-EGFP transgenic mice, which were kindly provided by Drs. Michael Sixt and Roland Wedlich-Soldner (Max Planck Institute for Biochemistry, Germany) (Riedl et al, 2010), and subjected to Kif5b gene silencing and observed in time-lapse analysis with an LSM710 microscope (ZEISS).

For observation of the cortical insulin granules, the phogrin-Dronpa-Green1 cDNA expression vector was transduced to primary beta cells using ViraPower adenovirus system (Invitrogen). The cells were then fixed in 4% paraformaldehyde/PBS for 10 min and observed using a TIRF/PALM microscope (ZEISS). For tracking of granule movements, primary beta cells were transduced with phogrin cDNA ligated with pEGFP-N1 vector (Clontech), treated successively with Ins1 media containing 2 mM and 20 mM glucose respectively for 30 min, and observed using an LSM 5LIVE-Duo microscope (ZEISS; 30 frames/17.7 s) or a CSU-X1 spinning disc microscope (Yokogawa; 200 frames/100 s) immediately after removing the cells from the $CO_2$ incubator. The images were analyzed using Imaris software for particle tracking (Bitplane AG).

## Imaging of F-actin

For F-actin imaging, the cells were fixed using half-Karnovsky fixative [2% paraformaldehyde, 2.5% glutaraldehyde, and 0.1 M cacodylate buffer (pH 7.4)] at 37 °C for 15 min, permeabilized with 0.1% Triton X-100 in PBS at room temperature for 5 min, blocked with 10% BSA in PBS at room temperature for 10 min, stained with phalloidin conjugated with Alexa-568 or Alexa-488 (Thermo Fisher) in the blocking solution, washed with PBS for 5 min three times, observed using a confocal laser scanning microscope (LSM 5LIVE-Duo, LSM710, LSM780-Airyscan, ZEISS).

For quantification of cortical F-actin (Fig. 4), the cortical fluorescent signals of optical sections, approximately at the level of half of the cell height, were selected with the help of the "Cell Outliner" plugin of the ImageJ software, and the median signal of each selection was analyzed.

## Rho-family GTPase activities

For imaging the activities of Rac1 and Cdc42 GTPases, pRaichu 1011x and 1054x vectors (Kiyokawa et al, 2006), kindly provided by Dr. Michiyuki Matsuda, Kyoto University, were transduced into the cells via adenoviral vectors, respectively. Cells were then pretreated with Ins1 medium containing 2 mM glucose for 30 min and stimulated with 20 mM glucose for the indicated period. The YFP/CFP ratio was then measured at 458 nm excitation using an LSM 5LIVE-Duo confocal microscope (ZEISS) and quantified using ImageJ software (Abramoff et al, 2004).

## Ca²⁺ imaging

$Ca^{2+}$ imaging was performed with loading the cell with Fluo4-AM (Dojindo, Japan) as previously described (Tanaka et al, 2016; Tanaka et al, 2005; Wang et al, 2018), using LSM710 and LSM780

confocal laser scanning microscopes (ZEISS). The recording was started with Ins1 medium without glucose, and then continued with stimulation with 20 mM glucose. Finally, the medium was supplemented with 30 mM $CaCl_2$, 125 mM KCl, and 10 µM ionomycin, and then washed with DDW, to normalize the experimental data as ΔF/Fmax.

### Patch-clamp recording

Patch-clamp recording of primary cultured beta cells at DIV1–2 were performed as previously described (Rorsman and Trube, 1986). The bath solution was 138 mM NaCl, 5.6 mM KCl, 1.2 mM $MgCl_2$, 2.6 mM $CaCl_2$, and 10 mM HEPES-NaOH [pH 7.4]; with 0 or 10 mM of glucose. The pipette was filled with 125 mM KCl, 30 mM KOH, 4 mM $MgCl_2$, 3 mM $Na_2ATP$, 2 mM $CaCl_2$, 10 mM EGTA, and 5 mM HEPES-KOH [pH 7.15]. The intracellular potential was recorded stepwise under 0 and 10 mM of glucose in the external medium. Inward $Ca^{2+}$ currents were recorded with a voltage clamp under 10 mM glucose with or without VGCC inhibitor cocktail that is 20 µM nifedipine against L-type VGCC (Gilon et al, 1997), 1 µM SNX482 against R-type VGCC (Bourinet et al, 2001), and 0.3 mM ascorbate against T-type VGCC (Nelson et al, 2007). The data acquisition and processing were performed using Axon pCLAMP 10 software (Molecular Devices).

### Immunoprecipitation

Immunoprecipitation was performed as previously described (Ueno et al, 2011). MIN6 cells were transduced with scrambled-control (SC) or KIF5B-knockdown (KD) adenoviral vectors for 6 days. They were then treated by 10 µM MG-132 for 6 h, and harvested using 10 mM HEPES [pH 7.4], 150 mM NaCl, 0.1% Triton X-100, and protease inhibitors (Roche). Postnuclear fractions were precipitated against protein A Sepharose beads (Cytiva) or µMACS Protein A Microbeads (Miltenyi Biotech) conjugated with 2 µg of antibody or normal rabbit IgG (Cappel). After extensive washing, the beads were boiled with 2 × Laemmli's sample buffer and subjected to immunoblotting using CanGetSignal immunoreaction enhancer solutions 1 and 2 (TOYOBO) and Amersham ECL Prime Western Blotting Detection System (Cytiva).

### Proximity ligation assay

Proximity ligation assay was conducted using a DuoLink system (Sigma Aldrich) with rabbit anti-$Ca_V$1.2, mouse anti-Hsp70, and rat anti-Hsp90 antibodies according to the manufacturer's protocols. The anti-mouse probe of the kit cross-reacted rat IgG, so that an anti-mouse (plus) and anti-rabbit (minus) pair of the Duolink probes was applied. The samples were subjected to a confocal laser scanning microscope (LSM710 and LSM780-Airycan, ZEISS); and z-projection at the maximum intensity was conducted using ImageJ ver. 1.54i software.

### qRT-PCR

qRT-PCR was performed as previously described (Tanaka et al, 2016; Yang et al, 2014). SC/KD-miRNA-transduced MIN6 cells were cultured in 2 mM glucose and treated with a total RNA isolation mini kit (Agilent) and 1st strand synthesis kit (Origine). The 1st strand cDNA was subjected to real-time PCR on a LC480

thermal cycler instrument II (Roche) using SYBR Premix Ex Taq (Tli RNaseH Plus, #RR420, TaKaRa) with the following primers: $K_{ir}6.2$ cDNA with 5′-CTCATCATCTACCACGTCATCGA-3′ and 5′-GTTTCTACCACGCCTTCCAAGA-3′ (Camerino et al, 2013); $Ca_V$1.2 cDNA with 5′-TCCTCATCGTCATTGGGAG-3′ and 5′-AG TTCTCCTCTGCACTCATAG-3′; $Ca_V$2.3 cDNA with 5′-GACTC TCATGTCACCACCGC-3′ and 5′-AGCCACTGGCATGTTCAT CA-3′; and beta-actin with 5′-GCACCACACCTTCTACAATG AG-3′ and 5′-GAAGGTCTCAAACATGATCTGG-3′.

### Quantification and statistical analysis

Insulin secretion levels, ATP/ADP ratio of islets, glucose uptake of beta cells, actin remodeling, SFK activation, immunofluorescence intensities, protein expression profiles, chaperone-binding capacities, proximity ligation assay, and blood sugar levels were subjected to one-or two-sided unpaired Welch's $t$ test or one-way ANOVA including Dunn's multiple comparisons test following a Kruskal–Walls test. $Ca^{2+}$ transients of beta cells, membrane depolarization of beta cells, Rho-family GTPase activation, and islet perifusion results were subjected to two-way ANOVA. The movements of insulin granules were subjected to MSD analysis using the MSD Analyzer plugin (https://tinevez.github.io/msdanalyzer/) on the MATLAB platform (Tarantino et al, 2014). The blinding procedure was not applied. All error bars indicate the mean ± SEM. In the case of immunofluorescence microscopy and qRT-PCR, the statistics is based on a group of dishes that are stained at the same time, so the error bars represent the variety between cells from each one dish (Figs. 3B, 4B,D,E, 5C,E,H, 6C,E, and 7B,H). In the case of western blotting, Rho-family GTPase activity, $Ca^{2+}$ transients, electrophysiology, BFA washout, and proximity ligation assay, the data were normalized using internal control so that the statistics is based on more than one experiments (Figs. 4F,G,H,J, 5B, 6E,J, and 7D). Also, systemic and islet experiments involve data from multiple individuals (Figs. 1C–G, 2E,G,J,K, and 3C–H). Statistical details of the experiments can be found in the figure legends, figures, results, and "Methods" section.

## Data availability

The source data of this paper are collected in the following database record: https://data.mendeley.com/datasets/ygvn8mh68r/2.

The source data of this paper are collected in the following database record: biostudies:S-SCDT-10_1038-S44319-024-00246-y.

## Peer review information

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

## Acknowledgements

The authors thank Tetsuo Noda (Cancer Inst, Japan) for ES cell technology,
Mitsuhisa Komatsu (Shinshu Univ) for islet recovery, Junqing Sun (Univ Tokyo)
for patch clamp recording, Phil Soriano (Mt. Sinai School of Medicine) for the
*ROSA-STOP* mice, Michael Sixt and Roland Wedlich-Soldner (Max Plank
Institute of Biochemistry, Germany) for the *Lifeact-mCherry* transgenic mice,
Claes B Wollheim (University Medical Center, Geneva, Switzerland) for the
Ins1 cells, Jun-ichi Miyazaki (Osaka University) for the MIN6 cells, Jonathan
Jones (Northwestern University Medical School) for the 804G cells, Gero
Miesenboeck (Sloan-Kettering Institute for Cancer Research) for the
*synapto.pHluorin* cDNA, Jennifer Lippincott-Schwartz (NIH) for the *mEmerald-
Sec61β* expression vector, Michiyuki Matsuda (Kyoto Univ) for the *Raichu*
biosensor cDNAs, and Luis F. Santana (University of Washington) for the
*Ca$_V$1.2-EGFP* cDNA. We are also grateful to Yuki Hashimotodani (Doshisha
Univ), Takuya Okada (Physio-tech, Co., Ltd., Japan), Ingo Kleppe and Bernhardt
Zimmermann (ZEISS), Fumiyoshi Ishidate (Kyoto Univ); and Shuo Wang,
Yoshinobu Shimazawa, Tatemitsu Rai, Shinsuke Niwa, Kyosuke Nakajima,
Shuzo Hasegawa, Momo Morikawa, Nobuhisa Onouchi, Takeshi Akamatsu,
Hiromi Sato, Haruyo Fukuda, and previous members of the Hirokawa
laboratory for technical assistance, support, and valuable discussions. This
study was supported by JSPS KAKENHI grant numbers JP23000013 and
JP16H06372 to NH and JP20K06634 to YT; a research grant-in-aid from JEOL
Ltd. to NH; Research Grants-in-Aid from Uehara Memorial Foundation, ONO
Medical Research Foundation, and Japan IDDM Network to YT.

## Author contributions

**Yosuke Tanaka**: Conceptualization; Data curation; Funding acquisition;
Investigation; Methodology; Writing—original draft; Writing—review and
editing. **Atena Farkhondeh**: Investigation. **Wenxing Yang**: Investigation. **Hitoshi
Ueno**: Investigation. **Mitsuhiko Noda**: Conceptualization; Investigation;
Methodology; Writing—review and editing. **Nobutaka Hirokawa**:
Conceptualization; Funding acquisition; Project administration; Writing—
review and editing.

Source data underlying figure panels in this paper may have individual
authorship assigned. Where available, figure panel/source data authorship is
listed in the following database record: biostudies:S-SCDT-10_1038-S44319-
024-00246-y.

## Disclosure and competing interests statement

The authors declare no competing interests.

# Expanded View Figures

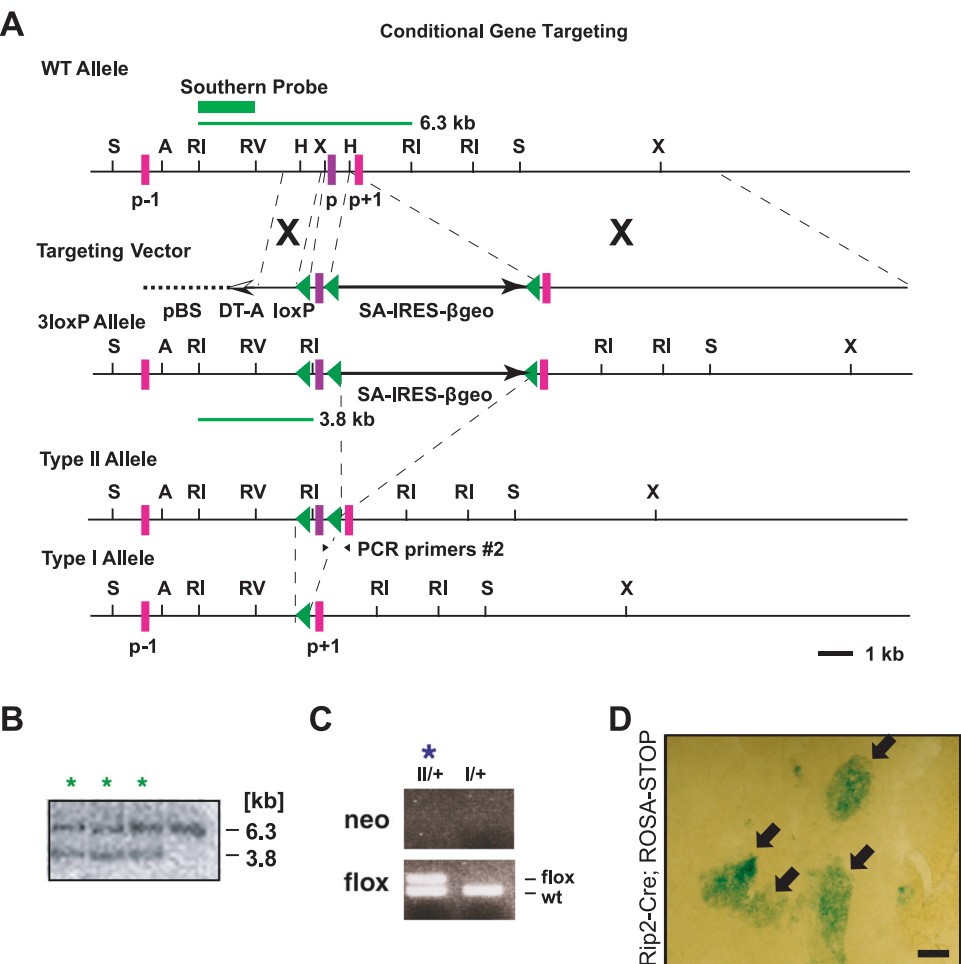

**Figure EV1.  Conditional knockout of mouse *Kif5b* gene.**

(**A–D**) Establishment of beta-cell-specific *Kif5b* gene conditional knockout (cKO) mice, represented by a gene targeting strategy in mouse ES cells (**A**), Southern blotting screening for homologous recombinants (**B**; asterisks); genotyping PCR for the floxed allele (**C**: asterisk); and characterization of Rip2-Cre activity in a pancreas section detected by a LacZ reporter, ROSA-STOP mice (**D**). p, the 74 bp P-loop exon flanked by *loxP* sites (green triangles). S, *SalI*; A, *ApaI*; RI, *EcoRI*; RV, *EcoRV*; H, *HindIII*; X, *XbaI*. Arrows in (**D**), specific Cre/*loxP* recombination sites in the pancreas of a *Rip2-Cre ROSA-STOP* double heterozygous mouse. Scale bar, 100 μm. Corresponding to Fig. 1A–C.

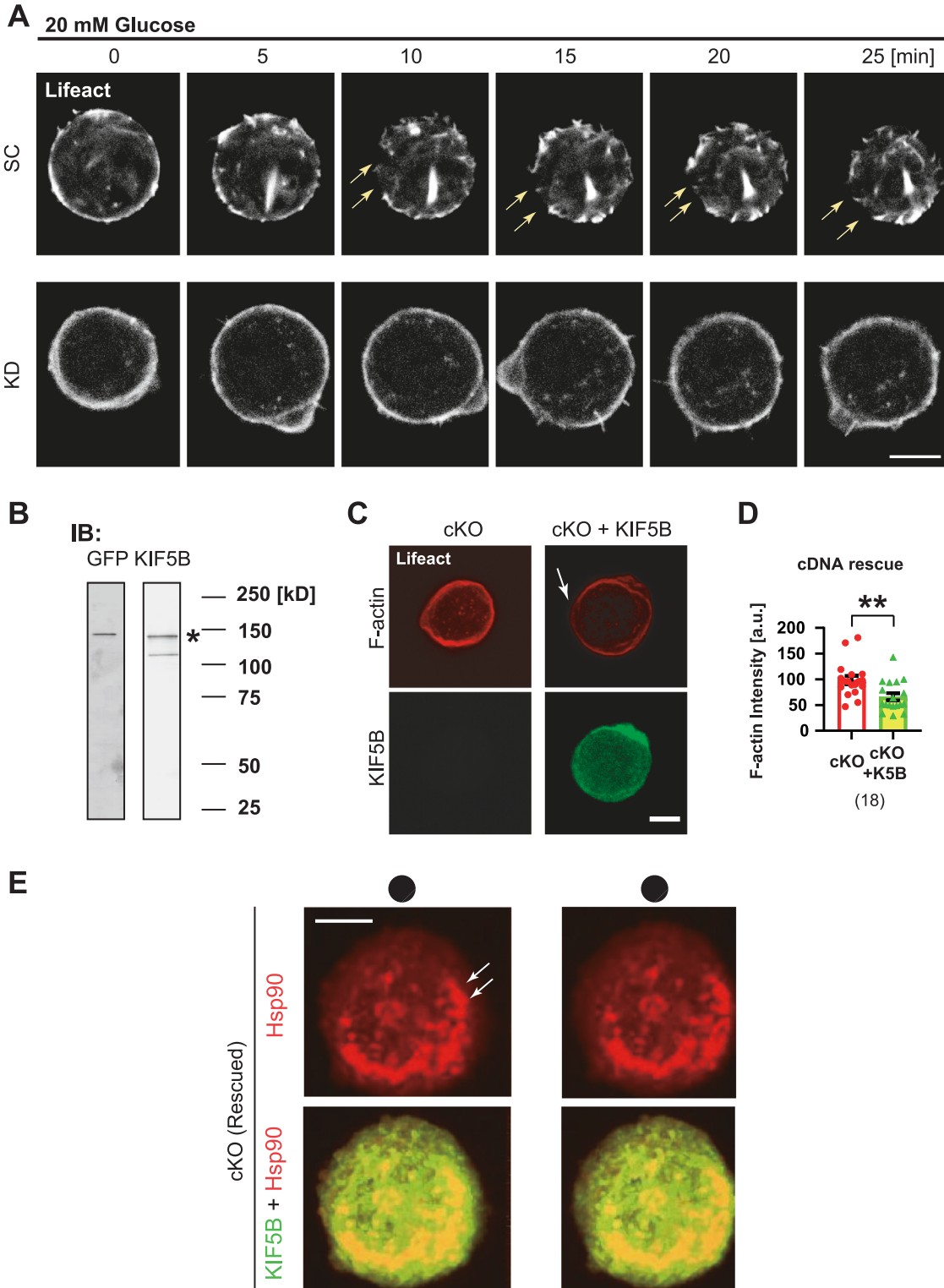

**Figure EV2. KIF5B facilitates cortical actin remodeling.**

(A) Time-lapse study of glucose-stimulated actin remodeling of primary beta cells from *Lifeact-mCherry* transgenic mouse pancreas, transduced with scrambled-control (SC) or KIF5B-knockdown (KD) miRNA expression vectors. Scale bar, 5 µm. Arrows, actin remodeling. Corresponding to Fig. 4A and Movie EV3. (B–D) Rescue study of the glucose-stimulated actin remodeling in cKO primary beta cells by transducing KIF5B-EYFP, represented by immunoblotting of the expressed proteins in Ins1 cells using a mouse anti-GFP antibody and a rabbit anti-KIF5B antibody (B), *Lifeact-mCherry* transgene labeling (C), and F-actin quantification (D). Scale bar, 5 µm. Asterisks in (B), bands for tagged KIF5B. Arrow in (C), actin remodeling. **P = 0.0042, two-sided unpaired Welch's *t* test; *n* = 18. Corresponding to Fig. 4A. (E) Stereoscopic fluorescence microscopy of a cKO primary beta-cell expressing tagRFP-Hsp90 (red) and KIF5B-EYFP (green). Scale bar, 5 µm. Data are represented by the mean ± SEM. Corresponding to Fig. 8A.

