## [Peer Review File · EMBO Reports]

Kinesin-1 mediates proper ER folding of the CaV1.2 channel and maintains mouse glucose homeostasis

Yosuke Tanaka, Atena Farkhondeh, Wenxing Yang, Hitoshi Ueno, Mitsuhiko Noda, and Nobutaka Hirokawa

Corresponding author(s): Nobutaka Hirokawa (hirokawa@m.u-tokyo.ac.jp)

Review Timeline:

Submission Date:	20th Dec 23
Editorial Decision:	1st Feb 24
Appeal Received:	12th Feb 24
Editorial Decision:	12th Mar 24
Appeal Received:	18th Apr 24
Editorial Decision:	26th Jun 24
Revision Received:	8th Jul 24
Accepted:	22nd Aug 24

Transaction Report:

Dear Prof. Hirokawa

Thank you for the submission of your manuscript to EMBO reports. We have now received the final referee report, and all reports are copied below.

I am sorry to say that the evaluation of your manuscript is not a positive one. As you will see, while the referees acknowledge that the manuscript reports interesting observations, they all point out that the data are not sufficient to support the main conclusions. The referees are concerned that the study does not provide compelling evidence that Hsp70 or Hsp90 association with CaV1.2 is necessary for its folding. The evidence that Hsp70/90 bind to CaV1.2 is based on overexpression. Other major concerns relate to causality and to whether CaV1.2 is the major downstream effector of KIF5B. Referee #3 further notes that there is a significant reduction in first-phase GSIS in cKO cells indicating that KIF5B also regulates 1st phase GSIS, which remains unexplained.

Given these concerns, the amount of work required to address them, the uncertain outcome of these experiments, and the fact that EMBO reports can only invite revision of papers that receive enthusiastic support from a majority of referees, I am sorry to say that we cannot offer to publish your manuscript. Please note that this decision is based on the scientific arguments listed above and not on the 2nd paragraph of referee #2 as we realize that the partial cut-off of bands in Figure 6B might also be due to pdf compression artefacts.

However, in case you feel that you can fully address the referee concerns (as mentioned above and in their reports) in a timely manner and obtain data that would considerably strengthen the message of the study, then we would have no objection to consider a new manuscript on the same topic in the near future. Please note that if you were to send a new manuscript this would be treated as a new submission rather than a revision and would be reviewed afresh, also with respect to the literature and the novelty of your findings at the time of resubmission.

At this stage of analysis, I am sorry to have to disappoint you. I nevertheless hope, that the referee comments will be helpful in your continued work in this area, and I thank you once more for your interest in our journal.

Yours sincerely,

Referee #1:

In this manuscript, the authors found that beta-cell-specific deletion of a kinesin molecular motor KIF5B in mice resulted in glucose intolerance. Mechanistically, they found that KIF5B and Hsp90 colocalized at ER tangles on the bottom of cells and demonstrated that KIF5B facilitated Hsp70-to-Hsp90 chaperone exchange during the biogenesis of the voltage-gated Ca²⁺ channel CaV1.2, which mediates glucose-stimulated Ca²⁺ transients. The role of KIF5B in regulating glucose-stimulated insulin secretion (GSIS) is well-recognized, but the underlying mechanism remains unknown. This study is therefore potentially interesting as it provides the possible explanation for the function of KIF5B in regulating GSIS. However, several issues have to be clarified before consideration of publication in EMBO reports.

1, Throughout the manuscript, does ca mean Ca²⁺?

2, "Very intriguingly, the resting membrane potential of cKO beta cells was significantly depolarized to -40 mV, while that of CT cells was properly maintained at -60 mV (Fig. 4I and J)"

Clearly, KIF5B affected the resting membrane potential of beta cells, suggesting that KIF5B has more client proteins besides CaV1.2. Does higher resting membrane potential of cKO beta cells contribute to the defect in GSIS?

3, "Consequently, spontaneous action potentials were increased in the resting state, but the size of glucose-stimulated depolarization was decreased....."
Any evidence?

4, In Figure 5B, the expression level of CaV2.3 (which also participates in GSIS) was also reduced in cKO beta cells. Does CaV2.3 also contribute to the function of KIF5B in regulating GSIS? Does KIF5B also regulate the membrane trafficking of CaV2.3?

5, "We compared the chaperone binding capacities of CaV1.2 and Kir6.2 by immunoprecipitation among the KIF5B-KD system in the presence of the proteasome inhibitor MG132 (Fig. 6B and C)."

The quality of Figure 6B is low.

6, In Figure 6 and Figure 7, the authors used an overexpression system to demonstrate the interaction between KIF5B and HSP90 (or HSP70). However, the system is artificial and may not reflect the molecular interaction in the native states.

7, There is no convincing evidence showing that CaV1.2 is the major downstream effector of KIF5B in regulating GSIS. Can activators of CaV1.2 rescue the phenotype of KIF5B cKO beta cells?

Referee #2:

Tanaka et al. report experiments that are interpreted to indicate that the moto protein KIF5B affects voltage gated calcium channels (CaV1.2) and the folding of these channels facilitated by the chaperones HSP70 and HSP90. Although the data suggest a role for KIF5B in affecting CaV1.2 function, the data supporting the involvement and effects of the HSP chaperones is very indirect, making the role of these potential interactions in CaV1.2 unclear. Further, there are significant problems with the data presented in support of the studies of HSP interactions.

There are serious problems with the western blot data presented in Fig. 6B and C. Each of the gel strips in the right hand side of the figure have the same systematic diagonal clipping (most evident in the strip for HSP90, but present in all strips). Consequently, the strips are missing complete bands for all of the proteins probed in the CaV1.2 samples and both Kir6.2 samples. Hence, none of the values presented in Fig. 6C can be considered accurate.

Besides this technical problem, there is no convincing evidence that the association of CaV1.2 with Hsp90 or Hsp70 is leading to properly folded channels. Indeed, the amount of HSP70 associated with the channel (which could indicate channels that are misfolded) appears to be the same in the SC and K5B-KD2 panels (with the caveat above about the gels).

It should also be noted that the 'SC' control cells are not well described in the text, making it difficult to understand the nature of this control.

Additionally, the data suggesting knockdown of CaV1.2 current density (Fig. 4H) should be validated using standard blockers of CaV1.2 to show whether the measured currents arise from CaV1.2 or some other class of calcium channels.

Referee #3:

1. Does this manuscript report a single key finding? YES.

The manuscript by Tanaka et. al., reports that the kinesin superfamily protein KIF5B regulates actin-cytoskeleton reorganization during the second phase of GSIS via acting as a scaffolding factor for HSP70-HSP90 exchange for Cav1.2 in ER tangles, which is an essential step for Cav1.2 ER exit and Cav1.2 regulated Ca²⁺ mediated GSIS.

2. Is the reported work of significance (YES), or does it describe a confirmatory finding or one that has already been documented using other methods or in other organisms, etc (NO)? YES.

3. Is it of general interest to the molecular biology community? NO. In my opinion islet biology-diabetes research community will be more interested in this manuscript.

4. Is the single major finding robustly documented using independent lines of experimental evidence (YES), or is it really just a preliminary report requiring significant further data to become convincing, and thus more suited to a longer format article (NO)? YES, but need additional experiments as documented below in the comments to the author section.

Overview and overall evaluation: The manuscript by Tanaka et. al., using in vivo (KIF5B conditional knockout mouse model), and in vitro (KIF5B conditional knockout primary mouse islets and clonal beta-cells) approaches determined the mechanistic contribution of kinesin superfamily protein KIF5B in the regulation of second phase GSIS. Essentially, the authors showed that KIF5B regulates actin-cytoskeleton reorganization during the second phase of GSIS via acting as a scaffolding factor for HSP70-HSP90 exchange for Cav1.2 in ER tangles, which is an essential step for Cav1.2 ER exit and Cav1.2 regulated Ca²⁺ mediated GSIS. I found the paper to be novel and interesting and has the great potential to provide important knowledge relevant to the field of beta-cell function and diabetes. Also, the findings are appropriately discussed in the context of earlier literature and the manuscript is well written. However, I have certain concerns that need to be addressed which I detailed below. If appropriately

revised I recommend the manuscript for publication in EMBO Reports.

Comments to the author:

1. Please describe what the authors are referring to by "constitutive" and "Regulatory" insulin secretion in Fig. 1K-L.
2. Fig. 1K shows a marked reduction in first-phase GSIS in cKO islets, but the authors are discussing primarily defects in 2nd phase GSIS in the cKO islets. Please comment in the discussion section on the probable mechanism of KIF5B regulated 1st phase GSIS.
3. In Pg. 12, 2nd paragraph: Please also discuss the results of cKO primary islets (Fig. 4D-F) for clarity.
4. Page 13: 2nd paragraph, starting with "To investigate the mechanistic link between KIF5B and Ca dynamics,....": Please provide the western blot data for Cav 2.3, BKca, Syntaxin-1, Na/Kappa ATPase, HSP70, HSP90, calnexin, LC3, STIP1, and Derlin.
5. In Fig 6B, the IP blot: the portion of the IP blots are damaged (broken top part) near the bands of interest. So, I am unable to review the findings relevant to Fig 6B properly. Please provide undamaged sets of the IP blots for correct reviewing.
6. Page 15: "Although Cav1.2 and Hsp70 expression levels were both decreased in KD cells" The reduced level of HSP70 in input is not clear.
7. According to Fig 5J, reduction in Cav1.2 level in cKO cells can be restored by MG132 treatment. In Fig 6B, the cells are treated with MG132, but still, in KIF5B KO cells, the Cav1.2 level is markedly reduced. Please explain this discrepancy.
8. Fig 6F: what type of cells? Control or cKO?
9. In Discussion: "We then detected a significant defect in CaV1.2 expression in the posttranslational level (Fig. 5)": In the result section, the authors discussed post-transcriptional changes.
10. In Fig 6 (Immunoprecipitation), the authors treated MIN6 cells with 10 M MG132 for 6 days. Please provide the evidence for cell viability after this treatment.
11. Fig. 6D and E: Please provide information about the type of cell treatments representing red and Blue dots (Fig. 6D) and bars (Fig. 6 E).
12. Co-immunoprecipitation studies (similar to Fig. 6B) using control (wild type) cells and KIF5B overexpressed cells would be helpful for a clear understanding of the proposed mechanism.

** As a service to authors, EMBO Press provides authors with the ability to transfer a manuscript that one journal cannot offer to publish to another journal, without the author having to upload the manuscript data again. To transfer your manuscript to another EMBO Press journal using this service, please click on Link Not Available

February 6, 2024

Dr. Martina Rembold, PhD
Senior Editor
EMBO reports

Dear Martina,

I was deeply disappointed to read your negative decision on our paper EMBOR-2023-58692-T by Tanaka et al., regarding a new molecular mechanism of kinesin-based insulin secretion in pancreatic beta cells. I am afraid to say that the following three main criticisms against its publication are largely based on serious misinterpretation and/or misunderstanding of our results, which I would like to sincerely discuss with you.

Firstly, you mentioned “The referees are concerned that the study does not provide compelling evidence that Hsp70 or Hsp90 association with Ca_v1.2 is necessary for its folding. The evidence that Hsp70/90 bind to Ca_v1.2 is based on overexpression.” This is based on a serious misinterpretation of our data on Figure 6B, C, D, and E. In Figure 6B/C, we performed intrinsic IP of control and knockdown MIN6 cell lysates against Ca_v1.2 and K_{ir}6.2. In the consequence, we found that [1] Ca_v1.2 binds to both Hsp70 and Hsp90, while K_{ir}6.2 only binds to Hsp70 in the presence of KIF5B, but [2] the binding between Ca_v1.2 and Hsp90 is significantly missed in the absence of KIF5B. This result was further confirmed using nontransfected primary cultured mouse pancreatic beta cells from control and conditional knockout (cKO) mice in Figure 6D/E. We performed proximity ligation assay against [Ca_v1.2 and Hsp70] pair and [Ca_v1.2 and Hsp90] pair, respectively, where the association points were depicted as fluorescent dots in the cytoplasm. In the consequence, we found a significant reduction of Ca_v1.2 and Hsp90 binding in the cKO cells compared with the control cells. These experiments provide compelling evidence for that KIF5B is essential for [Ca_v1.2 and Hsp90] binding but not for [Ca_v1.2 and Hsp70] binding in nontransfected beta cells. We argue that these binding changes newly suggest that the well-established Hsp70-to-Hsp90 exchange for protein folding (Dahiya *et al*, 2022; Genest *et al*, 2019; Moran Luengo *et al*, 2018; Moran Luengo *et al*, 2019; Rutledge *et al*, 2022; Wegele *et al*, 2006) is dependent on KIF5B activity.

In addition, reviewer 2 mentioned “Indeed, the amount of HSP70 associated with the channel (which could indicate channels that are misfolded) appears to be the same in the SC and K5BKD2 panels (with the caveat above about the gels),” as negative evidence against our

argument, but this would be based on a serious misinterpretation of the basic idea. The reviewer's argument that **the amount of HSP70 associated with the channel could indicate channels that are misfolded** is very misleading, because it has been reported that Hsp70 folds the clients on the halfway and holds it, and that Hsp90 releases the client from the "deadlock" by Hsp70 to complete the folding process (Moran Luengo *et al.*, 2018). Thus, the binding of the client to Hsp70 should normally occur also on the first half of the folding pathway, which may not indicate "the amount of channels that are misfolded" as this reviewer has mentioned. Rather than that, the amount of Hsp90-Cav1.2 binding should indicate the correct amount of the channels that are "fully folded," of which significant decrease in our KIF5B-deficient cells provides the very evidence of that Cav1.2 becomes difficult to complete the protein folding without KIF5B. Accordingly, we would argue that our experimental evidence that the Hsp70-Cav1.2 binding is unchanged but that Hsp90-Cav1.2 binding is impaired in the absence of KIF5B provides strong direct evidence that KIF5B is essential for Hsp70-to-Hsp90 chaperone exchange for the Cav1.2 folding in the ER. We would admit this point was hard to understand for the readers in the previous version, which we would like to provide a full textual revision.

Secondly, you questioned the causality of Cav1.2 for the GSIS phenotype in KIF5B-deficient animals. Actually, Cav1.2 knockout mouse phenotype well phenocopied our KIF5B cKO mouse phenotype as we discussed in p.18 (Schulla *et al.*, 2003). Please look at the following glucose-stimulated insulin secretion curves:

[1] Insulin secretion from Cav1.2 knockout islets (from Schulla *et al.*) Open: KO, Closed: CT.

REDACTED: Figure 4 from Schulla *et al.*, 2003

[2] Insulin secretion from KIF5B cKO mouse islets (our Fig. 1K)

Figure for referees not shown.

As depicted here, Cav1.2-deficient and KIF5B-deficient islets similarly exhibited [1] elevation of basal insulin secretion (please note that [1-2] is larger than [1-1] in the lower figure); [2] significantly impaired 1st-phase secretion (please note that [2-2] is smaller than [2-1]); and [3] almost complete loss of the second-phase secretion (please note that [3-2] is significantly smaller than [3-1]). This phenotypic similarity and the significant loss of Cav1.2 protein in KIF5B-deficient islets can sufficiently suggest the major involvement of Cav1.2 in the phenotype of KIF5B deficient mice. We would admit that the absence of this logic in the current Results section seemingly provided a gap between Fig. 5 and Fig. 6 stories, which we would like to supplement for the readers' sake.

Finally, you mentioned "Referee #3 further notes that there is a significant reduction in first-phase GSIS in cKO cells indicating that KIF5B also regulates 1st phase GSIS, which remains unexplained," but the significance of this message seems to be misinterpreted. The significant **loss** of 1st phase GSIS can be fully interpreted from the significant loss of $[Ca^{2+}]_i$ transients in KIF5B-deficient beta cells, as it has been established that first-phase insulin secretion is largely due to intracellular Ca elevation (Shigeto *et al*, 2006). Shigeto et al. mentioned "first phase insulin secretion from MIN6 cells consists of at least three components: one dependent on influx of L-type voltage-dependent calcium channel (component 1), one involving calcium release from endoplasmic reticulum via ryanodine receptors (component 2), and the final one that somehow involves voltage-gated Na⁺ channels (component 3). Our experiment demonstrated that components 2 and 3 are independent of an increase of Ca²⁺ entry into the cells since these components were not affected in the condition of extracellular calcium depletion." As components 1 and 2 involves intracellular Ca transients, it can be well interpreted that the significant loss of Ca transients in KIF5B-deficient beta cells (Fig. 4G) are the main cause of significant loss of 1st-phase insulin secretion. We believe that Referee #3 has mainly suggested to supplement these discussions on the Ca-1st phase relationship, which should not be a serious conceptual failure.

Accordingly, we consider that the three major conceptual problems that you have mentioned can be well interpreted according to these logics, and would not largely harm the value of this study. We deeply thank the reviewers for other kind comments which were well documented, constructive, and enthusiastic for this study, and we think we can revise those technical points within 2 months as our revision plan provided separately in a point-by-point manner. We would apologize for the lack of some words in the present manuscript which can easily be misinterpreted or misunderstood, but we hope you will understand that these points are not a substantial error of our arguments and kindly reconsider your decision to allow us the revision as our proposal. Thank you very much for your kind help and we look forward to hearing a positive response from you soon.

Point-by-point response to the referees' comments

Tanaka et al.

February 5, 2024

Referee #1:

In this manuscript, the authors found that beta-cell-specific deletion of a kinesin molecular motor KIF5B in mice resulted in glucose intolerance. Mechanistically, they found that KIF5B and Hsp90 colocalized at ER tangles on the bottom of cells and demonstrated that KIF5B facilitated Hsp70-to-Hsp90 chaperone exchange during the biogenesis of the voltage-gated Ca²⁺ channel Ca_v1.2, which mediates glucose-stimulated Ca²⁺ transients. The role of KIF5B in regulating glucose-stimulated insulin secretion (GSIS) is well-recognized, but the underlying mechanism remains unknown. This study is therefore potentially interesting as it provides the possible explanation for the function of KIF5B in regulating GSIS. However, several issues have to be clarified before consideration of publication in EMBO reports.

1, Throughout the manuscript, does ca mean Ca²⁺?

Yes, we will fix it so throughout the manuscript.

2, "Very intriguingly, the resting membrane potential of cKO beta cells was significantly depolarized to -40 mV, while that of CT cells was properly maintained at -60 mV (Fig. 4I and J)" Clearly, KIF5B affected the resting membrane potential of beta cells, suggesting that KIF5B has more client proteins besides Ca_v1.2.

Yes, although Ca_v1.2 knockout islets provided a similar insulin release pattern as KIF5B cKO islets (Schulla *et al.*, 2003), we do believe that PIP5K α -induced PIP₂ production is another target of KIF5B (Fig. 5G), of which knockdown in INS1 cells also caused a similar phenotype of depolarized resting potential and massive decrease in GSIS, possibly through inactivation of K_{ATP} channel and Rac1 GTPase (Zhang *et al.*, 2009). K_{ATP} channel K_{ir}6.2 knockout mouse causes a similar phenotype of depolarized resting potential, loss of Ca transients, and massive decrease in GSIS (Miki *et al.*, 1998). In those models, VGCCs are largely intact and depolarization-induced Ca²⁺ influx caused elevation of resting [Ca²⁺]_i levels which might deplete the ready-to-release pool. However, because in our case VGCCs are downregulated, the resting [Ca²⁺]_i level is intact or lower than the control (Fig. 4G), although the membrane potential could be depolarized by glucose stimulation in the KIF5B-deficient cells (Fig. 5G). This was why we focused on Ca_v1.2 as the main target of cell biology in this paper which would be at least one of the important components of the KIF5B phenotype, but of course we

would admit this is not the sole target because KIF5B is involved in very fundamental ER chaperone exchange program that underlies maturation of multiple membrane proteins (Fig. 6B).

Does higher resting membrane potential of cKO beta cells contribute to the defect in GSIS?

We understand that this difficult question is related with a long-stated question regarding the pathogenesis of glucotoxicity (Khaldi *et al*, 2004; Pertusa *et al*, 2002; Roche *et al*, 1998), where higher resting potential and GSIS defects are cooccurring. Thus, the answer should be partly 'Yes,' due to a potential "cellular fatigue" with some elusive mechanisms, especially in the first phase.

However, in our specific case in the second phase, the Ca resting level was not very high so that it is not likely to give a significant cellular fatigue (Fig. 4G). In addition, we clearly showed that loss of Ca transients was the reason of the loss of F-actin reorganization and the loss of full-fusion exocytosis (Fig. 2), which could be fully rescued by pharmacological F-actin reorganization. Thus, from those live imaging data, we could argue that the second phase GSIS impairment is especially due to the loss of Ca transients.

We will explicitly add these discussions according to the insightful comments of this reviewer.

3, "Consequently, spontaneous action potentials were increased in the resting state, but the size of glucose stimulated depolarization was decreased....." Any evidence?

We will tone down the description for avoiding overinterpretation.

4, In Figure 5B, the expression level of CaV2.3 (which also participates in GSIS) was also reduced in cKO beta cells. Does CaV2.3 also contribute to the function of KIF5B in regulating GSIS?

Logically, yes, but its contribution must be partly. According to previous genetic evidence, knockout of Ca_v2.3 mainly affected the late-phase insulin secretion (after 30 min) as the following panel (Jing *et al*, 2005). We'll explicitly include this discussion according to this insightful suggestion.

Figure for referees not shown.

Does KIF5B also regulate the membrane trafficking of CaV2.3?

Yes. We will add the degradation assay data of Ca_v2.3 that are also affected by KIF5B depletion.

5, "We compared the chaperone binding capacities of CaV1.2 and Kir6.2 by immunoprecipitation among the KIF5B-KD system in the presence of the proteasome inhibitor MG132 (Fig. 6B and C)." The quality of Figure 6B is low.

We will improve the quality of Figure 6B with adding the individual values of the original assay.

6, In Figure 6 and Figure 7, the authors used an overexpression system to demonstrate the interaction between KIF5B and HSP90 (or HSP70). However, the system is artificial and may not reflect the molecular interaction in the native states.

We are afraid to argue that the interaction between KIF5B and Hsp70/90 itself has been looked at a non-overexpression system. Figure 6A is immunofluorescence microscopy of primary beta cells. The IP study in Figure 6B/C is performed in knockdown/control MIN6 cells, where neither KIF5B nor HSPs are overexpressed. The proximity ligation assay in Figure 6D/E is also performed in primary beta cells. The data afterwards were inevitably provided by fluorescence-protein-tagged systems for the sake of live imaging, which enhances the nature of these unique non-membranous organelles containing KIF5B/Hsp90 on the cell bottom that might involve liquid-liquid phase separation. In addition, we will add some IHC results in intrinsic systems for the clarity of the latter half data according to these kind suggestions of this reviewer.

7, There is no convincing evidence showing that CaV1.2 is the major downstream effector of KIF5B in regulating GSIS. Can activators of CaV1.2 rescue the phenotype of KIF5B cKO beta cells?

Because the insulin secretion pattern in Ca_v1.2 knockout mice well phenocopies cKO of KIF5B (Schulla *et al.*, 2003), it is largely explainable. We will look at GSIS levels from Bay K 8644-treated cKO islets as kindly suggested.

Referee #2:

Tanaka *et al.* report experiments that are interpreted to indicate that the moto protein KIF5B affects voltage gated calcium channels (CaV1.2) and the folding of these channels facilitated by the chaperones HSP70 and HSP90. Although the data suggest a role for KIF5B in affecting CaV1.2 function, the data supporting the involvement and effects of the HSP chaperones is very indirect, making the role of these potential interactions in CaV1.2 unclear. Futher, there

are significant problems with the data presented in support of the studies of HSP interactions. There are serious problems with the western blot data presented in Fig. 6B and C. Each of the gel strips in the right hand side of the figure have the same systematic diagonal clipping (most evident in the strip for HSP90, but present in all strips). Consequently, the strips are missing complete bands for all of the proteins probed in the CaV1.2 samples and both Kir6.2 samples. Hence, none of the values presented in Fig. 6C can be considered accurate.

Yes, we will improve the data quality of western blotting in Figures 6B and C.

Besides this technical problem, there is no convincing evidence that the association of CaV1.2 with Hsp90 or Hsp70 is leading to properly folded channels. Indeed, the amount of HSP70 associated with the channel (which could indicate channels that are misfolded) appears to be the same in the SC and K5BKD2 panels (with the caveat above about the gels).

We are afraid to say that this is not the case. Because Hsp70 facilitates both folding and holding (=degradation) of ER clients in the first half of the ER folding steps, it is not likely that the amount of Hsp70 associated with the channel indicates channels that are misfolded. The channels that undergo correct folding also binds to Hsp70 as depicted on the left side of the following scheme (Fig. 7F):

Figure for referees not shown.

Rather than that, the amount of Hsp90-associated Ca_v1.2 can be interpreted as the one successfully cleared the ERQC steps to proceed the full folding steps. Because this parameter is significantly reduced in the cKO cells, our working hypothesis is considered well experimentally evidenced.

It should also be noted that the 'SC' control cells are not well described in the text, making it difficult to understand the nature of this control.

SC means cells transfected with scrambled control miRNA which does not affect the expression level of KIF5B at all, which as the negative control excluding the possible miRNA-transfection artifacts. We will describe this more explicitly.

Additionally, the data suggesting knockdown of CaV1.2 current density (Fig. 4H) should be validated using standard blockers of CaV1.2 to show whether the measured currents arise from CaV1.2 or some other class of calcium channels.

Yes, we will present electrophysiology data with the inhibitors as a negative control.

Referee #3:

1. Does this manuscript report a single key finding? YES. The manuscript by Tanaka et. al., reports that the kinesin superfamily protein KIF5B regulates actin-cytoskeleton reorganization during the second phase of GSIS via acting as a scaffolding factor for HSP70-HSP90 exchange for Cav1.2 in ER tangles, which is an essential step for Cav1.2 ER exit and Cav1.2 regulated Ca²⁺ mediated GSIS.
2. Is the reported work of significance (YES), or does it describe a confirmatory finding or one that has already been documented using other methods or in other organisms, etc (NO)? YES.
3. Is it of general interest to the molecular biology community? NO. In my opinion islet biology-diabetes research community will be more interested in this manuscript.
4. Is the single major finding robustly documented using independent lines of experimental evidence (YES), or is it really just a preliminary report requiring significant further data to become convincing, and thus more suited to a longer format article (NO)? YES, but need additional experiments as documented below in the comments to the author section.
Overview and overall evaluation: The manuscript by Tanaka et. al., using in vivo (KIF5B conditional knockout mouse model), and in vitro (KIF5B conditional knockout primary mouse islets and clonal beta-cells) approaches determined the mechanistic contribution of kinesin superfamily protein KIF5B in the regulation of second phase GSIS. Essentially, the authors showed that KIF5B regulates actin-cytoskeleton reorganization during the second phase of GSIS via acting as a scaffolding factor for HSP70-HSP90 exchange for Cav1.2 in ER tangles, which is an essential step for Cav1.2 ER exit and Cav1.2 regulated Ca²⁺ mediated GSIS. I found the paper to be novel and interesting and has the great potential to provide important knowledge relevant to the field of beta-cell function and diabetes. Also, the findings are appropriately discussed in the context of earlier literature and the manuscript is well written. However, I have certain concerns that need to be addressed which I detailed below. If appropriately revised I recommend the manuscript for publication in EMBO Reports.

Thank you very much for these enthusiastic supports of this manuscript. We will do our best to revise it according to this reviewer's kind and constructive suggestions.

Comments to the author:

1. Please describe what the authors are referring to by "constitutive" and "Regulatory"

insulin secretion in Fig. 1K-L.

Figure for referees not shown.

In this perfusion insulin secretion graph, we define “constitutive” as the mean amounts of [1-1] in CT and [1-2] in cKO islets, but we will change it “basal” for the clarity. “Regulatory” is defined as [the amount of insulin secretion from the stimulated islets] minus [basal secretion]. We will explicitly add these bars in the perfusion figure. Thank you very much for this comment.

2. Fig. 1K shows a marked reduction in first-phase GSIS in cKO islets, but the authors are discussing primarily defects in 2nd phase GSIS in the cKO islets. Please comment in the discussion section on the probable mechanism of KIF5B regulated 1st phase GSIS.

Yes, thank you very much for reminding us of this point. We should have described our interpretation why KIF5B-deficiency resulted in the decrease of the first phase secretion. Shigeto et al. mentioned “first phase insulin secretion from MIN6 cells consists of at least three components: one dependent on influx of L-type voltage-dependent calcium channel (component 1), one involving calcium release from endoplasmic reticulum via ryanodine receptors (component 2), and the final one that somehow involves voltage-gated Na⁺ channels (component 3). Our experiment demonstrated that components 2 and 3 are independent of an increase of Ca²⁺ entry into the cells since these components were not affected in the condition of extracellular calcium depletion.” (Shigeto *et al.*, 2006). As components 1 and 2 should result in intracellular Ca transients, it can be well interpreted that the significant loss of Ca transients in KIF5B-deficient beta cells (Fig. 4G) are the primary cause of significant loss of 1st-phase insulin secretion. The remaining component of phase 1 secretion may be due to component 3 or other unknown factors.

3. In Pg. 12, 2nd paragraph: Please also discuss the results of cKO primary islets (Fig. 4D-F) for clarity.

Yes, we will add this discussion as kindly suggested.

4. **Page 13: 2nd paragraph, starting with "To investigate the mechanistic link between KIF5B and Ca dynamics,....": Please provide the western blot data for Cav 2.3, BKca, Syntaxin-1, Na/Kappa ATPase, HSP70, HSP90, calnexin, LC3, STIP1, and Derlin.**

Yes, we will provide the western blotting.

5. **In Fig 6B, the IP blot: the portion of the IP blots are damaged (broken top part) near the bands of interest. So, I am unable to review the findings relevant to Fig 6B properly. Please provide undamaged sets of the IP blots for correct reviewing.**

Yes, we will improve the Fig. 6B data.

6. **Page 15: "Although CaV1.2 and Hsp70 expression levels were both decreased in KD cells" The reduced level of HSP70 in input is not clear**

We will remove this expression.

7. **According to Fig 5J, reduction in Cav1.2 level in cKO cells can be restored by MG132 treatment. In Fig 6B, the cells are treated with MG132, but still, in KIF5B KO cells, the Cav1.2 level is markedly reduced. Please explain this discrepancy.**

Yes, MG132 increased the level of CaV1.2 expression of cKO cells but still not sufficient to reach the CT level. We will explicitly add this discussion as kindly suggested.

8. **Fig 6F: what type of cells? Control or cKO?**

These are rescued cKO cells.

9. **In Discussion: "We then detected a significant defect in CaV1.2 expression in the posttranslational level (Fig. 5)": In the result section, the authors discussed post-transcriptional changes.**

Posttranscriptional is correct.

10. **In Fig 6 (Immunoprecipitation), the authors treated MIN6 cells with 10mM MG132 for 6 days. Please provide the evidence for cell viability after this treatment.**

Thank you for pointing out this error in Methods section. The KD vector transfection was for 6 days to completely reduce KIF5B protein, but MG132 was added only for 6-7 h at the final stage so the cell viability was intact. We will revise the text as kindly pointed out.

11. **Fig. 6D and E: Please provide information about the type of cell treatments representing red and Blue dots (Fig. 6D) and bars (Fig. 6 E).**

The color coding along the z-axis in Fig. 6D was performed using ImageJ software and the error bars are mean \pm SEM in Fig. 6E. We will more explicitly describe those in the legends and methods section.

12. Co-immunoprecipitation studies (similar to Fig. 6B) using control (wild type) cells and KIF5B overexpressed cells would be helpful for a clear understanding of the proposed mechanism.

We will try to examine if KIF5B overexpression enhances the binding between Cav1.2 and the Hsp90/calnexin/derlin/STIP1 chaperone-exchange complex.

References:

- Dahiya V, Rutz DA, Moessmer P, Muhlhofer M, Lawatscheck J, Rief M, Buchner J (2022) The switch from client holding to folding in the Hsp70/Hsp90 chaperone machineries is regulated by a direct interplay between co-chaperones. *Molecular cell* 82: 1543-1556 e1546
- Genest O, Wickner S, Doyle SM (2019) Hsp90 and Hsp70 chaperones: Collaborators in protein remodeling. *The Journal of biological chemistry* 294: 2109-2120
- Jing X, Li DQ, Olofsson CS, Salehi A, Surve VV, Caballero J, Ivarsson R, Lundquist I, Pereverzev A, Schneider T *et al* (2005) CaV2.3 calcium channels control second-phase insulin release. *The Journal of clinical investigation* 115: 146-154
- Khaldi MZ, Guiot Y, Gilon P, Henquin JC, Jonas JC (2004) Increased glucose sensitivity of both triggering and amplifying pathways of insulin secretion in rat islets cultured for 1 wk in high glucose. *Am J Physiol Endocrinol Metab* 287: E207-217
- Miki T, Nagashima K, Tashiro F, Kotake K, Yoshitomi H, Tamamoto A, Gono T, Iwanaga T, Miyazaki J, Seino S (1998) Defective insulin secretion and enhanced insulin action in KATP channel-deficient mice. *Proceedings of the National Academy of Sciences of the United States of America* 95: 10402-10406
- Moran Luengo T, Kityk R, Mayer MP, Rudiger SGD (2018) Hsp90 Breaks the Deadlock of the Hsp70 Chaperone System. *Molecular cell* 70: 545-552 e549
- Moran Luengo T, Mayer MP, Rudiger SGD (2019) The Hsp70-Hsp90 Chaperone Cascade in Protein Folding. *Trends in cell biology* 29: 164-177
- Pertusa JA, Neshler R, Kaiser N, Cerasi E, Henquin JC, Jonas JC (2002) Increased glucose sensitivity of stimulus-secretion coupling in islets from *Psammomys obesus* after diet induction of diabetes. *Diabetes* 51: 2552-2560
- Roche E, Farfari S, Witters LA, Assimacopoulos-Jeannet F, Thumelin S, Brun T, Corkey BE, Saha AK, Prentki M (1998) Long-term exposure of beta-INS cells to high glucose concentrations increases anaplerosis, lipogenesis, and lipogenic gene expression. *Diabetes* 47: 1086-1094
- Rutledge BS, Choy WY, Duennwald ML (2022) Folding or holding?-Hsp70 and Hsp90 chaperoning of misfolded proteins in neurodegenerative disease. *The Journal of biological chemistry* 298: 101905
- Schulla V, Renstrom E, Feil R, Feil S, Franklin I, Gjinovci A, Jing XJ, Laux D, Lundquist I, Magnuson MA *et al* (2003) Impaired insulin secretion and glucose tolerance in beta cell-selective Ca(v)1.2 Ca²⁺ channel null mice. *The EMBO journal* 22: 3844-3854
- Shigeto M, Katsura M, Matsuda M, Ohkuma S, Kaku K (2006) First phase of glucose-stimulated insulin secretion from MIN 6 cells does not always require extracellular calcium influx. *J Pharmacol Sci* 101: 293-302
- Wegele H, Wandinger SK, Schmid AB, Reinstein J, Buchner J (2006) Substrate transfer from the

chaperone Hsp70 to Hsp90. *Journal of molecular biology* 356: 802-811

Zhang J, Luo R, Wu H, Wei S, Han W, Li G (2009) Role of type Ialpha phosphatidylinositol-4-phosphate 5-kinase in insulin secretion, glucose metabolism, and membrane potential in INS-1 beta-cells. *Endocrinology* 150: 2127-2135

Dear Prof. Hirokawa

Thank you for your message asking us to reconsider our decision and invite revision of your manuscript. I am sorry that it took me so long to come back to you with an answer but I have now carefully read your rebuttal, I have re-read the referee reports and your manuscript.

I note that the referees raised a number of concerns, some of which I specifically pointed out in my decision letter. I should also note that referee #2 and #3 rated the technical quality of your study low in the summary evaluation table returned with their reports, providing overall limited support for the publication of your study in EMBO Reports.

That said, I carefully read your point-by-point response, your clarifications of certain concerns and the revisions you outline.

Regarding the causality of CaV1.2 for the GSIS phenotype in KIF5B-deficient animals you mention that the KIF5B KO phenocopies CaV1.2 KO mice, which is indeed indication that CaV1.2 and KIF5B might act in the same pathway in the context of insulin secretion but this has formally not been tested. You now indicate that you plan to test whether application of the L-type calcium channel agonist Bay K 8644 rescues the phenotype of KIF5B cKO beta cells, which, I agree, will strengthen this part.

I also acknowledge that the Hsp70/90 - KIF5B interaction data are based on endogenous proteins, which will address the concern regarding overexpression-based artefacts.

Another referee concern, which was also mentioned in my decision letter, related to the absence of convincing evidence that the association of CaV1.2 with Hsp90 or Hsp70 is leading to correctly folded channels. I appreciate the clarification that the binding of Hsp90 to the Hsp70-channel complex is required to complete channel folding and that thus the reduced binding of Hsp90 indicates impaired folding. That said, the assays used did not specifically investigate the folded state of CaV1.2 in the presence of absence of KIF5B but relied on the binding to Hsp90 or the localization to e.g., ER tangles. I considered it important to provide more direct evidence for the folded state of CaV1.2 by biochemical assays. Alternatively, a more detailed analysis of other co-chaperones that are essential for the handover of Hsp70 clients to Hsp90 could be analysed to further support the conclusion that the Hsp70 to Hsp90 switch is specifically affected. It seems that you proposed experiments along these lines in response to point 12 of referee 3.

Overall, I feel that most of the most critical concerns of the referees can be addressed in the revision you propose but I also think that further evidence for the proposed effect on CaV1.2 folding should be provided, as outlined above.

Should you be able to address all these concerns, I would not oppose to consider a re-submission of your manuscript. I must however stress that the new manuscript would be treated as a new submission and would be re-evaluated at submission stage, in particular in what concerns the novelty of the findings. In case of a positive evaluation, I will send the revised manuscript back to the same referees, thus effectively treating the manuscript as a revision.

Should you decide to embark on this revision and re-submit your manuscript, please refer to our communication in the cover letter.

Kind regards and success with the experiments ahead,

Martina

Point-by-point response to the referees' comments**Tanaka et al.****April 18, 2024****Referee #1:**

In this manuscript, the authors found that beta-cell-specific deletion of a kinesin molecular motor KIF5B in mice resulted in glucose intolerance. Mechanistically, they found that KIF5B and Hsp90 colocalized at ER tangles on the bottom of cells and demonstrated that KIF5B facilitated Hsp70-to-Hsp90 chaperone exchange during the biogenesis of the voltage-gated Ca²⁺ channel CaV1.2, which mediates glucose-stimulated Ca²⁺ transients. The role of KIF5B in regulating glucose-stimulated insulin secretion (GSIS) is well-recognized, but the underlying mechanism remains unknown. This study is therefore potentially interesting as it provides the possible explanation for the function of KIF5B in regulating GSIS. However, several issues have to be clarified before consideration of publication in EMBO reports.

Thank you for appreciating the potential relevance of this manuscript. We have done our best to fully revise it according to this reviewer's kind and insightful suggestions that considerably improved the manuscript. We have changed the expression "ER tangles" into "ER sheets" as widely accepted.

1, Throughout the manuscript, does ca mean Ca²⁺?

Yes, we fixed it so throughout the manuscript.

2, "Very intriguingly, the resting membrane potential of cKO beta cells was significantly depolarized to -40 mV, while that of CT cells was properly maintained at -60 mV (Fig. 4I and J)" Clearly, KIF5B affected the resting membrane potential of beta cells, suggesting that KIF5B has more client proteins besides CaV1.2.

Thank you for agreeing with our argument that KIF5B affected the resting membrane potential of beta cells. As the molecular basis of this effect, we consider that downregulation of Cav1.2 itself, in addition to that of BK_{Ca} channel, Na/K ATPase, and

PIP5K α (**Fig. 5A and B**), are likely to cause the constitutive membrane depolarization. BK_{Ca} channel and Na/K ATPase hyperpolarize the membrane by K⁺ currents (Dufer *et al*, 2009; Fridlyand *et al*, 2013), so that their shortage may result in membrane excitation. The PIP5K α product PIP₂ is known to activate K_{ATP} channel, which may also enhance the hyperpolarization (de la Cruz *et al*, 2016; Liang *et al*, 2014). We identified that both of Cav1.2 and BK_{Ca} are downregulated either by KIF5B knockdown (**Fig. 5A and B**) or STIP1 knockdown (**Fig. 6K**). Furthermore, we identified that KIF5B knockdown could disrupt the STIP1–Hsp90 association (**Fig. 7E**), so KIF5B’s contribution for the integrity of the chaperone exchange machinery should be very fundamental. Because STIP1 is an essential cochaperone for Hsp70-to-Hsp90 chaperone exchange (Donnelly *et al*, 2013; Moran Luengo *et al*, 2018), it is likely that both proteins are clients of KIF5B- and STIP1-mediated Hsp70-to-Hsp90 chaperone exchange, as this reviewer has insightfully suggested.

Figure for referees not shown.

Fig. 5A–C. (A and B) Immunoblotting of a KIF5B-knockdown (KD) system in MIN6 cells using scrambled control (SC) and KD miRNA, for the indicated proteins (A); and its normalized quantification (B). *** $p < 0.001$; **** $p < 0.0001$; Welch’s t test between KD and SC; $n = 3–8$. (C) Quantitative RT–PCR results of the KD system for the indicated genes. $n = 6$.

Figure for referees not shown.

Fig. 6K. Immunoblotting of scramble control (SC) and STIP1-knockdown (KD) MIN6 cells against the indicated epitopes. Note that STIP1 deficiency induced downregulation of Cav1.2 and BK_{Ca} proteins. Reproduced twice.

Figure for referees not shown.

Fig. 7E. Vesicle IP of the MG-132-treated MIN6 cell lysates among the KIF5B KD system against STIP1. Note that the Hsp90 level in the STIP1 immunoprecipitants (IP) was greatly decreased by KIF5B deficiency. Repeated twice.

Alternatively, recent report suggested that Cav1.2 augments BK_{Ca} activity through a direct interaction (Plante *et al*, 2021). As Cav1.2-deficient mouse islets nicely phenocopied the leaky basal insulin secretion as well as GSIS impairments of our KIF5B cKO islets in glucose perfusion experiment as the following figures (Schulla *et al*, 2003), we consider that Cav1.2 decrease itself may also account for the baseline changes in the KIF5B cKO cells.

GSIS from Cav1.2 -deficient mouse pancreatic islets (Schulla *et al.*, 2003)

REDACTED: Figure 4 from Schulla *et al.*, 2003

GSIS from KIF5B-deficient mouse pancreatic islets (our Fig. 1F)

Figure for referees not shown.

We added the following discussion in **p. 24, paragraph 1**, according to this insightful suggestion of this reviewer.

“Our data suggested that KIF5B deficiency augmented the basal insulin secretion as well as constitutive depolarization (Figs. 1F and 4I). This could be mostly explained by the downregulation of Na/K ATPase and BK_{Ca} in KIF5B KD cells (Fig. 5A and B). Alternatively, Cav1.2 knockout mice exhibited a similar elevation of basal insulin secretion (Schulla *et al.*, 2003). Because Cav1.2 augments BK_{Ca} activity through a direct interaction (Plante *et al.*, 2021), the Cav1.2 decrease itself may also account for the baseline changes in the KIF5B cKO cells.”

Does higher resting membrane potential of cKO beta cells contribute to the defect in GSIS?

We assume if this question is related with a long-stated question regarding the

pathogenesis of glucotoxicity (Khaldi *et al*, 2004; Pertusa *et al*, 2002; Roche *et al*, 1998), where higher resting potential and GSIS defects are cooccurring. Thus, the answer should be partly ‘Yes,’ due to a potential “cellular fatigue” with some elusive mechanisms, especially in the first phase.

However, in our specific case in the second phase, the Ca^{2+} resting level was not very high so that it is not likely to give a significant cellular fatigue (**Fig. 4H**). In addition, we clearly showed that loss of Ca^{2+} transients was the reason of the loss of F-actin reorganization and the loss of full-fusion exocytosis according to the pharmacological rescue experiments using ionomycin and cytochalasin B (CyB) (**Figs. 2–4**). In addition, we newly identified that the GSIS in cKO islets could be restored by the L-type VGCC agonist Bay-K 8644 (**Fig. 5F**). From those data, we argue that the second phase GSIS impairment is especially due to the loss of Ca^{2+} transients.

Figure for referees not shown.

Fig. 4H. 20 mM glucose-stimulated calcium transients of the primary beta cells labeled with Fluo4-AM. *** $p < 0.001$, $n = 3$, two-way ANOVA on periods after the stimulation.

Figure for referees not shown.

Figure 2. KIF5B stabilizes insulin exocytosis through actin remodeling

(A) Typical images of the surface of synapto.pHluorin-transduced primary beta cells of the indicated conditions in TIRF microscopy. Double arrows, full fusion. CyB, 10 $\mu\text{g}/\text{mL}$ cytochalasin B treatment. Scale bars, 5 μm . Corresponding to Movie EV1.

(B) Typical time-lapse images of insulin exocytosis on the cell surface of the indicated genotypes by time-lapse total internal reflection fluorescence (TIRF) microscopy, recorded at 30–60 min after glucose stimulation.

(C) Typical traces of fluorescence intensities at the exocytosis points on the plasma membrane of the indicated genotypes.

(D) Histogram of the duration of each exocytosis event at 30–60 min of glucose stimulation.

(E) Quantification of the occurrence of full-fusion events (defined to be longer than 1.2 s in Panel D) and kiss-and-run events (shorter than 1.2 s) in each genotype. Note that KIF5B deficiency significantly reduced the exocytosis duration. ns, $p > 0.05$; ** $p < 0.01$, Welch's t test, $n = 8-9$.

(F and G) Temporal projection of the time-lapse images of CT and cKO primary beta cells (F) and its quantification for the long directional motility (G). Scale bars, 5 μm . **** $p < 0.0001$, Welch's t test, $n = 31-46$. Double arrow, full fusion. Single arrows, cortical long-range directional motilities.

(H–K) Time course of pharmacological rescue of insulin granule dynamics of cKO cells by CyB treatment, represented by an experimental design (H); temporal projection of time-lapse images (I); time course of the occurrence of peripheral directional motility after CyB treatment (J), and time course of the occurrence of indicated types of exocytosis after CyB treatment (K). Scale bars, 5 μm . * $p < 0.05$, ** $p < 0.01$, compared with the data at 1–10 min, Welch's t test, $n = 4-8$ (J and K).

Figure for referees not shown.

Figure 3. KIF5B facilitates short directional motility of insulin granules

(A and B) Images of cortical insulin granules of control (CT) and cKO primary beta cells, transfected with a *phogrin-Dronpa-Green1* expression vector, starved, and stimulated by 20 mM glucose for 30 min (A), accompanied by quantification of cortical insulin granules (B). Bar, 2 μ m. ns, $p > 0.05$, $n = 3-5$, Welch's t test.

(C–H) Quantification of insulin granule motility tagged by phogrin-EGFP with a 5LIVE-Duo confocal microscope, in primary beta cells of the indicated genotypes and treatments for 100 s, at 30 min after glucose stimulation, represented by particle tracks (C), MSD trajectories (D), MSD curves of all tracks (E), percentage of >2 SD tracks (F), MSD curves of >2 SD tracks (G), and α analysis of >2 SD tracks for directional movements (H); corresponding to Movie EV2. Scale bar, 2 μm . Color coding in C and D, the time sequence. Note that the cKO beta cell granules were significantly less motile than the CT beta cell granules, which was significantly reversed by the CyB treatments.

Figure for referees not shown.

Figure 4. KIF5B is essential for Ca^{2+} transients of beta cells and actin remodeling
(A–D) Actin remodeling assay using fluorescent phalloidin-labeled CT and cKO primary beta cells with the indicated treatments (A and C), accompanied by quantification (B and D). Scale bar, 5 μm . ns, $p > 0.05$, *** $p < 0.001$; Welch's t test, $n = 10\text{--}13$. Quantified at the positions of largest diameter. Arrows, the sites of cortical F-actin remodeling. Corresponding to Fig. EV2A–D and Movie EV3.

Figure for referees not shown.

Fig. 5F. Rescue of impaired GSIS from cKO mouse islets by 1 μ M Bay-K 8644. **** $p < 0.01$, one-way ANOVA, n = 8–9.**

Accordingly, we explicitly added the following discussion in p. 15, ll. 3–9.

“These data suggested that pharmacological activation of the reduced level of Cav1.2 in cKO islets could still restore the GSIS. Thus, we considered that cKO mouse islets may preserve the capacity for GSIS if proper Ca^{2+} transients were formed, and that the reduced expression of Cav1.2 in cKO islets could be a major cause of the GSIS failure. The negative effects of ‘cellular fatigue’ through constitutive depolarization (Khaldi *et al.*, 2004; Pertusa *et al.*, 2002; Roche *et al.*, 1998) was thus unlikely especially in the case of this second phase GSIS.”

3, "Consequently, spontaneous action potentials were increased in the resting state, but the size of glucose stimulated depolarization was decreased....." Any evidence?

We have removed the latter half for avoiding overinterpretation. It now reads as **"Consequently, spontaneous action potentials tended to increase in the resting state in cKO beta cells."** (p. 13, ll. 10–11).

4, In Figure 5B, the expression level of CaV2.3 (which also participates in GSIS) was also reduced in cKO beta cells. Does CaV2.3 also contribute to the function of KIF5B in regulating GSIS?

Logically, yes, but its contribution must be limited. According to previous genetic evidence, knockout of Cav2.3 mainly affected the late-phase insulin secretion (after 30 min) as the following panel (Jing *et al.*, 2005).

REDACTED: Figure 4B from Jing *et al.*, 2005

Cf. Perfusion of Cav2.3 KO islets (from Jing *et al.*, 2005)

Accordingly, we explicitly included this discussion according to this suggestion as the following (p. 22, ll. 13–18):

"Because our beta-cell specific KIF5B cKO mice largely phenocopied the Cav1.2 cKO mice (Schulla *et al.*, 2003), rather than Cav2.3 KO mice (Jing *et al.*, 2005), and that the Cav1.2 agonist Bay-K 8644 nicely restored the GSIS in KIF5B cKO islets (Fig. 5F), the Cav1.2 channel was considered as the most physiologically relevant client of the KIF5B–Hsp machinery in pancreatic beta cells."

Does KIF5B also regulate the membrane trafficking of CaV2.3?

Yes. We added [1] TIRF microscopy of Cav2.3 (**Fig. 5D**), [2] a degradation assay data of Cav2.3 (**Fig. 6B and C**) and [3] TIRF/STORM microscopy of for surface expression of Cav2.3 (**Fig. 6G**), which were also affected by KIF5B depletion, according to this insightful question.

Figure for referees not shown.

Fig. 5D and E. Immunofluorescence microscopy of primary beta cells of the indicated genotypes, against Cav1.2, Cav2.3, and Kir6.2, using TIRF microscopy (D), and its quantification (E). Bar, 5 μ m. ns, $p > 0.05$; ** $p < 0.01$; **** $p < 0.0001$; n = 11–21, two-way ANOVA.

Figure for referees not shown.

Fig. 6B and C. Degradation assay. Immunofluorescence against Cav2.3 (B; the left panel) of primary beta cells of the indicated genotypes after the CHX treatment of the indicated periods; accompanied by their quantification (C; the right panel). Bars, 5 μ m. *** $p < 0.001$, Welch's t test at the indicated time; 17–24 (C).

Figure for referees not shown.

Fig. 6G. TIRF/STORM microscopy of primary mouse beta cells of the indicated genotypes against Cav1.2 (green) and Cav2.3 (magenta). Scale bars, 5 μm .

5, "We compared the chaperone binding capacities of Cav1.2 and Kir6.2 by immunoprecipitation among the KIF5B-KD system in the presence of the proteasome inhibitor MG132 (Fig. 6B and C)." The quality of Figure 6B is low.

We have significantly improved the quality of these figures (Fig. 7C and D). We especially apologize for the systematic diagonal clipping in the right hand of the figure on each of the gel strips according to a technical accident due to Adobe Illustrator software.

Figure for referees not shown.

Fig. 7C and D. Vesicle IP of MIN6 cell lysates transduced with scrambled control (SC) and KIF5B-knockdown (KD) miRNAs in the presence of the proteasome inhibitor MG-132 (MG), precipitated using Cav1.2 or Kir6.2 antibodies or normal rabbit IgG (NRG) and immunoblotted for the indicated proteins (C), accompanied by quantification of Cav1.2-coprecipitated fractions (D). Note that Cav1.2-binding capacities of derlin1, calnexin-1, and Hsp90 chaperones and that of the adaptor protein STIP1 in KD cell lysates were significantly lower than those in SC cell lysates.

6, In Figure 6 and Figure 7, the authors used an overexpression system to demonstrate the interaction between KIF5B and HSP90 (or HSP70). However, the system is artificial and may not reflect the molecular interaction in the native states.

We are afraid to argue that the interaction between KIF5B and Hsp70/90 itself has been looked at a non-overexpression system. The IP study in **Figure 7C and D** (please see the previous page) is performed in knockdown/control MIN6 cells, where neither KIF5B nor Hsp proteins were overexpressed.

The proximity ligation assay in **Figure 6I and J** is also performed in non-transfected primary beta cells.

Figure for referees not shown.

Fig. 6I and J. z-projection of proximity ligation assay in CT and cKO primary beta cells showing the protein binding between Cav1.2 and the indicated Hsp proteins (I); accompanied by quantification (J). ** $p < 0.01$; *** $p < 0.001$; Welch's t test, $n = 6$.

The data afterwards were inevitably provided by fluorescence-protein-tagged systems for the sake of live imaging, which enhances the nature of these unique non-membranous organelles containing KIF5B/Hsp90 on the cell bottom that might involve liquid-liquid phase separation.

Nevertheless, we added native immunocytochemistry results in wild type primary beta cells for the clarity of the latter half data according to these kind suggestions of this reviewer (**Fig. 8D**).

Figure for referees not shown.

Fig. 8D. Immunocytochemistry of a wild type islet cell against KIF5B and Hsp90 using Airyscan microscopy. Reproduced twice.

Thank you so much for this insightful suggestion that greatly enhanced the quality of the data in intrinsic conditions.

7, There is no convincing evidence showing that CaV1.2 is the major downstream effector of KIF5B in regulating GSIS. Can activators of CaV1.2 rescue the phenotype of KIF5B cKO beta cells?

First, as mentioned earlier in the answer to the point 1, we added a GSIS assay using the L-type VGCC (Ca_v1.2) agonist, Bay-K 8644. This agonist could very nicely restore the GSIS of KIF5B cKO mouse islets (**Fig. 5F**), suggesting that decreased Ca_v1.2 expression in KIF5B cKO beta cells is most relevant process to explain the aberrant insulin secretion.

Figure for referees not shown.

Fig. 5F. Rescue of impaired GSIS from cKO mouse islets by 1 μM Bay-K 8644. ***p* < 0.01, one-way ANOVA, n = 8–9.

Second, as also mentioned above, the insulin secretion pattern in Ca_v1.2 knockout mouse islets (Schulla *et al.*, 2003) well phenocopied that in KIF5B cKO mouse islets (Fig. 1F). Actually, Ca_v1.2 knockout mouse islets very nicely phenocopied our KIF5B cKO mouse islets on the following three points: (1) elevated basal insulin release in the resting state, (2) significant decrease in 1st-phase GSIS, and (3) almost complete abolishment of 2nd-phase GSIS. As mentioned in the point 1, we consider that (1) can be explained by the fact that Ca_v1.2 augments BK_{Ca} activity through a direct interaction (Plante *et al.*, 2021).

GSIS from Cav1.2 -deficient mouse pancreatic islets (Schulla *et al.*, 2003)

REDACTED: Figure 4 from Schulla *et al.*, 2003

GSIS from KIF5B-deficient mouse pancreatic islets (our Fig. 1F)

Figure for referees not shown.

Thus, we believe that the contribution of Cav1.2 in KIF5B pathway is very significant. We deeply thank this reviewer for insightful comments that considerably improved the quality of this manuscript both in conceptual and technical manners.

Referee #2:

Tanaka et al. report experiments that are interpreted to indicate that the motor protein KIF5B affects voltage gated calcium channels (CaV1.2) and the folding of these channels facilitated by the chaperones HSP70 and HSP90. Although the data suggest a role for KIF5B in affecting CaV1.2 function, the data supporting the involvement and effects of the HSP chaperones is very indirect, making the role of these potential interactions in CaV1.2 unclear. Further, there are significant problems with the data presented in support of the studies of HSP interactions.

There are serious problems with the western blot data presented in Fig. 6B and C. Each of the gel strips in the right hand side of the figure have the same systematic diagonal clipping (most evident in the strip for HSP90, but present in all strips). Consequently, the strips are missing complete bands for all of the proteins probed in the CaV1.2 samples and both Kir6.2 samples. Hence, none of the values presented in Fig. 6C can be considered accurate.

Yes, we would apologize for the systemic technical errors in data processing in previous Fig. 6B and C. We have thoroughly improved the data in new **Fig. 7C and D**.

Figure for referees not shown.

Fig. 7C and D. Vesicle IP of MIN6 cell lysates transduced with scrambled control (SC) and KIF5B-knockdown (KD) miRNAs in the presence of the proteasome inhibitor MG-132 (MG), precipitated using Cav1.2 or Kir6.2 antibodies or normal rabbit IgG (NRG)

and immunoblotted for the indicated proteins (C), accompanied by quantification of Cav1.2-coprecipitated fractions (D). Note that Cav1.2-binding capacities of derlin1, calnexin-1, and Hsp90 chaperones and that of the adaptor protein STIP1 in KD cell lysates were significantly lower than those in SC cell lysates.

Besides this technical problem, there is no convincing evidence that the association of CaV1.2 with Hsp90 or Hsp70 is leading to properly folded channels. Indeed, the amount of HSP70 associated with the channel (which could indicate channels that are misfolded) appears to be the same in the SC and K5BKD2 panels (with the caveat above about the gels).

We are afraid to say that this is not the case. Because Hsp70 facilitates both folding and degradation of ER clients in the first half of the ER folding steps, the amount of Hsp70 associated with the channel may include both fates of proteins. Thus, the Hsp70-bound fraction cannot solely indicate the amount of channels only that are misfolded. The channels that undergo correct folding also binds to Hsp70 as depicted on the CT cells in the following scheme (**Fig. 7F**):

Figure for referees not shown.

Fig. 7F. Schematic representation of the working hypothesis on differential KIF5B- and heat-shock-protein (Hsp)-dependencies of opposing ER clients Cav1.2 and Kir6.2 in control (CT) and KIF5B conditional knockout (cKO) mouse beta cells. In cKO cells, Cav1.2 fails in chaperone exchange to undergo ERAD-mediated degradation, but Kir6.2 is intact because it is independent on the KIF5B–Hsp machinery.

Rather than that, the amount of Hsp90-associated Cav1.2 can be interpreted as the one successfully cleared the ERQC step to proceed the full folding steps. Because this parameter was significantly reduced in the cKO cells, our working hypothesis is considered well experimentally evidenced.

It should also be noted that the 'SC' control cells are not well described in the text, making it difficult to understand the nature of this control.

SC means the cells being transfected with a negative control miRNA for the RNA interference, which was provided by the *Block-iT Pol II miR RNAi* Expression Vector Kit (Thermo Fisher), whose sequence is designed to be totally unrelated with the mouse/human exome sequences. Thus, SC miRNA is applied as a negative control to exclude any miRNA-transfection artifacts. We explicitly described this issue as the following according to this insightful question (p. 14, ll. 5-8, and p. 35, l. 5–p. 36, l. 10).

“To investigate the molecular mechanism between KIF5B and Ca²⁺ transients, we conducted gene silencing by expressing *Kif5b*-antisense knockdown (KD) or scrambled sequence control (SC) miRNAs using mammalian expression vectors in MIN6 insulinoma cells (Fig. 5A and B).” (p. 14)

Gene silencing

For gene silencing, antisense miRNAs were transcribed from mammalian PolIII-based expression vectors derived from a Block-iT Pol II miR RNAi Expression vector kit (Thermo Fisher). For *Kif5b* knockdown, the oligonucleotides 5'-TGCTGTTTCAGGGCTGACTCCAAAGCGTTTTGGCCACTGACTGACGCTT TGGACAGCCCTGAAA-3' and 5'-CCTGTTTCAGGGCTGTCCAAAGCGTCAGTCAGTGGCCAAAACGCTTTG GAGTCAGCCCTGAAAC-3' (KD1); and 5'-TGCTGTTCTCTGTGACTCTGGATCTGGTTTTGGCCACTGACTGACCAGATCCAGTCACAGAGAA-3' and 5'-CCTGTTCTCTGTGACTGGATCTGGTCAGTCAGTGGCCAAAACCAGATCC AGAGTCACAGAGAAC-3' (KD2) were obtained from Invitrogen, annealed, and inserted into the provided vector with *tagRFP* cDNA as an expression marker and equally subjected to the knockdown analyses to obtain consistent results. For *Stip1* knockdown, the oligonucleotides 5'-CCTGTGCAGCCGTCCATAGGGCCTGTCAGTCAGTGGCCAAAACAGGCC TATGAGGACGGCTGCAC-3' and 5'-TGCTGTGCAGCCGTCCATAGGCCTGTTTTGGCCACTGACTGACAGG CCTATGGACGGCTGCA-3' (#Mmi520981, Invitrogen) were used. As a negative control, a scrambled sequence provided by the kit were used to exclude the possibility of nonspecific adverse effects of miRNA expression. Those miRNA expression vectors were introduced into adenoviral vectors using a ViraPower Adenoviral Expression System, purified by CsCl centrifugation, and subjected to cells according to manufacturer's protocols.” (pp. 35–36)

Additionally, the data suggesting knockdown of CaV1.2 current density (Fig. 4H) should be validated using standard blockers of CaV1.2 to show whether the measured currents arise from CaV1.2 or some other class of calcium channels.

Yes, according to this insightful suggestion, we added electrophysiology traces of the voltage-gated Ca²⁺ currents with or without the VGCC inhibitor cocktail (**Fig. 4K**). In those data, the voltage-gated inward currents were almost entirely abolished by the VGCC inhibitors. These data provide direct evidence for that the observed voltage-gated inward currents were truly based on VGCCs on this experimental condition.

Figure for referees not shown.

Fig. 4K. Traces of voltage-gated Ca²⁺ currents in patch-clamp recording of mouse primary beta cells of the indicated genotypes with or without VGCC inhibitors, at the range of -20 to +20 mV. Note that the inhibitor treatment significantly abolished the inward currents.

We deeply thank this reviewer for very insightful comments that significantly improved this paper.

Referee #3:

1. Does this manuscript report a single key finding? YES. The manuscript by Tanaka et. al., reports that the kinesin superfamily protein KIF5B regulates actin-cytoskeleton reorganization during the second phase of GSIS via acting as a scaffolding factor for HSP70-HSP90 exchange for Cav1.2 in ER tangles, which is an essential step for Cav1.2 ER exit and Cav1.2 regulated Ca²⁺ mediated GSIS.
2. Is the reported work of significance (YES), or does it describe a confirmatory finding or one that has already been documented using other methods or in other organisms, etc (NO)? YES.
3. Is it of general interest to the molecular biology community? NO. In my opinion islet biology-diabetes research community will be more interested in this manuscript.
4. Is the single major finding robustly documented using independent lines of experimental evidence (YES), or is it really just a preliminary report requiring significant further data to become convincing, and thus more suited to a longer format article (NO)? YES, but need additional experiments as documented below in the comments to the author section. Overview and overall evaluation: The manuscript by Tanaka et. al., using in vivo (KIF5B conditional knockout mouse model), and in vitro (KIF5B conditional knockout primary mouse islets and clonal beta-cells) approaches determined the mechanistic contribution of kinesin superfamily protein KIF5B in the regulation of second phase GSIS. Essentially, the authors showed that KIF5B regulates actin-cytoskeleton reorganization during the second phase of GSIS via acting as a scaffolding factor for HSP70-HSP90 exchange for Cav1.2 in ER tangles, which is an essential step for Cav1.2 ER exit and Cav1.2 regulated Ca²⁺ mediated GSIS. I found the paper to be novel and interesting and has the great potential to provide important knowledge relevant to the field of beta-cell function and diabetes. Also, the findings are appropriately discussed in the context of earlier literature and the manuscript is well written. However, I have certain concerns that need to be addressed which I detailed below. If appropriately revised I recommend the manuscript for publication in EMBO Reports.

Thank you very much for these enthusiastic supports of this manuscript. We have done our best to revise it according to this reviewer's kind and constructive suggestions. We have changed the expression "ER tangles" into "ER sheets" as widely accepted.

Comments to the author:

1. Please describe what the authors are referring to by "constitutive" and "Regulatory" insulin secretion in Fig. 1K-L.

Thank you for this insightful suggestion. We have considerably improved the analyses of measurements in **Fig. 1F and G**, to depict [1] 'Basal' fraction that is the average value of secreted insulin during non-stimulated periods, [2] 'Increment in 0-10 min' fraction that is the average value of all secreted insulin minus that of the 'Basal' fraction in each experiment during the first 10 min of glucose stimulation that might be largely corresponding to the first-phase GSIS, and [3] 'Increment in 10-44 min' fraction as well that might be largely corresponding to the second-phase GSIS. Accordingly, we can now explicitly present that elevated basal insulin secretion, decreased first-phase GSIS, and significantly decreased second-phase GSIS in cKO islets.

Figure for referees not shown.

Figure 1F and G. Perifusion assay of CT and cKO mouse pancreatic islets stimulated with 20 mM glucose for 45 min (F), quantified for the respective amounts of basal secretion (plotted in an inverted manner) and the first- (0–10 min) and second-phase (10–44 min) of GSIS increments. *** $p < 0.001$; Welch's t test, $n = 12$.

2. Fig. 1K shows a marked reduction in first-phase GSIS in cKO islets, but the authors are discussing primarily defects in 2nd phase GSIS in the cKO islets. Please comment in the discussion section on the probable mechanism of KIF5B regulated 1st phase GSIS.

Yes, thank you very much for reminding us of this point. We should have described our interpretation why KIF5B-deficiency resulted in the decrease of the first phase secretion. We are now starting from the KIF5B phenotype and Ca²⁺ involvement in insulin release in the very first paragraph of the Discussion according to this insightful suggestion (p. 23, ll. 7–13).

“Cav1.2 enterprises glucose-stimulated Ca²⁺ transients that controls first- and second-phase GSIS, partly through Ca²⁺-dependent SNARE complex formation (Trexler & Taraska, 2017) and partly through cortical F-actin remodeling that modulates insulin granule dynamics (Kalwat & Thurmond, 2013). This finding uncovers an outstanding molecular mechanism in the stimulation-secretion coupling of GSIS that acts against diabetes, as well as the relevance of kinesin-1 molecular motor in protein folding in the ER.”

3. In Pg. 12, 2nd paragraph: Please also discuss the results of cKO primary islets (Fig. 4D-F) for clarity.

Yes, upon thoroughly reading the references, we noticed that the pSFK–Cdc42–Rac1 pathway and K_{ATP} –VGCC– Ca^{2+} pathway may mutually crosstalk to enterprize the GSIS, as we summarized in the additional scheme in **Fig. 5I** and discussed as the following way. Thank you very much for this insightful suggestion.

Figure for referees not shown.

Fig. 5I. Schematic representation of possible changes in GSIS signaling in KIF5B cKO beta cells (Arrows). Checkmarks, normal expression. Red arrows, changes according to KIF5B deficiency. Cav1.2 and PIP5K α protein downregulation (underlined) in KIF5B-deficient beta cells may result in the abolishment of Ca^{2+} transients and downregulation of PIP₂, respectively. White ovals, pharmacological reagents that directly stimulated the respective pathways: Bay-K, Bay-K 8644; Iono, ionomycin; CyB, cytochalasin B. Corresponding to fig. 5A.

In the consequence, we added the following discussion in p. 24, l. 9 –p. 25, l. 4.

“Regarding the second-phase insulin secretion, our data have revealed that KIF5B is essential for a glucose-stimulated sequence of SFK and Rho-family GTPase activation (Fig. 4E–G), actin remodeling (Fig. 4A and B), changes in insulin granule kinetics and elevation of full-fusion exocytosis probability (Figs. 2 and 3), which collectively maintains the second-phase GSIS. Although the pSFK–Cdc42–PAK1–

Rac1 signaling pathway can partly crosstalk with Cav1.2-mediated Ca²⁺ transients (Veluthakal & Thurmond, 2021), the glucose-stimulated activation mechanism of SFK within 1 min is still elusive as far as we know. The significant decrease of PIP₂ in KIF5B-deficient beta cells (Fig. 5G and H) may affect PIP₂-dependent FAK activation (Zhou *et al*, 2015) as a prerequisite to FAK–SFK coactivation (Huveneers & Danen, 2009). We have previously identified that KIF26A and KIF21B kinesins regulate the FAK–SFK interaction and Rac1 nucleotide cycling, respectively, in neurons (Morikawa *et al*, 2018; Wang *et al*, 2018). If our prediction was correct, gene silencing or overexpression of those related kinesins could also enhance the performance of GSIS, which will be subjected to future research.”

4. Page 13: 2nd paragraph, starting with "To investigate the mechanistic link between KIF5B and Ca dynamics,....": Please provide the western blot data for Cav 2.3, BKca, Syntaxin-1, Na/Kappa ATPase, HSP70, HSP90, calnexin, LC3, STIP1, and Derlin.

Yes, we added western blotting panels in **Fig. 5A**, as kindly suggested.

Figure for referees not shown.

Fig. 5A–C. (A and B) Immunoblotting of a KIF5B-knockdown (KD) system in MIN6 cells using scrambled control (SC) and KD miRNA, for the indicated proteins (A); and its normalized quantification (B). *** $p < 0.001$; **** $p < 0.0001$; Welch's t test between KD and SC; $n = 3–8$. (C) Quantitative RT–PCR results of the KD system for the indicated genes. $n = 6$.

5. In Fig 6B, the IP blot: the portion of the IP blots are damaged (broken top part) near the bands of interest. So, I am unable to review the findings relevant to Fig 6B properly. Please provide undamaged sets of the IP blots for correct reviewing.

We would apologize for the systemic technical errors in data processing in previous Fig. 6B and C. We have thoroughly improved the data in new **Fig. 7C and D**.

Figure for referees not shown.

Fig. 7C and D. Vesicle IP of MIN6 cell lysates transduced with scrambled control (SC) and KIF5B-knockdown (KD) miRNAs in the presence of the proteasome inhibitor MG-132, precipitated using Cav1.2 or Kir6.2 antibodies or normal rabbit IgG (NRG) and immunoblotted for the indicated proteins (C), accompanied by quantification of Cav1.2-coprecipitated fractions (D). Note that Cav1.2-binding capacities of derlin1, calnexin-1, and Hsp90 chaperones and that of the adaptor protein STIP1 in KD cell lysates were significantly lower than those in SC cell lysates.

6. Page 15: "Although CaV1.2 and Hsp70 expression levels were both decreased in KD cells" The reduced level of HSP70 in input is not clear

Yes, you are right. The Hsp70 reduction in KIF5B-KD cells was somewhat rescued by MG treatment here as the abovementioned **Fig. 7C**, so that we have explicitly included the letters "Input (MG-132)" within the panel of **Fig. 7C**, and we also removed the expression from the text that now reads **"Interestingly, the association of Hsp70 with Cav1.2 tended to be unaltered."** (p. 18, ll. 5–6).

7. According to Fig 5J, reduction in Cav1.2 level in cKO cells can be restored by MG132 treatment. In Fig 6B, the cells are treated with MG132, but still, in KIF5B KO cells, the Cav1.2 level is markedly reduced. Please explain this discrepancy.

Yes, MG-132 increased the level of Cav1.2 expression of cKO cells but still not sufficient to reach the CT level. We have also improved the exposure time of immunoblot in **Fig. 6B** (new **Fig. 7C**). We explicitly added this argument as kindly suggested as the following:

"We then compared the chaperone- and co-chaperone-binding capacities of Cav1.2 and Kir6.2 by vesicle immunoprecipitation (IP) among the KIF5B-KD system (Fig. 7C and D), where the Cav1.2 level was partially rescued by MG-132 treatment." (p. 18, ll. 3–5).

8. Fig 6F: what type of cells? Control or cKO?

These are rescued cKO cells. We have explicitly added this on the figure panels as the following (new **Fig. 8A–C**).

Figure for referees not shown.

Figure 8. KIF5B recruits Hsp90 onto ER sheets for proper Cav1.2 protein folding

(A) Time lapse imaging in the bottom of a rescued primary cKO beta cell, expressing tagRFP-Hsp90 (red) and KIF5B-EYFP (green). Arrows, microdroplets. Note that two co-accumulated microdroplets appeared to exchange their components through a double-labeled dynamic tubule (open and closed arrowheads). Corresponding to Fig. EV2E and Movie EV5.

(B and C) Immunofluorescence microscopy in the bottom of primary cKO beta cells expressing tagRFP-Hsp90 (red) and KIF5B-EYFP (green), against Cav1.2 and Hsp70 (B) and against $K_{ir}6.2$ and Paxillin (C). Scale bars, 10 μ m. Arrows, co-accumulated microdroplets. Reproduced 5–10 times.

9. In Discussion: "We then detected a significant defect in CaV1.2 expression in the posttranslational level (Fig. 5)": In the result section, the authors discussed post-transcriptional changes.

Posttranscriptional is correct. We have improved this as kindly suggested as the following:

"Because the difference in transcription levels of Cav1.2, Cav2.3, and Kir6.2 were subtle (Fig. 5C), the reduction appeared to occur primarily at the post-transcriptional level." (p. 14, ll. 11–13)

10. In Fig 6 (Immunoprecipitation), the authors treated MIN6 cells with 10mM MG132 for 6 days. Please provide the evidence for cell viability after this treatment.

Thank you for pointing out this confusing expression in Methods section. The KD vector transfection was for 6 days to completely reduce KIF5B protein, but MG132 was added only for 6 h at the final stage so the cell viability was intact. We revised the text as kindly pointed out. It now reads **"Immunoprecipitation was performed as previously described (Ueno et al., 2011). MIN6 cells were transduced with scrambled control (SC) or KIF5B knockdown (KD) adenoviral vectors for 6 days. They were then treated by 10 μ M MG-132 for 6 h, and harvested using 10 mM HEPES [pH 7.4], 150 mM NaCl, 0.1% Triton-X 100, and protease inhibitors (Roche)." (p. 44, ll. 5–9)**

11. Fig. 6D and E: Please provide information about the type of cell treatments representing red and Blue dots (Fig. 6D) and bars (Fig. 6 E).

Sorry for these confusing color coding and we would also apologize the missing of the legend box in the previous version. Because the z-axis information was not so informative, we now remade the panel of new **Fig. 6I and J** as monochrome as follows, according to this insightful suggestion. In the bar graph, the color coding represents the genotypes similar to other figures.

Figure for referees not shown.

Fig. 6I and J. z-projection of proximity ligation assay in CT and cKO primary beta cells showing the protein binding between CaV1.2 and the indicated Hsp proteins (I); accompanied by quantification (J). $**p < 0.01$; $***p < 0.001$; Welch's *t* test, $n = 6$.

12. Co-immunoprecipitation studies (similar to Fig. 6B) using control (wild type) cells and KIF5B overexpressed cells would be helpful for a clear understanding of the proposed mechanism.

This is a very good suggestion. We tried to overexpress KIF5B in MIN6 cells but the increment in Cav1.2 levels was subtle possibly according to the shortage of Hsp70/Hsc70 protein expression that is required for kicking off the initial step of ER folding. So, we included Hsc70 which is the chaperone for the first step. This worked really well and we found that KIF5B/Hsc70 dual overexpression could significantly increase the expression of Cav1.2 protein in beta cells (**Fig. 7G and H**).

Then we conducted Cav1.2-IP assays to label the possibly involving ER chaperones in the immunoprecipitants (**Fig. 7I**), which were found to be massively missing from Cav1.2 binding in KIF5B-knockdown cells (**Fig. 7C and D reproduced in p. 29 of this document**).

Very intriguingly, we reproductively detected that Cav1.2 more strongly binds to calnexin and Hsp90 but more weakly binds to derlin-1. Because calnexin is the rate-limiting luminal chaperones whose gene silencing could significantly increase the Cav1.2 production rate, and that derlin-1 is also a luminal chaperone that facilitates Cav1.2 degradation, these results were consistent with our hypothesis that our KIF5B/Hsp machinery helps correctly folded Cav1.2 in the ER by the corporation of multiple chaperones in the ER to augment its production.

Figure for referees not shown.

Fig. 7G–I. (G and H) Cav1.2 immunocytochemistry of MIN6 cells that had been transduced with EYFP-KIF5B and/or TagRFP-Hsc70 or without them (NT; G); accompanied by their quantification (H). Scale bar, 5 μ m. ns, $p > 0.05$; ** $p < 0.01$, one-way ANOVA, $n = 5–13$. Arrow in G, enhanced Cav1.2 expression according to dual overexpression.

(I) Vesicle IP of non-transduced (NT) and KIF5B- and Hsc70-overexpressing (K5+H70 OE) MIN6 cell lysates against Cav1.2. Asterisks, tagged protein bands. The tagRFP-Hsc70 band was overlapped with a band of possibly ubiquitinated form. Reproduced twice.

We explicitly described this argument as the following.

“This KIF5B- and Hsc70-overexpression was found to alter the chaperone-binding capacities of Cav1.2 by vesicle IP. The binding capacities of Cav1.2 to calnexin-1 and Hsp90 were predominantly increased, but that to derlin-1 was decreased (Fig. 7I). The increase in Hsp90-bound Cav1.2 nicely supported our hypothesis that KIF5B enhances the Hsp70-to-Hsp90 chaperone exchange. Calnexin-1 serves for efficient protein production through the calnexin/calreticulin cycle (Kozlov & Gehring, 2020), and its gene silencing significantly increased the Cav1.2 production rate in mouse neonatal cardiomyocytes (Bousette *et al*, 2014). Thus, overproduced Cav1.2 protein may be partially accumulated in this rate-limiting step before exiting the ER. As derlin-1 behaves like a protein degrading chaperone for ERAD (Altier *et al*, 2011), the decrease in the Cav1.2–derlin-1 complex was also reasonable. Accordingly, KIF5B–Hsp system appeared to enhance the Cav1.2 protein folding via the ER chaperones.” (p. 19, ll. 7–18)

We deeply thank this reviewer for the insightful comments and discussions that considerably improved the quality of this paper.

References:

- Altier C, Garcia-Caballero A, Simms B, You H, Chen L, Walcher J, Tedford HW, Hermosilla T, Zamponi GW (2011) The Cavbeta subunit prevents RFP2-mediated ubiquitination and proteasomal degradation of L-type channels. *Nature neuroscience* 14: 173-180
- Bousette N, Abbasi C, Chis R, Gramolini AO (2014) Calnexin silencing in mouse neonatal cardiomyocytes induces Ca²⁺ cycling defects, ER stress, and apoptosis. *J Cell Physiol* 229: 374-383
- de la Cruz L, Puente EI, Reyes-Vaca A, Arenas I, Garduno J, Bravo-Martinez J, Garcia DE (2016) PIP2 in pancreatic beta-cells regulates voltage-gated calcium channels by a voltage-independent pathway. *American journal of physiology* 311: C630-C640
- Donnelly BF, Needham PG, Snyder AC, Roy A, Khadem S, Brodsky JL, Subramanya AR (2013) Hsp70 and Hsp90 multichaperone complexes sequentially regulate thiazide-sensitive cotransporter endoplasmic reticulum-associated degradation and biogenesis. *The Journal of biological chemistry* 288: 13124-13135
- Dufer M, Haspel D, Krippeit-Drews P, Aguilar-Bryan L, Bryan J, Drews G (2009) Activation of the Na⁺/K⁺-ATPase by insulin and glucose as a putative negative feedback mechanism in pancreatic beta-cells. *Pflügers Archiv : European journal of physiology* 457: 1351-1360
- Fridlyand LE, Jacobson DA, Philipson LH (2013) Ion channels and regulation of insulin secretion in human beta-cells: a computational systems analysis. *Islets* 5: 1-15
- Huveneers S, Danen EH (2009) Adhesion signaling - crosstalk between integrins, Src and Rho. *Journal of cell science* 122: 1059-1069
- Jing X, Li DQ, Olofsson CS, Salehi A, Surve VV, Caballero J, Ivarsson R, Lundquist I, Pereverzev A, Schneider T *et al* (2005) CaV2.3 calcium channels control second-phase insulin release. *The Journal of clinical investigation* 115: 146-154
- Kalwat MA, Thurmond DC (2013) Signaling mechanisms of glucose-induced F-actin remodeling in pancreatic islet beta cells. *Exp Mol Med* 45: e37
- Khaldi MZ, Guiot Y, Gilon P, Henquin JC, Jonas JC (2004) Increased glucose sensitivity of both triggering and amplifying pathways of insulin secretion in rat islets cultured for 1 wk in high glucose. *Am J Physiol Endocrinol Metab* 287: E207-217
- Kozlov G, Gehring K (2020) Calnexin cycle - structural features of the ER chaperone system. *The FEBS journal* 287: 4322-4340
- Liang T, Xie L, Chao C, Kang Y, Lin X, Qin T, Xie H, Feng ZP, Gaisano HY (2014) Phosphatidylinositol 4,5-bisphosphate (PIP2) modulates interaction of syntaxin-1A with sulfonylurea receptor 1 to regulate pancreatic beta-cell ATP-sensitive potassium channels. *The Journal of biological chemistry* 289: 6028-6040
- Moran Luengo T, Kityk R, Mayer MP, Rudiger SGD (2018) Hsp90 Breaks the Deadlock of the

Hsp70 Chaperone System. *Molecular cell* 70: 545-552 e549

Morikawa M, Tanaka Y, Cho HS, Yoshihara M, Hirokawa N (2018) The Molecular Motor KIF21B Mediates Synaptic Plasticity and Fear Extinction by Terminating Rac1 Activation. *Cell reports* 23: 3864-3877

Pertusa JA, Neshler R, Kaiser N, Cerasi E, Henquin JC, Jonas JC (2002) Increased glucose sensitivity of stimulus-secretion coupling in islets from *Psammomys obesus* after diet induction of diabetes. *Diabetes* 51: 2552-2560

Plante AE, Whitt JP, Meredith AL (2021) BK channel activation by L-type Ca(2+) channels Ca(V)1.2 and Ca(V)1.3 during the subthreshold phase of an action potential. *J Neurophysiol* 126: 427-439

Roche E, Farfari S, Witters LA, Assimacopoulos-Jeannet F, Thumelin S, Brun T, Corkey BE, Saha AK, Prentki M (1998) Long-term exposure of beta-INS cells to high glucose concentrations increases anaplerosis, lipogenesis, and lipogenic gene expression. *Diabetes* 47: 1086-1094

Schulla V, Renstrom E, Feil R, Feil S, Franklin I, Gjinovci A, Jing XJ, Laux D, Lundquist I, Magnuson MA *et al* (2003) Impaired insulin secretion and glucose tolerance in beta cell-selective Ca(v)1.2 Ca²⁺ channel null mice. *The EMBO journal* 22: 3844-3854

Trexler AJ, Taraska JW (2017) Regulation of insulin exocytosis by calcium-dependent protein kinase C in beta cells. *Cell Calcium* 67: 1-10

Veluthakal R, Thurmond DC (2021) Emerging Roles of Small GTPases in Islet beta-Cell Function. *Cells* 10

Wang L, Tanaka Y, Wang D, Morikawa M, Zhou R, Homma N, Miyamoto Y, Hirokawa N (2018) The Atypical Kinesin KIF26A Facilitates Termination of Nociceptive Responses by Sequestering Focal Adhesion Kinase. *Cell reports* 24: 2894-2907

Zhou J, Bronowska A, Le Coq J, Lietha D, Grater F (2015) Allosteric regulation of focal adhesion kinase by PIP(2) and ATP. *Biophysical journal* 108: 698-705

Dear Prof. Hirokawa

Thank you for the submission of your revised manuscript to our journal. It was sent back to the same referees who evaluated the original submission. Unfortunately, referee #2 was not responsive, but given that referee #1 and #3 consider the revised version much approved and support publication, I have decided to proceed with publication of your manuscript.

Since this is a resubmission, you had not received our formatting guidelines. I am therefore inviting you now to revise and format you study along our guidelines listed below. Once your revised manuscript has been submitted, we will perform a number of quality control steps. In order to be able to complete this procedure quickly, I list some specific requirements here:

1. We need the manuscript as .docx file and the figures submitted individually.
2. Summary is called Abstract, it may not exceed 175 words.
3. We need the Author Checklist completed. You can download it from our webpage (see also below).
4. Tanaka et al 2023 is a preprint. In the text cite it as (preprint: Tanaka et al, 2023). In the reference list add [PREPRINT] at the end of the citation.
5. Figure legends: specify whether 'n' is biological or technical replicates. The exact p-values need to be listed, not the range (e.g., $p < 0.01$). If a description is relevant for several panels like in Figure 2 the definition of scale bars and p-values, please add the prefix Data information:...
6. Movie legends need to be removed from the manuscript text. We need them as individual README.txt files and each legend needs to be zipped with its movie.
7. Remove the author contributions from the manuscript text and make sure that the contributions are specified in the manuscript submission system. The information provided there is the one that is used for publication.
8. The information on funding provided in the Acknowledgement section must match the one in the manuscript submission system. Again, the information in the system is the one that will be linked to your submission in PubMed. I think that "Japan IDDM Network" is missing. Please check.
9. We need a Data Availability section at the end of the Methods section. If you have no data to deposit, please state: This study includes no data deposited in external repositories.
10. We need a synopsis image that shows a summary or model of your findings (550 pixels width, up to 600 pixels high). Make sure to check the image at this final resolution, whether all is legible and looks good. We also need a 1-2 sentence summary and a few bullet points that describe the key findings of your study.
11. I think the rest looks already very good and is overall formatted according to our guidelines. But please check the instructions below.

- 1) a .docx formatted version of the manuscript text (including legends for main figures, EV figures and tables). Please make sure that the changes are highlighted to be clearly visible.
 - 2) individual production quality figure files as .eps, .tif, .jpg (one file per figure). Please download our Figure Preparation Guidelines (figure preparation pdf) from our Author Guidelines pages <https://www.embopress.org/page/journal/14693178/authorguide> for more info on how to prepare your figures.
 - 3) a .docx formatted letter INCLUDING the reviewers' reports and your detailed point-by-point responses to their comments. As part of the EMBO Press transparent editorial process, the point-by-point response is part of the Review Process File (RPF), which will be published alongside your paper.
 - 4) a complete author checklist, which you can download from our author guidelines (<<https://www.embopress.org/page/journal/14693178/authorguide>>). Please insert information in the checklist that is also reflected in the manuscript. The completed author checklist will also be part of the RPF.
 - 5) Please note that all corresponding authors are required to supply an ORCID ID for their name upon submission of a revised manuscript (<<https://orcid.org/>>). Please find instructions on how to link your ORCID ID to your account in our manuscript tracking system in our Author guidelines (<<https://www.embopress.org/page/journal/14693178/authorguide#authorshipguidelines>>)
 - 6) We replaced Supplementary Information with Expanded View (EV) Figures and Tables that are collapsible/expandable online. A maximum of 5 EV Figures can be typeset. EV Figures should be cited as 'Figure EV1, Figure EV2' etc... in the text and their respective legends should be included in the main text after the legends of regular figures.
- For the figures that you do NOT wish to display as Expanded View figures, they should be bundled together with their legends in a single PDF file called *Appendix*, which should start with a short Table of Content. Appendix figures should be referred to in

the main text as: "Appendix Figure S1, Appendix Figure S2" etc. See detailed instructions regarding expanded view here: <<https://www.embopress.org/page/journal/14693178/authorguide#expandedview>>

7) Please note that a Data Availability section at the end of Materials and Methods is now mandatory. In case you have no data that requires deposition in a public database, please state so instead of refereeing to the database.

See also < <https://www.embopress.org/page/journal/14693178/authorguide#dataavailability>>. Please note that the Data Availability Section is restricted to new primary data that are part of this study.

Additional information on source data and instruction on how to label the files are available <<https://www.embopress.org/page/journal/14693178/authorguide#sourcedata>>.

10) Figure legends and data quantification:

- the name of the statistical test used to generate error bars and P values,
 - the number (n) of independent experiments (please specify technical or biological replicates) underlying each data point,
 - the nature of the bars and error bars (s.d., s.e.m.)
- If the data are obtained from n {less than or equal to} 5, show the individual data points in addition to the SD or SEM.
- If the data are obtained from n {less than or equal to} 2, use scatter blots showing the individual data points.

11) Our journal encourages inclusion of *data citations in the reference list* to directly cite datasets that were re-used and obtained from public databases. Data citations in the article text are distinct from normal bibliographical citations and should directly link to the database records from which the data can be accessed. In the main text, data citations are formatted as follows: "Data ref: Smith et al, 2001" or "Data ref: NCBI Sequence Read Archive PRJNA342805, 2017". In the Reference list, data citations must be labeled with "[DATASET]". A data reference must provide the database name, accession number/identifiers and a resolvable link to the landing page from which the data can be accessed at the end of the reference. Further instructions are available at <<https://www.embopress.org/page/journal/14693178/authorguide#referencesformat>>.

12) All Materials and Methods need to be described in the main text using our 'Structured Methods' format, which is required for all research articles. According to this format, the Methods section includes a Reagents and Tools Table (listing key reagents, experimental models, software and relevant equipment and including their sources and relevant identifiers) followed by a Methods and Protocols section describing the methods using a step-by-step protocol format. The aim is to facilitate adoption of the methodologies across labs. More information on how to adhere to this format as well as a downloadable template (.docx) for the Reagents and Tools Table can be found in our author guidelines:

13) As part of the EMBO publication's Transparent Editorial Process, EMBO Reports publishes online a Review Process File to accompany accepted manuscripts. This File will be published in conjunction with your paper and will include the referee reports, your point-by-point response and all pertinent correspondence relating to the manuscript.

Yours sincerely,

Referee #1:

I have no more questions

Referee #3:

The authors addressed all my concerns and revised the manuscript accordingly. I do not have any more questions/suggestions.

All editorial and formatting issues were resolved by the authors.

Prof. Nobutaka Hirokawa
The University of Tokyo
Department of Cell Biology and Anatomy, Graduate School of Medicine
7-3-1 Hongo
Bunkyo-ku, Tokyo 113-0033
Japan

Dear Prof. Hirokawa,

I am very pleased to accept your manuscript for publication in the next available issue of EMBO reports. Thank you for your contribution to our journal.

Yours sincerely,
